# Comparison of Inorganic Chlorine in the Antarctic and Arctic lowermost stratosphere by separate Late Winter aircraft measurements

Markus Jesswein[1], Heiko Bozem[2], Hans-Christoph Lachnitt[2], Peter Hoor[2], Thomas Wagenhäuser[1], Timo Keber[1], Tanja Schuck[1], and Andreas Engel[1]

[1]University of Frankfurt, Institute for Atmospheric and Environmental Sciences, Frankfurt, Germany
[2]Johannes Gutenberg University of Mainz, Institute for Atmospheric Physics, Mainz, Germany

**Correspondence:** Markus Jesswein (jesswein@iau.uni-frankfurt.de)

**Abstract.** Stratospheric inorganic chlorine ($Cl_y$) is predominantly released from long-lived chlorinated source gases and, to a small extent, very short-lived chlorinated substances. $Cl_y$ includes the reservoir species (HCl and $ClONO_2$) and active chlorine species (i.e. $ClO_x$). The active chlorine species drive catalytic cycles that deplete ozone in the polar winter stratosphere. This work presents calculations of inorganic chlorine ($Cl_y$) derived from chlorinated source gas measurements on board the High

Altitude and Long Range Research Aircraft (HALO) during the Southern hemisphere Transport, Dynamic and Chemistry (SouthTRAC) campaign in austral late winter and early spring 2019. Results are compared to $Cl_y$ of the Northern Hemisphere derived from measurements of the POLSTRACC-GW-LCYCLE-SALSA (PGS) campaign in the Arctic winter of 2015/2016. A scaled correlation was used for PGS data, since not all source gases were measured. Using the SouthTRAC data, $Cl_y$ from a scaled correlation was compared to directly determined $Cl_y$ and agreed well. An air mass classification based on in situ $N_2O$

measurements allocates the measurements to the vortex, the vortex boundary region, and mid-latitudes. Although the Antarctic vortex was weakened in 2019 compared to previous years, $Cl_y$ reached 1687±19 ppt at 385 K, therefore up to around 50 % of total chlorine was found in inorganic form inside the Antarctic vortex, whereas only 15 % of total chlorine was found in inorganic form in the southern mid-latitudes. In contrast, only 40 % of total chlorine was found in inorganic form in the Arctic vortex during PGS and roughly 20 % in the northern mid-latitudes. Differences inside the two vortices reach as much as 540 ppt,

with more $Cl_y$ in the Antarctic vortex 2019 than in the Arctic vortex 2016 (at comparable distance to the local tropopause). To our knowledge, this is the first comparison of inorganic chlorine within the Antarctic and Arctic polar vortices. Based on the results of these two campaigns, the differences in $Cl_y$ inside the two vortices are substantial and larger than the inter-annual variations previously reported for the Antarctic.

## 1 Introduction

The Antarctic ozone hole is a recurring event, which was first documented by Farman et al. (1985) and has been observed annually since the 1980s. Polar ozone depletion is predominantly driven by anthropogenic chlorine and bromine from long-lived halogenated species (Molina and Rowland, 1974; Engel and Rigby, 2018). The primary mechanisms for the depletion

of ozone ($O_3$) in the polar stratosphere are the catalytic cycles with halogen-containing free radicals as chain carriers (Molina et al., 1987). Chlorine substances involved in rapid ozone depletion are Cl, $Cl_2$, ClO, and ClOOCl, and can be summarized as

$ClO_x$. Additionally, hydrogen chloride (HCl) and chlorine nitrate ($ClONO_2$) contribute to ozone depletion as they enable the production of active chlorine through heterogeneous reactions on polar stratospheric clouds (PSC) during polar winter with low temperatures (e.g., Crutzen and Arnold, 1986; Molina et al., 1987; Solomon, 1999). They are therefore called reservoir species. Chemically active chlorine ($ClO_x$) and the reservoir gases together form the total inorganic chlorine ($Cl_y$), also called available chlorine. Equivalent effective stratospheric chlorine (EESC) is a simple metric that sums the effect of ozone depleting

substances (ODS) as an equivalent amount of inorganic chlorine in the stratosphere (Newman et al., 2007; Daniel et al., 1995). Changes to the EESC are mainly due to $Cl_y$, as $Br_y$ makes up a smaller fraction (Strahan et al., 2014).

The size of the Antarctic ozone hole varies and depends on the amount of $Cl_y$ and on stratospheric temperature and dynamics (e.g., Newman et al., 2004). However, $Cl_y$ data are sparse in the polar stratosphere and likewise the amount of measurements of total organic chlorine ($CCl_y$). In contrast, there are many more observations from e. g. remote sensing instruments of nitrous

oxide ($N_2O$), which can be used to determine $Cl_y$. A common tool to determine $Cl_y$ is the usage of scaled correlations. Strahan et al. (2014) used Microwave Limb Sounder (MLS) $N_2O$ measurements and a scaled correlation between $N_2O$ and $Cl_y$ from Schauffler et al. (2003) to show that inter-annual variability of $Cl_y$ for 2004–2012 (-200 to + 150 ppt) can be up to 10 times larger than the expected 20–22 ppt year$^{-1}$ decline rate due to the Montreal Protocol. Strahan and Douglass (2018) again used MLS measurements of $O_3$, HCl, and $N_2O$ to show that Antarctic $Cl_y$ levels have decreased by $223 \pm 93$ ppt over a 9 year period

(2013–2016 compared to 2003–2007), equivalent to an annual rate of $25 \pm 10$ ppt year$^{-1}$ ($\sim 0.8\,\%$ year$^{-1}$). It is thus important to know whether and how correlations can be transferred from another time period and possibly another region (other hemisphere). Due to the phase-out of the long-lived chlorinated species, $Cl_y$ shows a long term negative trend (e.g., Newman et al., 2007), whereas $N_2O$ exhibits a positive trend (Engel and Rigby, 2018). This leads to a changing relationship between $Cl_y$ and $N_2O$ with time, which must be taken into account. The concept of mean arrival time ($\Gamma^*$) can be used to normalize correlations of

chemically active tracers and scale them to the time of interest (Plumb et al., 1999). The normalized correlations do not change with time and the resulting scaled correlations are used to calculate $Cl_y$.

Ozone destruction in the stratosphere is closely linked to the polar vortex. Due to a temperature difference and consequently to a latitudinal pressure gradient between the polar and mid-latitude stratosphere (e.g., Schoeberl and Hartmann, 1991), a state with a strong westerly wind in the stratosphere is established (polar night jet). This jet acts as a transport barrier, leading to

strong latitudinal gradients of potential vorticity and long-lived substances like $N_2O$ (e.g., Hartmann et al., 1989). This isolation of the vortex leads to different concentrations of trace gases within the vortex compared to those in the stratosphere of the mid-latitudes. The effect is further enhanced by diabatic descent over the winter, leading to substantially different distributions of trace gases inside and outside the vortex on the same potential temperature ($\Theta$) surface. The polar vortex core can thus be described as a quasi-isolated vessel, separated from the mid-latitude stratosphere by the vortex boundary region. In order to

differentiate between air masses inside and outside the vortex, a classification of the measurements is needed.

In this study, inorganic chlorine ($Cl_y$) was quantified in the Arctic and Antarctic vortex. Calculations of $Cl_y$ are based only on long-lived chlorinated substances. There is an additional contribution to total stratospheric chlorine from the very short-lived

chlorinated substances. Engel and Rigby (2018) estimated a contribution of 115 (75–169) ppt from very short-lived substances for 2016. Hossaini et al. (2019) estimated a contribution of about $111 \pm 22$ ppt, of which $13 \pm 4.6$ ppt are already in inorganic form, which is not considered in this analysis. A new air mass classification system was used for this purpose, based on high resolution in situ measurements during the campaigns, to map measurements to the vortex, vortex boundary region and mid-latitudes. Results of the SouthTRAC campaign from the Antarctic winter/spring 2019 are used to compare $Cl_y$ of the Southern Hemisphere with $Cl_y$ of the Northern Hemisphere from measurements of the POLSTRACC-GW-LCYCLE-SALSA (PGS) campaign in the Arctic winter of 2015/2016. An overview of the atypical Antarctic vortex 2019 can be found in Wargan et al. (2020). The evolution of the 2015/2016 Arctic vortex is reported in Manney and Lawrence (2016). Since not all source gases were measured during PGS, a scaled correlation was used and showed the capability of this method as a proxy for sparse data in comparison to the determination from the source gases used for SouthTRAC measurements. Sect. 2 is a brief introduction to the SouthTRAC campaign and the observations used for this study. Sect. 3 explains the identification of vortex, vortex boundary, and mid-latitude region. The derivation of inorganic chlorine and the comparison of the methods, the distribution during the late Antarctic winter of 2019, and the comparison of Arctic and Antarctic $Cl_y$ is discussed in Sect 4. Sect 5. sums up and concludes the findings.

## 2 The SouthTRAC campaign

In late winter and early spring of 2019, the Southern hemisphere Transport, Dynamics, and Chemistry (SouthTRAC) campaign took place to investigate dynamical and chemical composition aspects of the Antarctic upper troposphere and lower stratosphere (UTLS) and gravity waves up to the mesosphere (Rapp et al., 2020). Flights were performed with the German High Altitude and LOng Range Research Aircraft (HALO), which is capable of reaching an altitude of around 14.5 km or 420 K potential temperature. To meet the dynamical and chemical objectives (see https://www.pa.op.dlr.de/southtrac/science/scientific-objectives/), the aircraft was based in Rio Grande, Argentina (RGA, 53°S, 67°W). Thus, regions of gravity-wave breaking (Southern Atlantic and Eastern Pacific) and Antarctica were in the range of the aircraft. The campaign was split in two phases. The first phase took place from September 6[th] to October 9[th], 2019 to target the dynamical objectives (e.g., Rapp et al., 2020). The second phase took place from November 2[nd] to 15[th], 2019 to, among others, sample polar vortex remnants. Furthermore, the transfer flights were part of the scientific flights and provide additional information for all objectives.

HALO performed 23 scientific flights with in total 183 hours of measurement time. Nine of these flights were transfer flights from Oberpfaffenhofen (EDMO), Germany, to Rio Grande (RGA), Argentina, and back via Sal (SID), Cabo Verde, and Buenos Aires (EZE), Argentina (see Fig. 1a). Within the first transfer from EDMO to RGA, there was an additional local flight operated from SID. The remaining flights were local flights with ten in the first phase and three in the second phase. The second phase was terminated early due to technical problems but still provided 27 hours of measurements. Thus, it was possible to investigate a region of around 36–70°S and 32–84°W (see Fig. 1b). The airplane reached a maximum potential temperature of 409 K during the campaign.

The following is a brief explanation of the meteorological data and the instruments and types of measurements used for this work.

## 2.1 Meteorological data

HALO was equipped with a wide range of in situ and remote-sensing instruments. In addition to the scientific instruments installed for the measurement campaign, the Basic Halo Measurements and Sensor System (BAHAMAS) is part of HALO. BAHAMAS is installed permanently and provides meteorological and aircraft parameters along the flight trajectory (DLR, 2020).

The local tropopause information along the flight tracks of HALO was created using the Chemical Lagrangian Model of the Stratosphere (CLaMS) (e.g., Grooß et al., 2014). The underlying meteorological data are taken from ECMWF ERA-5 (Hersbach et al., 2020). In this work, the potential vorticity (PV) based dynamical tropopause is used (e.g., Gettelman et al., 2011), taking 2 PVU (potential vorticity unit) for the dynamical tropopause. Since the PV tropopause is not physically meaningful in the tropics, the potential temperature level of 380 K was taken as the tropopause if the 2 PVU level lies above.

## 2.2 Halocarbons and SF$_6$

The Gas chromatograph for Observational Studies using Tracers (GhOST) is a two-channel gas chromatographic instrument. The first channel combines an isothermally operated gas chromatograph (GC) with an electron capture detector (ECD) to measure SF$_6$ and CFC-12 at a time resolution of 1 min (hereinafter referred to as GhOST-ECD). A similar set-up was used during the SPURT campaign (Bönisch et al., 2009; Engel et al., 2006). The second channel is a combination of temperature programmed GC with a quadrupole mass spectrometer (QP-MS, hereinafter referred to as GhOST-MS). Because of very small mole fractions of the halocarbons, a cryogenic pre-concentration system is installed prior to the GC (Obersteiner et al., 2016; Sala et al., 2014). GhOST was operated successfully during several aircraft campaigns to mainly target brominated halocarbon species, as discussed in Keber et al. (2020). For the SouthTRAC campaign, the ionization mode was changed from negative chemical ionization (NCI) to electron impact ionization (EI) to record full mass spectra. For each substance, one molecular fragment is selected for which the chromatographic peak is not disturbed by other substances. Furthermore, the MS was operated in selected ion monitoring (SIM), scanning pre-selected mass fractions at a preset retention time window. A larger sample volume is needed in EI mode compared to NCI mode. The change of ionization decreased the time resolution from 4 min to 6 min per measurements cycle, of which 147 seconds are needed for sampling air. The performance of the GhOST-MS channel for the chlorinated substances used in this work is shown in Table 1. Displayed are the precisions and detection limits measured shortly before the campaign in the laboratory. Additionally, based on in flight calibration, a precision during the flight can also be calculated. As the conditions in the airplane are more variable than in a laboratory, especially when changing the flight level, this affects the precisions of the measurements. The frequency of calibration measurements during a flight is much lower than in the laboratory, making it less stable than the laboratory value. Therefore, precision drops for most of the substances by up to a factor of 4. The exceptions are CFC-11, CFC-113, and methyl chloroform. Methyl chloroform shows a significantly better precision during the campaign, whereas the precision of CFC-12 and CFC-113 measurements was

much poorer. It is difficult to determine exactly what the poorer precisions of these two substances can be attributed to. The chromatographic peak of CFC-11 is very narrow and variable environmental conditions (due to changes in altitude, pressure, and temperature in the cabin) have an influence on the peak shape. The amount of water in the analysis system is also important and is kept as low as possible by drying before pre-concentration. As the chromatographic peak of CFC-113 is close to the chromatographic peak of water, small changes in water can affect the chromatographic peak of CFC-113. CFC-12 and $SF_6$ with the GhOST-ECD channel were measured with a precision of 0.2 % and 0.64 %, respectively. The instrument was tested for non-linearities and memory effect and correction was done where necessary (see Sala et al. (2014) for details). Mixing ratios in this work are reported as dry mixing ratios on AGAGE (Advanced Global Atmospheric Gases Experiment) scales.

## 2.3 $N_2O$

Measurements of $N_2O$ were performed with the University of Mainz Airborne Quantum Cascade Laser-spectrometer (UMAQS), which also provided data for $CH_4$, CO, $CO_2$ and OCS during SouthTRAC. The instrument is based on direct absorption spectroscopy using a continuous-wave quantum cascade laser with a sweep rate of 2 kHz (Müller et al., 2015). During SouthTRAC the instrument was calibrated in situ against two standards of different concentrations, which are compared against primary standards (NOAA) prior to and after campaign phases.

Under typical flight conditions at flight level $N_2O$ was measured with an overall uncertainty of 0.6 ppb relative to the calibration standards. The noise of the 1 Hz data was 0.1 ppb (1-sigma). Note that this is an upper limit since the data are corrected post-flight for drift effects based on the in-flight calibrations.

## 3 Defining vortex, vortex boundary and mid-latitude region

Chlorine activation tends to occur in the coldest regions of the stratosphere and is therefore typically co-located with the polar vortex. Furthermore, as the polar night jet acts as a barrier, air composition is different inside and outside the vortex. High concentrations of reactive halogenated substances can be maintained inside the vortex because there is little mixing with the surrounding area. During the HALO flights, the aircraft encountered air masses with different characteristics due to their origin. To make systematic conclusions about the distribution of trace gases, a reliable, accurate method of separating the measurements in terms of their region of origin is needed. For this reason, the following describes how air masses have been classified using highly resolved in situ measurements.

### 3.1 Air mass classification by in situ measurements

The maximum gradient of PV is a commonly used indicator to define the location of the vortex edge, also known as the Nash criterion (Nash et al., 1996). PV is a model-derived quantity. Although the underlying meteorological reanalyses have a fairly high resolution these days (e.g., Hersbach et al., 2020), small-scale features like vortex filaments with different chemical compositions may not be well resolved. In this work, an extended version of the vortex definition by Greenblatt et al. (2002) is used instead. The technique by Greenblatt et al. (2002) uses the tight correlation between $N_2O$ and potential temperature to

determine the inner edge of the vortex boundary. $N_2O$ can be measured in situ with a high time resolution to reveal small scale

structures in the atmosphere. As already mentioned in the introduction, air inside the polar vortex has a substantially different composition regarding trace gases than air outside the vortex. A tracer like $N_2O$ exhibits a horizontal gradient across the vortex edge in the stratosphere with lower mixing ratios inside the vortex and higher mixing ratios outside the vortex. In addition, $N_2O$ has small variability inside the vortex on constant isentropic surfaces (variability of about 6 ppb (Greenblatt et al., 2002)). This is an indication of well mixed air inside the polar vortex due to the long isolation in polar winter. The vertical profile of

$N_2O$ in the mid-latitudes shows a weak gradient but a high variability as it is influenced by both tropical and polar air (Krause et al., 2018; Marsing et al., 2019). In between there is a transition region (vortex boundary region), which is influenced by the vortex as well as by mid-latitudes. Towards tropopause altitudes, the transport barrier of the polar vortex disappears, and a classification is not possible.

Based on the method of Greenblatt et al. (2002), one flight is chosen to generate a vortex reference profile. This flight should

ideally be completely in the vortex. However, during the SouthTRAC campaign there was no flight that only sampled vortex air. In addition, there is an interest in not only distinguishing between vortex and non-vortex air, but also in assigning the campaign measurements to the vortex, vortex boundary region, and mid-latitudes. For this reason, several flights were used to create reference profiles for the vortex and mid-latitudes. The composition of the lowermost stratosphere is affected by diabatic descent inside and outside the polar vortex and quasi-isentropic mixing with air from lower latitudes. In addition, mixing

at the extratropical tropopause affects the lowest 20–25 K above the local tropopause (Hoor et al., 2004, 2005). Therefore, classification was done with two vertical coordinates, potential temperature ($\Theta$) and potential temperature above the local tropopause ($\Delta\Theta$).

The vortex reference profile (see Fig. 2) was generated from all flights that are assumed to contain measurements within vortex air. Data from these flights were pre-filtered by taking only the measurements polewards of 60°S equivalent latitude

(Butchart and Remsberg, 1986) and 20 K above the local tropopause. With an iterative filter procedure (see appendix A) the lower envelope of the remaining measurements is obtained and is used to generate the vortex profile function (Werner et al., 2010). For the mid-latitude profile (see Fig. 2), all flights were taken into account, focusing only on measurements between 40° and 60° S equivalent latitude and again 20 K above the local tropopause. This time, the upper envelope of the measurements was evaluated by the iteration procedure to build a reference profile function for the mid-latitudes. As an intermediate step to

the final profiles the measurements of the lower and upper envelope are binned in 5 Kelvin intervals of $\Theta$ or $\Delta\Theta$ (see Figure S 2 c and d). Mean values of the binned profiles are then used to generate a polynomial fit function for the vortex profile and the mid-latitude profile (Figure S 3). The two reference profiles in $\Theta$-coordinates are displayed in Figure 2a, the two reference profiles in $\Delta\Theta$-coordinates in Figure 2b.

In general, it cannot be assumed that a single $N_2O$ vortex profile can be representative for the entire winter. Subsidence of

185 the vortex air by several kilometers due to radiative cooling (Schoeberl and Hartmann, 1991) leads to a changing $N_2O$ profile throughout the polar winter. In the lower stratosphere of the Southern Hemisphere, descent generally stops around mid-October (Manney et al., 1994). However, $N_2O$ data of the SouthTRAC flights did not reveal strong diabatic descent during the time of the campaign (below $\Theta = 400$ K). Therefore, only one reference vortex profile was generated for the campaign.

A vortex and mid-latitude reference $N_2O$ data set ($N_2O_{vor}$ and $N_2O_{mid}$) can be calculated from $\Theta$ or $\Delta\Theta$ by using the fit function for the vortex and mid-latitude profiles for every measurement point of the UMAQS instrument for all flights. The following then applies for each $N_2O$ measurement: if the mixing ratio is below the respective $N_2O_{vor}$ plus the prescribed vortex cutoff, then it is assigned to the vortex. Otherwise, if the mixing ratio is above the respective $N_2O_{mid}$ minus the associated variability, then it is assigned to the mid-latitudes. Mixing ratios above the respective $N_2O_{vor}$ plus the prescribed vortex cutoff and below the respective $N_2O_{mid}$ minus the associated variability are assigned to the boundary region. Measurements for which the respective $N_2O_{vor}$ plus the prescribed vortex cutoff and the $N_2O_{mid}$ minus the associated variability overlap cannot be uniquely classified and are assigned to both the vortex and the mid-latitudes in later analysis. For the prescribed vortex cutoff, the value of 20 ppb proposed by Greenblatt et al. (2002) was used. The associated variability of the mid-latitude profile was set to 15 ppb, as the variability in $N_2O$ in the mid-latitudes is roughly 10% (Strahan et al., 1999).

## 3.2 Overview of the sample regions

In 2019, extraordinary meteorological conditions led to a sudden rise in stratospheric temperatures over Antarctica. This minor sudden stratospheric warming (minor SSW) event affected the shape, location and strength of the polar vortex. From mid-August to early September 2019, the polar vortex was displaced and weakened towards the eastern South Pacific and South America (Safieddine et al., 2020; Wargan et al., 2020). The SouthTRAC campaign flights took place from early September to early October and in the first half of November; thus they took place shortly after the minor SSW event and captured the late winter evolution of the Antarctic polar vortex.

Figure 3 displays an overview of air mass classification of the local flights of the SouthTRAC campaign (classification in $\Theta$-coordinates). Measurements below 20 K of $\Delta\Theta$ are not classified and are not taken into account here. There are no $N_2O$ measurements available from flight ST08 on September 11th, so no classification was possible. Vortex air sampling varies from flight to flight, depending on the objective of each flight. The vortex and boundary regions were sampled in both phases of the campaign. The first phase includes some flights that predominantly sampled the vortex or vortex boundary region (e. g., flight ST15 on 29 September or flight ST16 on 30 September). In general, vortex air represents about 23 % of flight time in the stratosphere and vortex boundary air about 14 %. More than half (56 %) of the flight time in the stratosphere was in mid-latitude air. The use of the $\Delta\Theta$-coordinates for the air mass classification leads to similar percentage division (not shown here).

## 4 Inferred inorganic chlorine

## 4.1 Up-sampling GhOST-MS measurements

The GhOST-MS measurements have a time resolution of 6 min of which the enrichment and therefore the sampling of air takes around 147 seconds. With a maximum cruising speed of around 258 m/s, this means that air is sampled along a distance of approximately 38 km per measurement during enrichment. This could lead to a rather coarse resolution where fine structures like filaments and small-scale dynamical perturbations are sometimes not well resolved.

Measuring CFC-12 in both the ECD and MS channels of the instrument allows the measurements of the organic source gases to be up-sampled by using the better-resolved measurements of CFC-12 from the ECD channel. Measurements of CFC-12 in the ECD channel not only have a higher time resolution of 1 min, but they also have better precision than data from the MS channel. As shown in Figure 4, the correlation between CFC-12 measurements of the two channels for all flights is linear over the whole range of mixing ratios, with a coefficient of determination of $R^2 = 0.968$. Firstly, for each organic source gas, a

linear or polynomial fit function is calculated based on the correlation with CFC-12 measurements in the GhOST-MS channel for all flights (correlations contained in the supporting information). Secondly, these fit functions are then used together with the CFC-12 measurements of the ECD channel to calculate the up-sampled mixing ratios of the organic source gases. Flight ST14 from 26 September in Figure 5 is an example to demonstrate the benefit of the up-sampling. Displayed are the original measurements of CFC-11 of the MS channel as well as the up-sampled CFC-11 values. The background colors indicate to

which region the samples can be assigned (classification in Θ-coordinates), as described in Sect. 3. The up-sampled values show higher variability and follow well the classification by region. Especially with the sharp gradients, e. g. at 04:10 UTC and at 05:50 UTC in Figure 5, the original lower-resolution data did not capture well the transitions between the regimes, compared to the up-sampled data. In the following, the up-sampled data are used for further evaluations.

## 4.2    Semi-direct and indirect calculation of inorganic chlorine

Organic chlorine (CCl$_y$) can be calculated directly from the up-sampled GhOST-MS measurements. Thus, Cl$_y$ can be calculated from Eq. 1 if the mixing ratios of the major chlorine-containing substances at the stratospheric entry point (Cl$_{total}$) are known.

$$\chi_{\mathrm{Cl_y}} = \chi_{\mathrm{Cl_{total}}} - \chi_{\mathrm{CCl_y}} \tag{1}$$

Air enters the stratosphere predominantly through the tropical tropopause layer (TTL). During transport into and within the

stratosphere, an air parcel exhibits irreversible mixing and cannot be regarded as conserved (Hall and Plumb, 1994). Instead, an air parcel in the stratosphere consists of a multitude of components with different transit times, representing their travel times since they entered the stratosphere. The distribution of the transit times is called the age spectrum $G$ and the first moment is the mean age $\Gamma$ (Hall and Plumb, 1994). The concept of the age spectrum can be used to determine mean age values based on observations of chemically inert tracers in the stratosphere. For this purpose, in addition to the age spectrum, tropospheric

time series of the inert tracers are required (Engel et al., 2002). This was done for the SouthTRAC campaign by using SF$_6$ measurements of the GhOST-ECD and tropospheric time trends taken from the AGAGE (Advanced Global Atmospheric Gases Experiment) Network (Prinn et al., 2018). Since only the mean age is given, a width parameterization is used to derive the age spectrum by using the ratio of moments ($\Delta^2$ / $\Gamma$). Hauck et al. (2019) showed that the ratio of moments undergoes seasonal variability and is probably much larger than previously implemented values (e. g. Engel et al., 2002; 0.7 year). A

ratio of moments of 1.25 years is chosen here. The age spectrum together with the tropospheric time trend of the substances of interest can be used to calculate the stratospheric mixing ratio that would be present without chemical degradation, which thus

represents the entry mixing ratio. In the following, $Cl_y$ derived from the difference between the estimated entry mixing ratios and observed $CCl_y$ from the in situ measurements (see Eq. 1) is referred to as the semi-direct calculation of $Cl_y$.

For the case where measurements of all major chlorine-containing substances are not available, $Cl_y$ has in the past been calculated indirectly based on correlations derived from previous measurement campaigns. This also applies to $Cl_y$ from the Northern Hemisphere, later in the analyses (see section 4.4). For instance Wetzel et al. (2015) and Marsing et al. (2019) used $Cl_y$ based on a correlation derived from two balloon flights inside the Arctic polar vortex in 2009 and 2011 from a cryogenic whole-air sampler (Engel et al., 2002). In order to account for tropospheric trends, the correlations between CFC-12 and the other long-lived substances were adapted with a modified method described in Plumb et al. (1999) using the mean arrival time $\Gamma^*$ to derive a correlation function valid for the respective time when the correlation is applied. Plumb et al. (1999) showed that the age spectrum of an inert tracer is not well suited to describe the propagation of chemically active tracers into and through the stratosphere. They introduced a modified age spectrum, called the normalized arrival time distribution $G^*$, which combines chemical loss and transport. The mean arrival time $\Gamma^*$ represents the first moment of this distribution. The mean arrival time $\Gamma^*$ for all the relevant chlorine substances can be parameterized in terms of stratospheric lifetime $\tau$ and mean age $\Gamma$ (Plumb et al., 1999). Using $G^*$ instead of $G$ and therefore $\Gamma^*$ instead of $\Gamma$ is better suited for chemically active tracers, because the tail of the transit time distribution is less weighted, especially for substances with shorter stratospheric lifetimes (Engel et al., 2018a; Ostermöller et al., 2017). To transfer the correlations from the balloon observations in 2009 and 2011 to the measurements during SouthTRAC in 2019, the observed mixing ratios are first divided by the respective estimated entry values to derive the normalized mixing ratios. The entry values are calculated by the modified age spectrum and tropospheric time trends from the AGAGE Network for the time of the balloon measurements. Multiplying the normalized mixing ratios by the entry mixing ratio for the time during SouthTRAC then allows a comparison to the directly determined correlations during SouthTRAC. Here, we compare the directly observed correlation from SouthTRAC to the indirectly determined correlations based on previous balloon observations transferred to 2019. Note that the indirectly determined values are based on observations that were not only performed about 10 years earlier but that were also from the Northern Hemisphere instead of the Southern Hemisphere. Figure 6 displays scaled correlations from the balloon observations (red) and correlations from the SouthTRAC data (black) of three long-lived substances against CFC-12. The balloon-based correlations correspond well to the correlations measured during the SouthTRAC campaign. Thus, the balloon-based correlations can be used to determine $CCl_y$ from CFC-12 alone. As already mentioned earlier, $Cl_{total}$ is also needed for the calculation of $Cl_y$ (see Eq. 1). For this, the mean age values derived for the balloon measurements are used and $Cl_{total}$ is calculated for the conditions during SouthTRAC. $Cl_y$ is then derived as the difference between $Cl_{total}$ and $CCl_y$.

With the good agreement between observed correlations and scaled correlations from the balloon measurements and the previously described determination of $Cl_y$, we explore whether $Cl_y$ can be successfully estimated from CFC-12 alone. That is, a correlation function for the conditions during Antarctic late winter 2019 has then been derived for the indirect calculation of $Cl_y$ as a function of CFC-12 mixing ratios (Eq. 2). The coefficients for the correlation function with CFC-12 as the reference substance, based on the balloon measurements, can be taken from Table 2. In addition, the fit coefficients are given if one wants to use $N_2O$ as the reference. $N_2O$ shows a compact correlation with long-lived chlorinated substances and has been used in

many publications for the determination of $Cl_y$ (e.g., Schauffler et al., 2003; Strahan et al., 2014; Strahan and Douglass, 2018). Using CFC-12 from the GhOST-ECD channel and $N_2O$ from the UMAQS instrument, we obtain comparable values for $Cl_y$ (see figure S 7 in the supporting information). In the following, CFC-12 from the GhOST-ECD channel is used as the reference in Eq. 2 for the indirect determination of inorganic chlorine.

$$\chi_{Cl_y} = c_0 + c_1 \chi_{ref} + c_2 (\chi_{ref})^2 \qquad\qquad (2)$$

Figure 7 shows semi-directly and indirectly determined inorganic chlorine as a function of mean age. $Cl_y$ values were binned in intervals of 0.2 years and the mean values are displayed. For both methods, inorganic chlorine increases with mean age of air, as more molecules of the organic source gases are converted to the inorganic form. The difference between the two methods is rather small, with less than around 30 ppt difference between 1 and 4 years of mean age and a maximum difference of about 65 ppt at 5 years of mean age. Recent research suggests that $SF_6$-based mean age is biased because its suggested lifetime has been overestimated (e.g., Ray et al., 2017). As a guideline, Figure 7 additionally shows a corrected mean age of air using one of the linear fit functions from Leedham Elvidge et al. (2018), based on a comparison of $SF_6$-based mean age with a combined mean age based on five alternative age tracers. The fundamental picture does not change, however, hence we use the uncorrected mean age of air. The small deviation over the entire range of mean age indicates that adapted correlations from previous measurement campaigns and also from the Northern Hemisphere lead to comparable values in inorganic chlorine determined for the Southern Hemisphere. Hence the metric can be used for the calculations of $Cl_y$ in the case where only measurements of CFC-12 are available. Since it was possible during SouthTRAC to measure the organic source gases, the $Cl_y$ determined semi-directly from the measurements was used for further evaluation. However, the good comparability of the two methods offers the possibility to compare $Cl_y$ from different measurement campaigns, which differ regarding the number of measured chlorinated substances (see section 4.4). With the semi-directly determined $Cl_y$ during SouthTRAC, correlation functions can be determined. Table 2 contains the coefficients for the correlation functions based on CFC-12 and $N_2O$ as references. The correlation functions are limited to the minimum mixing ratio of the respective reference substance taken during the SouthTRAC campaign.

### 4.3  Chlorine partitioning in the Antarctic winter 2019 lower stratosphere

Since inorganic chlorine plays a major role in ozone depletion, it is worth investigating its distribution in the Antarctic strato-sphere. For the analysis only measurements polewards of $40°$ equivalent latitude are used. As a vertical coordinate $\Theta$ was chosen. All measurements have been binned into 5 K potential temperature bins between 270 and 420 K (see Fig. 8). Bins which contain less than five data points are not included in the analysis. The uncertainties represented by the error bars are the $1\sigma$ standard deviations of the means. Up to the potential temperature at which air mass classification begins, the $Cl_y$ is estimated based on all measurements. From the potential temperature at which air mass classification begins, $Cl_y$ is estimated

separately in each region. Measurements within the overlap area in the classification (see Fig 2) are counted both as vortex and mid-latitude measurements.

In Figure 8, the inferred $Cl_y$ throughout the troposphere is close to zero and increases in the tropopause region. The tropospheric measurements during SouthTRAC are thus consistent with $Cl_{total}$ derived from ground based AGAGE measurements. The vertical profiles of vortex, vortex boundary, and mid-latitude declared measurements show different gradients. In the mid-latitudes, the inorganic chlorine hardly increases between 330 and 380 K, with values between 171±78 ppt and 260±117 ppt, whereas the increase is stronger between 380 and 400 K, reaching a value of 446±124 ppt. The profile of the vortex boundary

region increases in the range from 502±110 ppt up to 1090±377 ppt in the $\Theta$ interval of 360 to 395 K. The variability of the vortex boundary profile increases with height. This is partly due to the air mass classification, since the range of values in the vortex boundary region increases with increasing potential temperature (see Fig 2). Inorganic chlorine within the vortex could be obtained from $\Theta$ between 330 to 385 K. $Cl_y$ inside the vortex increases significantly up to a value of 1687±19 ppt. Thus, in late winter and early spring at the highest measured potential temperatures, about half of the recorded chlorine is found in

inorganic form. Despite this amount of inorganic chlorine in the lower stratosphere, the total polar ozone column was higher than usual in September 2019. As a result of the minor SSW event, chlorine deactivation began earlier in 2019 and the ozone hole was about 10 x $10^6$ km$^2$ in size, thus only 20% of that in mid-September 2018 (Wargan et al., 2020).

### 4.4    Comparison of $Cl_y$ in the Antarctic (SouthTRAC) and Arctic (PGS) polar winter

To compare $Cl_y$ in the Antarctic polar vortex and in the Arctic polar vortex, measurements performed on the HALO aircraft

during the PGS campaign were used. PGS consisted of the three partial missions POLSTRACC (Polar Stratosphere in a Changing Climate), GW-LCYCLE (Investigation of the Life cycle of gravity waves) and SALSA (Seasonality of Air mass transport and origin in the Lowermost Stratosphere) to probe stratospheric air during the Arctic winter in 2015/2016 (Oelhaf et al., 2019). Flights of the PGS campaign were conducted from 17 December 2015 until 18 March 2016 and can be separated into two main phases. For this study, only the flights from the second main phase from 26 February to 18 March are investigated,

since they took place in a comparable period (later winter). The separation into vortex, vortex boundary, and mid-latitude measurements is based on the above mentioned method using $N_2O$ measurements performed by the TRIHOP instrument on board of HALO during PGS (Krause et al., 2018) (see Figure S 6 in the supporting information). In section 4.2, the indirect method for the determination of $Cl_y$, where a direct measurement of all relevant chlorinated substances is not possible, was shown to provide values comparable to those obtained by the semi-direct method. During PGS, CFC-12 was measured

with the ECD channel but the MS channel was in NCI mode and could not measure all chlorinated substances. $Cl_y$ was therefore calculated using the indirect method and CFC-12 measurements from the GhOST-ECD channel during PGS. As for SouthTRAC, the scaled correlations from observations of the cryogenic whole-air sampling on two balloon flights inside the Arctic polar vortex in 2009 and 2011 were used (correlation function for the Arctic winter 2015/2016 can be taken from the supporting information).

Figure 9 displays the mean vertical profiles of $Cl_y$ inside the vortex and $Cl_{total}$ of the respective hemisphere. Measurements from the individual campaigns have been binned into 5 K potential temperature (a) and potential temperature difference to the

local tropopause (b). The PV-based dynamical tropopause was used for PGS, as was done for the SouthTRAC analysis. The mean PV-based tropopause was at 306 K during PGS, only slightly lower than that during SouthTRAC at 308 K. Independent of the vertical coordinate, total chlorine ($Cl_{total}$) in the lower stratosphere decreased from the time of PGS (2015/2016) to the time of SouthTRAC (2019). The difference between $Cl_{total}$ from controlled substances during PGS and that during South-TRAC is about 60±9.6 ppt. This difference can be explained by a combination of temporal trends of controlled substances and interhemispheric gradients. Using the rate of decline from Engel and Rigby (2018) of -12.7±0.9 ppt year$^{-1}$ for the controlled substances, a difference of about 45 ppt is expected due to the time difference between the two campaigns. The remaining difference of about 15 ppt can be explained by the higher $Cl_{total}$ in the Northern Hemisphere.

Using $\Theta$ as the vertical coordinate (Fig. 9a), vertical profiles of vortex classified $Cl_y$ of PGS and SouthTRAC show different results. Although the $Cl_y$ vortex profiles are similar until around 350 K, the SouthTRAC profile increased more steeply, reaching values more than 444 ppt larger than those during PGS at the same potential temperatures. Differences are slightly larger when using $\Delta\Theta$ as the vertical coordinate (Fig 9b). Although the two $Cl_y$ profiles lie close together between 20 and 50 K $\Delta\Theta$, the differences between them increase to 540 ppt at 65 K $\Delta\Theta$. The fraction of total chlorine in the form of $Cl_y$ during PGS at the same distance from the local tropopause as the maximum SouthTRAC $Cl_y$ fraction is about 20% in the mid-latitudes (not shown) and about 40% inside the vortex (Fig. 9b).

Figure 10 shows the difference between PGS and SouthTRAC $Cl_y$ in a latitude-altitude cross section. As a horizontal coordinate, equivalent latitude* was used, i. e. the geographic latitude for all tropospheric observations and equivalent latitude for all stratospheric ones (Keber et al., 2020). As a vertical coordinate, $\Delta\Theta$ was used. Since the tropopause height of the two hemispheres is different and changes with the season, the tropopause relative coordinate $\Delta\Theta$ accounts for tropopause variability and allows for better comparison of $Cl_y$. The data have been binned in 5° latitude and 5 K of potential temperature relative to the local tropopause. Only bins which contain at least five data points were considered in this analysis. The difference was calculated by subtracting each Southern Hemispheric latitude-altitude bin from the equivalent Northern Hemispheric latitude-altitude bin. Values in the troposphere differ only slightly. In the lower stratosphere, a separation into two areas can be seen. In the lower stratosphere at higher latitudes, overall higher mixing ratios were derived during SouthTRAC in comparison to PGS. There are two bins with substantially higher values during SouthTRAC with around 900 ppt more $Cl_y$ between 80 and 90 K of $\Delta\Theta$ and 65 to 70° equivalent latitude. This difference is much larger than the difference when comparing vortex classified measurements (Fig. 9). Thus, it is very likely that vortex core and vortex edge values are compared due to the different Arctic and Antarctic vortex size, stability and strength of the transport barrier. Therefore, performing the comparison in equivalent latitude/potential temperature coordinates removes only some of the sources of discrepancy. The stratosphere of the mid-latitudes shows consistently higher $Cl_y$ values during PGS. The highest values of $Cl_y$ reached are 315 ppt higher during PGS between 65 and 70 K of $\Delta\Theta$ and 40 to 45° equivalent latitude. The already mentioned difference of $Cl_{total}$ between PGS and SouthTRAC amounts to 60±9.6 ppt. This maximum difference of 60±9.6 ppt will only propagate completely to $Cl_y$ when all chlorine is converted to the inorganic form. Taking into account the difference in $Cl_{total}$ between the two campaigns would reduce the observed differences in the mid-latitudes, where mixing ratios of $Cl_y$ were higher during PGS, but would increase the differences in the high latitudes, where higher $Cl_y$ was derived for SouthTRAC. The observed differences in $Cl_y$

are thus clearly larger than can be explained by the temporal difference in $Cl_{total}$. Possible reasons for the observed differences can be derived from the hemispheric difference of the Brewer-Dobson circulation, using the age of air as a common metric for transport. Konopka et al. (2015) showed that the age of air is always younger north of 60 °N than south of 60 °S in the corresponding season. Older air will be higher in $Cl_y$ as a larger fraction of total chlorine has already been converted to the inorganic form. Therefore, the observed differences in $Cl_y$, with higher $Cl_y$ values in the southern high latitudes than in the northern high latitudes (see Fig. 9 and 10), are consistent with the differences in the age of air found by Konopka et al. (2015). In addition, Engel and Rigby (2018) updated the long-term total column HCl and $ClONO_2$ (representing $Cl_y$) record reported by Mahieu et al. (2014) in the stratosphere, at Jungfraujoch (46.5 °N) and at Lauder(45 °S), through the end of 2016. A negative trend of $Cl_y$ is observed at both stations but with a non-significant trend for the Jungfraujoch data over the last decade and a slightly larger negative trend from the Lauder data. In addition, trends between 1997 and 2016 are given at both stations, with -0.42 ± 0.23% year$^{-1}$ for HCl and -0.60 ± 0.39% year$^{-1}$ for $ClONO_2$ at Jungfraujoch station and -0.51 ± 0.12% year$^{-1}$ and -0.74 ± 0.59% year$^{-1}$ at Lauder station, respectively. Furthermore, the Global Ozone Chemistry And Related trace gas Data records for the Stratosphere (GOZCARDS) shows short-term lower-stratospheric HCl trends, which are negative at southern latitudes and slightly positive (marginal significance) at northern mid-latitudes between 2005 and 2015/2018 (Froidevaux et al., 2015, 2019). Although the trends given are indicative of the interhemispheric differences in $Cl_y$ decline, they do not explain the difference in $Cl_y$ at mid-latitudes of 200 ppt or more shown in Figure 10. The lowest 20 K above the local tropopause show in general minor differences between the two hemispheres. There are two exceptions. One bin between 0 and 5 K of $\Delta\Theta$ and 30 to 35 ° equivalent latitude and one between 10 and 15 K of $\Delta\Theta$ and 45 to 50 ° equivalent latitude, both with around 200 ppt more $Cl_y$ during PGS. However, this $\Theta$ range is in general affected by cross tropopause mixing in both hemispheres, leading to generally small differences in the extratropical tropopause layer.

## 5   Summary and Conclusion

This study is based on high-resolution measurements of chlorinated hydrocarbons and $N_2O$ taken during the SouthTRAC campaign in the Antarctic lower stratosphere in late austral winter 2019. Extending the method of Greenblatt et al. (2002), it was possible to allocate the measurements to the vortex, vortex boundary region, and mid-latitudes. The classification of air masses based on high-resolution in situ measurements of $N_2O$ offers the possibility to detect and account for even small structures and follows well the sharp gradient between the regimes. However, the weakness of this air mass classification appears near the tropopause, where it was difficult to make a distinction between vortex, vortex boundary, and mid-latitudes.

Inorganic chlorine was calculated semi-directly from the GhOST-MS measurements of the major organic source gases and mean age as well as indirectly using a correlation adapted from observations of balloon flights in the Arctic polar vortex in 2009 and 2011. In order to compare the indirect method to the semi-direct method, first the measurements of the GhOST-MS were up-sampled to a higher time resolution. The simultaneous accurate measurements of CFC-12 in the GhOST-MS and GhOST-ECD channels were used. The indirect method shows good agreement with the semi-direct method despite a time interval

of 10 years and the use of measurements of the Northern Hemisphere. Thus the indirect method serves as a good alternative calculation of inorganic chlorine, in case not all organic source gases are measured.

2019 was a special year for the Antarctic polar vortex with extraordinary meteorological conditions, which led to a minor sudden stratospheric warming. The Antarctic polar vortex was weakened and shifted towards the eastern South Pacific and South America during SouthTRAC (e.g., Wargan et al., 2020; Safieddine et al., 2020). Despite a weakened vortex, up to 50% of the total chlorine could be found in inorganic form inside the vortex at highest $\Delta\Theta$ levels of 75 K above the tropopause. Furthermore, inorganic chlorine for mid-latitudes and vortex boundary region could be derived during SouthTRAC, with only about 15% of the total chlorine in inorganic form in the mid-latitudes.

Measurements from the PGS campaign, which took place in the Arctic polar winter 2015/2016, were used to compare Arctic and Antarctic $Cl_y$. To our knowledge, a comparison of $Cl_y$ of the Arctic and Antarctic vortex has not been published previously. For PGS, $Cl_y$ was calculated using the indirect method based on scaled correlation from the observations of balloon flights in the Arctic polar vortex in 2009 and 2011. Additionally, region classification was done using $N_2O$ measurements, as for the Southern Hemisphere data. In contrast to the Antarctic polar vortex in 2019, the Arctic polar vortex in 2015/2016 was one of the strongest compared to previous years (Matthias et al., 2016). At a comparable level of $\Delta\Theta$ inside the vortex, only around 40% of total chlorine was found in inorganic form, whereas roughly 20% was found at mid-latitudes. Inside the respective vortex, the amount of $Cl_y$ was higher during SouthTRAC than during PGS by up to 540 ppt (at the same $\Delta\Theta$ level). Trends due to the Montreal Protocol are estimated to be negative at about -20 ppt year[-1], which is not evident in this comparison. The differences in $Cl_y$ values inside the two vortices are substantial and even larger than the inter-annual variations reported by Strahan et al. (2014) for the Antarctic. For the comparison of the Arctic and Antarctic $Cl_y$ in this study, only one winter in each hemisphere was investigated. Furthermore, the respective campaigns only show a part of the winter seasons. These intervals do not correspond completely. For a more meaningful conclusion about the $Cl_y$ loading in the two polar vortices, further sampling at different seasons and over several years would be required.

Investigating the difference of $Cl_y$ in a latitude-altitude cross section from PGS and SouthTRAC, higher values at higher latitudes were found for SouthTRAC, whereas higher values for the mid-latitude lower stratosphere were found for PGS. In the troposphere and near the tropopause, differences become smaller. $Cl_y$ increases with increasing mean age of air, for which Konopka et al. (2015) derived similar hemispheric differences based on model simulations using meteorological reanalysis data. A comparison of the available data with chemical transport models should be the subject of further studies. Furthermore, such interhemispheric differences should also be captured by chemistry climate models, which are used not only to understand past changes but also predict future changes in chemical composition. The higher values from SouthTRAC at higher latitudes may reflect the difference in spatial extent, isolation, and location of the Southern Hemisphere vortex. The Arctic vortex exhibits larger variability as it is more affected by weather systems or wave activity. The Antarctic vortex on the other hand is larger and stronger and is typically less affected by wave disturbances.

*Data availability.* The observational data of the HALO flights during the SouthTRAC campaign are available via the HALO database https://halo-db.pa.op.dlr.de (last access: 7 August 2021).

## Appendix A: Filter procedure for vortex and mid-latitude profiles

A filter procedure was used to derive the lower envelope for the vortex profile and the upper envelope for the mid-latitude profile. Figure S1 in the supporting information displays the procedure for the task using either $\Delta\Theta$ or $\Theta$ as the vertical coordinate. The process is initialized by binning the $N_2O$ measurements into intervals of e.g., $\Delta\Theta$. The bin size must be adjusted to the number of measurements available for the vortex and mid-latitude profile to make the filter procedure work properly. For every bin, the mean value, standard deviation, and relative standard deviation are calculated. This is necessary as the condition for the filter needs a binned profile to begin with. While the maximum relative standard deviation is larger than the preset outlier limit, the measurements that are not flagged as outliers are binned in intervals of $\Delta\Theta$ (this is done twice in the first iteration step, since the binned profile is already needed for the initialization and no outliers are set for the beginning of the filtering process). Every bin is checked, whether the relative standard deviation is larger than the outlier limit. In this case, all measurements of $N_2O$ which are higher (or lower, if the upper envelope is requested) than the mean of the respective bin are flagged as outliers and removed from further iterations. The iteration process stops when the maximum relative standard deviation is below the preset outlier limit.

For the vortex profile, bin size was set to 2 Kelvin. The variability of $N_2O$ on a constant $\Theta$ surface inside the vortex is about 6 ppb (Greenblatt et al., 2002). For the range of $N_2O$ mixing ratios in this work, 6 ppb corresponds roughly to about 3 % and was thus set as the outlier limit. Four iterations were done to get the lower envelope (grey samples in Figure S 2 a) and b) in the supporting information). For the mid-latitude profile, the bin size was set to 2 Kelvin. Strahan et al. (1999) showed that the variability of $N_2O$ in the Southern Hemisphere lower stratosphere of the mid-latitudes is approximately between 5 and 15 % (see plate 6 therein). Therefore, a value of about 10 % is set for the outlier limit, which leads to two iteration steps for the remaining measurements. For the profiles only those measurements are used which are not marked as outliers.

*Author contributions.* MJ and AE performed the study and MJ wrote the paper. Measurements were performed by MJ, TK, TS, TW, and AE (GhOST), and HB, H-CL and PH (UMAQS). All authors contributed to the final paper.

*Competing interests.* The authors declare that they have no conflict of interest.

*Acknowledgements.* This work was done at the University of Frankfurt. Funding from BMBF Programme ROMIC-II under Grant nr. 01LG1908B (SCI-HI) and from the German science foundation (DFG) under grant numbers EN367/13-1, EN367/14-1, EN367/16-1 EN367/17-1, HO4225/15-1 and HO4225/14-1 for the PGS and SouthTRAC campaigns. The authors would like to thank the DLR staff for the operation

of the HALO and the support during the campaign as well as the coordinators and colleagues for a productive cooperation during the cam-

paign. We further thank Jens-Uwe Grooß from the Forschungszentrum Jülich for the calculation of the tropopause and equivalent latitude for

the HALO campaigns. The authors would further like to thank the site operators at the Cape Grim, Mace Head, Trinidad Head, Ragged Point,

and Cape Matatula stations. AGAGE is supported principally by NASA (USA) grants to MIT and SIO, and also by: BEIS (UK) and NOAA

(USA) grants to Bristol University; CSIRO and BoM (Australia): FOEN grants to Empa (Switzerland); NILU (Norway); SNU (Korea); CMA

(China); NIES (Japan); and Urbino University (Italy). The authors would like to thank the editor, Marc von Hobe, and the two reviewers,

Michelle Santee and another anonymous one for their helpful comments.

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

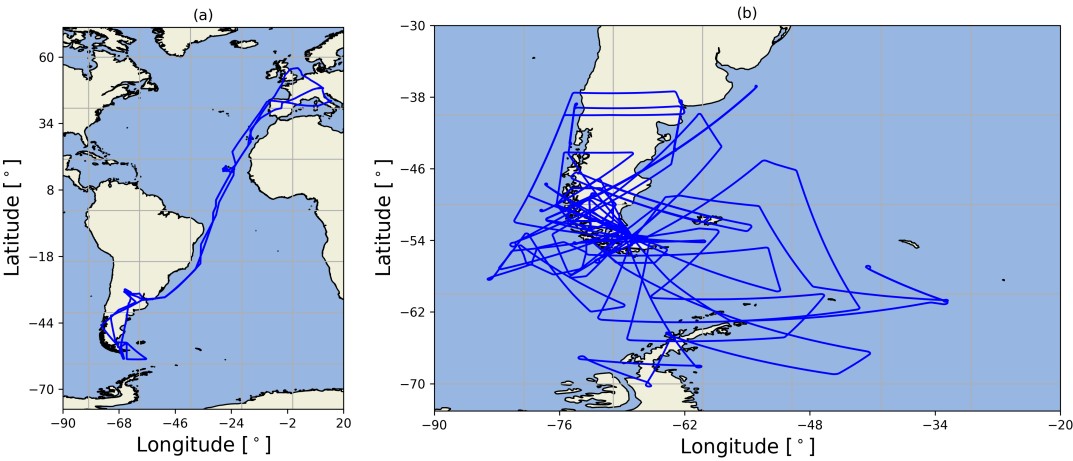

**Figure 1.** Flight tracks of HALO of (a) the transfers from and to Oberpfaffenhofen, Germany (48°N, 11°E) and (b) during the two phases with the base in Rio Grande, Argentina (53°S, 67°W).

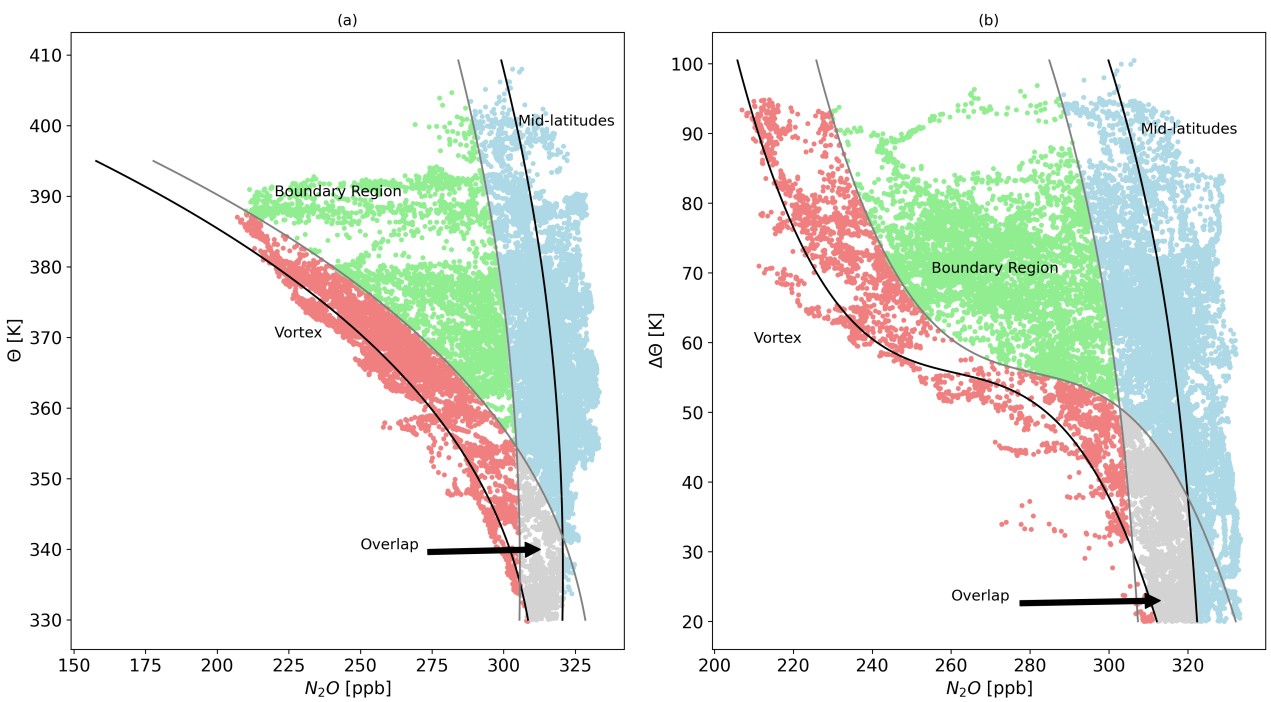

**Figure 2.** Mid-latitude and vortex profiles (black) of $N_2O$ versus (a) potential temperature ($\Theta$) and (b) potential temperature difference ($\Delta\Theta$) from the local PV tropopause. The vortex cutoff criterion (see main text for details) of 20 ppb is illustrated by the grey profile on the right side of the vortex profile. Mid-latitude variability of 15 ppb is illustrated with the grey profile on the left side of the mid-latitude profile. In between, there is the vortex boundary region. Overlap region is declared for the area where the vortex cutoff and mid- latitude variability cross. Additionally, $N_2O$ measurements classified to the respective region are displayed. Vortex measurements in red, vortex boundary region measurements in green, mid-latitude measurements in blue, and overlapping measurements in grey.

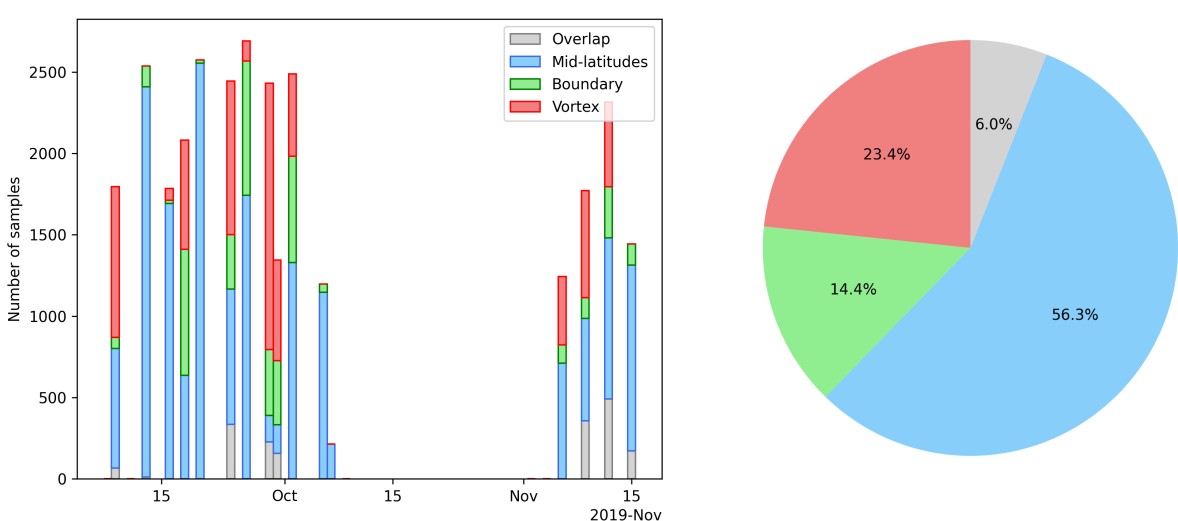

**Figure 3.** Air sampling statistics of the SouthTRAC campaign. On the left, the number of classified measurements. Each column represents a single scientific flight. Stacked bars indicate vortex (red), vortex boundary (green), mid-latitudes (blue), and undefined (grey) amounts. On the right side, percentage of each region to the total of all measurements in the scope of the classification (above $\Delta\Theta$ of 20K).

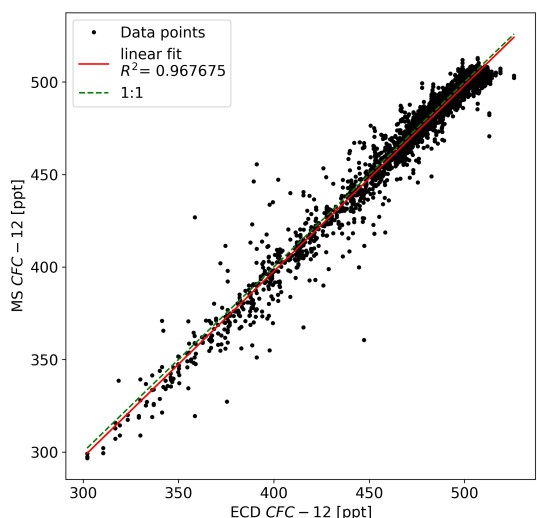

**Figure 4.** Correlation between CFC-12 measured in the GhOST-MS channel and in the GhOST-ECD channel.

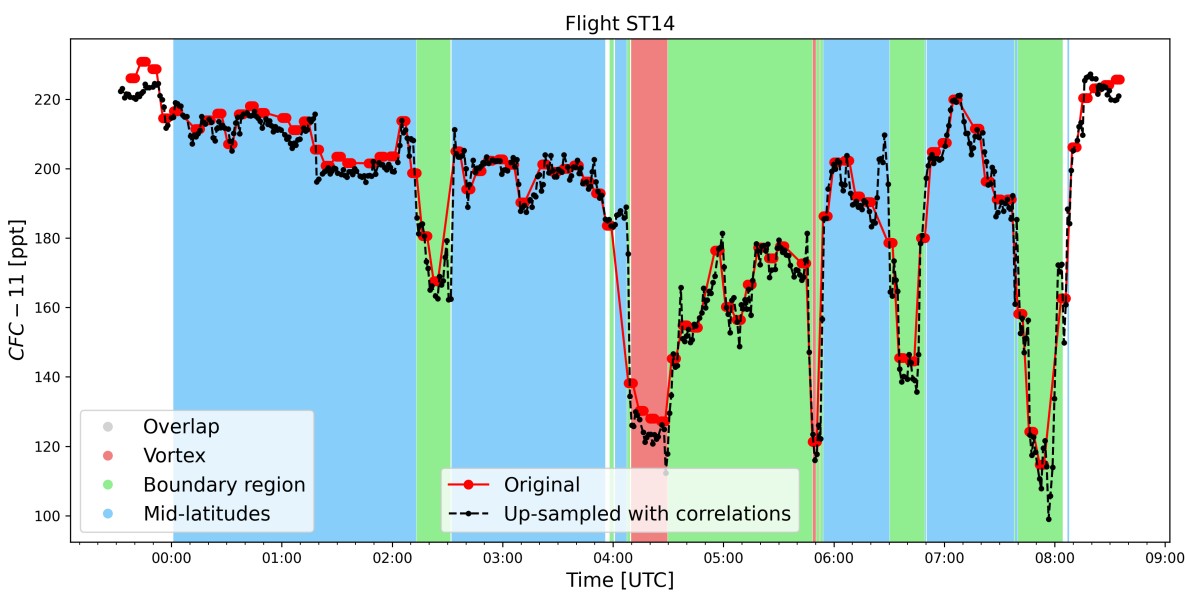

**Figure 5.** CFC-11, measured in the GhOST-MS during flight ST14 on 26 September 2019. Original data shown in red, measurements up-sampled using CFC-12 from the ECD channel in black. Background colors indicate in which region the samples were taken, using the air mass classification in Θ-coordinates.

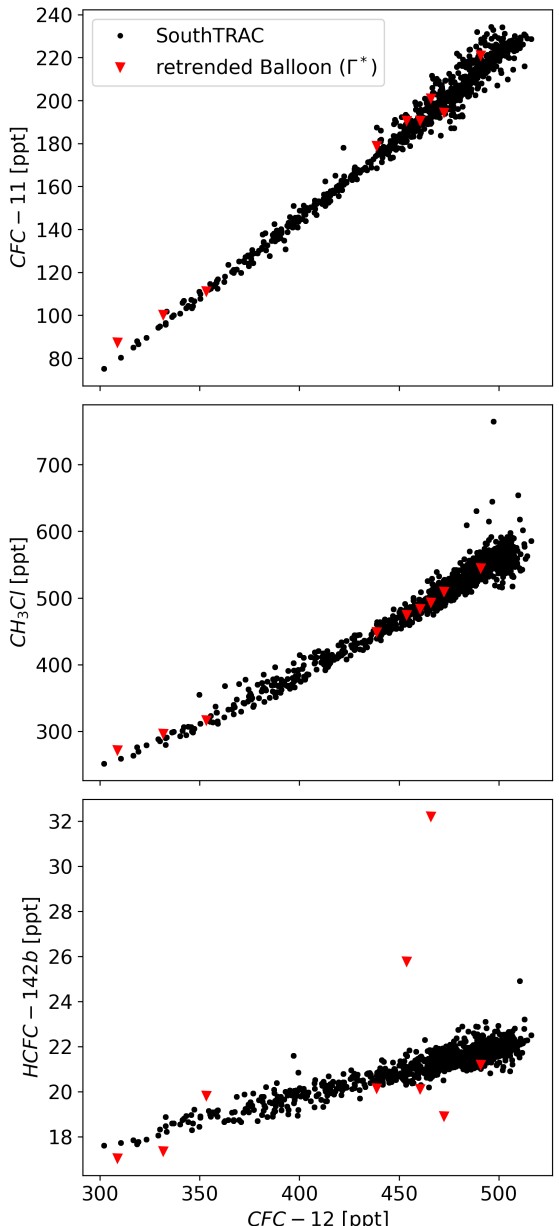

**Figure 6.** Correlation between CFC-12 and CFC-11, CFC-12 and CH₃Cl, and between CFC-12 and HCFC-142b. In black the raw measurements by GhOST-MS, in red the balloon observations scaled to the time of the SouthTRAC campaign using mean arrival time.

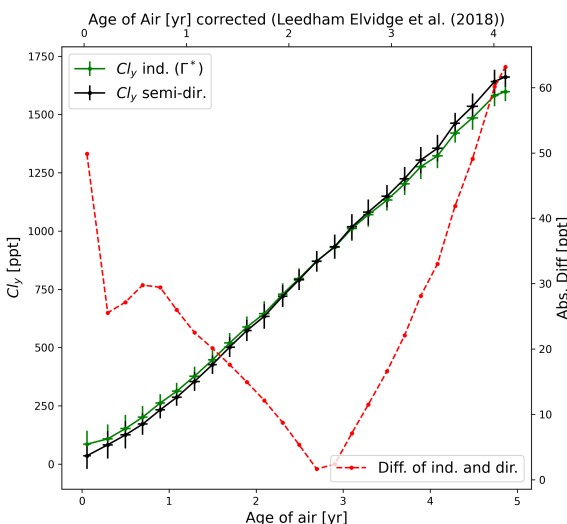

**Figure 7.** Indirectly (green) determined Cl$_y$ based on balloon observations in 2009 and 2011 and semi-directly (black) determined Cl$_y$ as a function of age of air (bottom axis) and corrected age of air (top) using a linear fit y = 0.85x - 0.02 from Leedham Elvidge et al. (2018). In red, absolute difference between these methods.

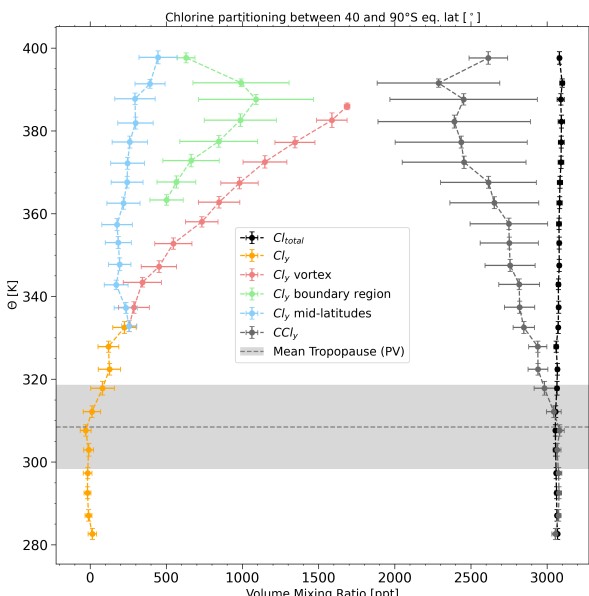

**Figure 8.** Vertical profiles of $Cl_y$, $Cl_y$ by region from 20 K above the local tropopause, $CCl_y$, and $Cl_{total}$ averaged over $40° - 90°$ S equivalent latitude for all flights during SouthTRAC. The data are displayed as a function of potential temperature. Vertical and horizontal error bars denote $1\sigma$ variability. The dashed line shows the averaged PV-based dynamical tropopause with the $1\sigma$ variability as shaded area.

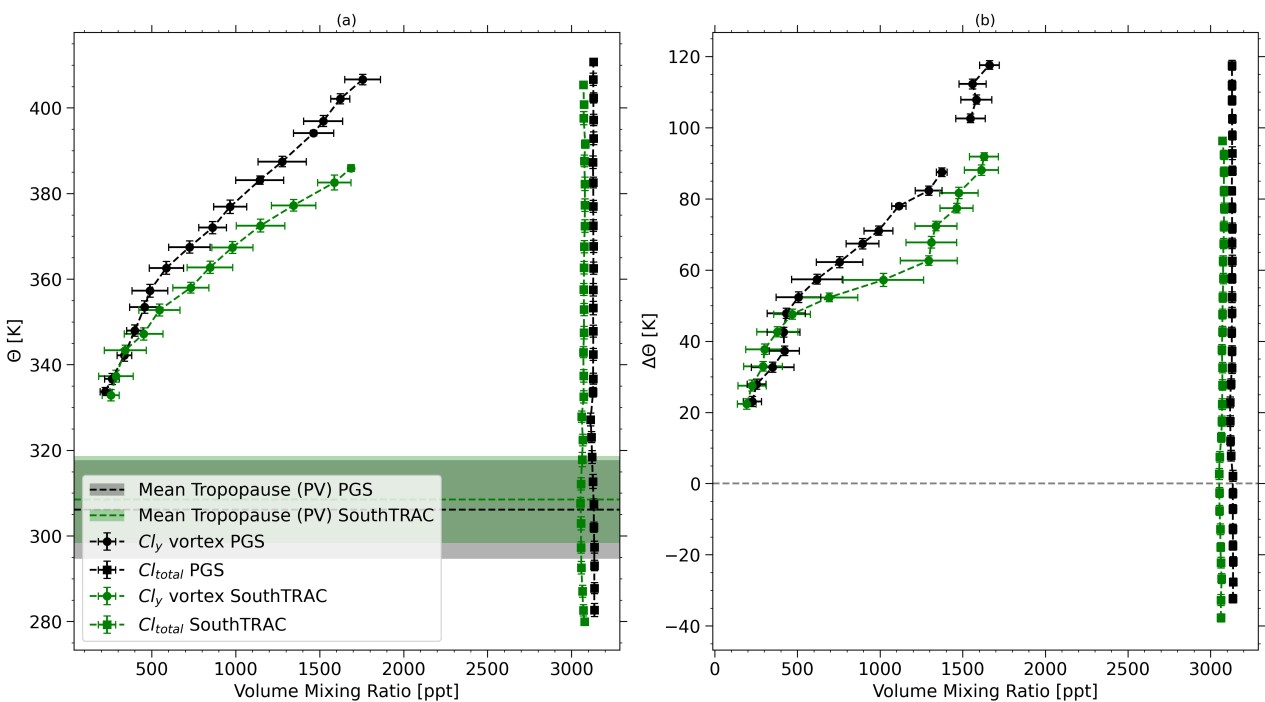

**Figure 9.** (a) Comparison of the vertical profiles of $Cl_y$ inside the respective vortex where classification was possible as well as total chlorine from PGS (black) and SouthTRAC (green). Data are averaged over $40°$ to $90°$ equivalent latitude of the respective hemisphere and are displayed as a function of potential temperature. Vertical and horizontal error bars denote $1\sigma$ variability. PV tropopause for PGS (black) and SouthTRAC (green) are displayed as dashed horizontal lines with the $1\sigma$ variability as shaded areas. (b) as in (a) but as a function of potential temperature relative to the local tropopause (PV), displayed with a grey dashed line.

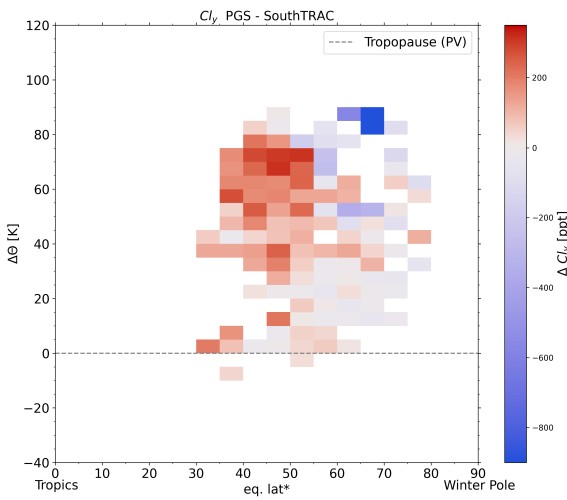

**Figure 10.** Difference of the latitude-altitude cross section of Cl$_y$ from PGS and SouthTRAC. Data are binned using equivalent latitude$^*$ and $\Delta\Theta$.

**Table 1.** Chlorinated species measured with the GhOST- MS. Precisions and limit of detection (LOD) of GhOST have been determined in the laboratory shortly before the SouthTRAC (ST) campaign, and mean precisions were calculated during the flights.

| Name | Formula | Laboratory | | ST |
|------|---------|------------|------|------|
| | | Prec. (%) | LOD (ppt) | Prec. (%) |
| CFC-11 | $CCl_3F$ | 0.22 | 0.36 | 1.13 |
| CFC-12 | $CCl_2F_2$ | 0.30 | 0.47 | 0.71 |
| CFC-113 | $C_2Cl_3F_3$ | 0.64 | 0.18 | 2.93 |
| Methyl chloride | $CH_3Cl$ | 0.39 | 0.76 | 1.11 |
| Tetrachloromethane | $CCl_4$ | 0.44 | 0.22 | 0.98 |
| Methyl chloroform | $CH_3CCl_3$ | 8.67 | 0.53 | 3.76 |
| HCFC-22 | $CHClF_2$ | 0.41 | 1.31 | 0.84 |
| HCFC-141b | $C_2H_3Cl_2F$ | 0.82 | 0.39 | 1.23 |
| HCFC-142b | $C_2H_3ClF$ | 0.84 | 0.50 | 1.63 |

**Table 2.** Coefficients of the correlation function to indirectly derive $Cl_y$ with the respective reference substance for the time of the SouthTRAC campaign (2019.75). Calculation of $Cl_y$ with CFC-12 or $N_2O$ and coefficients based on the balloon observations in 2009 and 2011 (Balloon) as well as coefficients based on the SouthTRAC measurements (SouthTRAC).

| Data source | $\chi_{ref}$ | $c_0$ [ppt] | $c_1$ | $c_2$ [ppt$^{-1}$] |
|-------------|--------------|-------------|-------|---------------------|
| Balloon | CFC-12 | 2965.27 | -2.80700 | $-6.06944*10^{-3}$ |
| Balloon | $N_2O$ | 2990.74 | -2.16187 | $-2.10586*10^{-2}$ |
| SouthTRAC (limited to about 300 ppt) | CFC-12 | 3024.26 | -2.61888 | $-6.87717*10^{-3}$ |
| SouthTRAC (limited to about 250 ppb) | $N_2O$ | 3884.40 | -8.02682 | $-1.14510*10^{-2}$ |