# Peer review of "Comparison of Inorganic Chlorine in the Antarctic and Arctic lowermost stratosphere by separate Late Winter aircraft measurements"

_Atmospheric Chemistry and Physics, 2021_

## Referee Comment (RC2)

**Review of "Comparison of Inorganic Chlorine in the Southern Hemispheric lowermost stratosphere during Late Winter 2019" by Jesswein et al.**

This paper analyzes in situ measurements of halocarbons and $N_2O$ obtained during the 2019 SouthTRAC and 2015/2016 PGS HALO aircraft campaigns. Air masses are classified as belonging to vortex, edge, and mid-latitude regions. Estimated $Cl_y$ abundances in the two hemispheres are compared. Although the measurements and the derived $Cl_y$ profiles will be of interest to the readership of ACP, in my opinion the manuscript suffers from a number of flaws in the analysis and/or the description thereof. Consequently, I have a rather large number of specific comments that I would like to see addressed before the paper can be accepted for publication. In some cases, my concerns can be resolved by simply correcting and clarifying the discussion, but others will require additional analysis or other substantial changes.

Below both major substantive issues and minor points of clarification, wording suggestions, and grammar / typo corrections are listed together for each Section (including Supporting Information) in sequential order through the manuscript.

Respectfully,
Michelle Santee

Title
- The comparison being made in this paper is not clear from the title. It would be better to craft a different title capturing the idea that in situ measurements from separate aircraft campaigns are being used to compare $Cl_y$ abundances in the Antarctic and the Arctic LMS.

Abstract
- L1: The wording "… long-lived chlorinated source gases. These include the reservoir species" seems to imply that HCl and $ClONO_2$ are chlorinated source gases. Thus, These include --> $Cl_y$ includes
- L5: in late winter --> in austral late winter
- L8: The sentence "$Cl_y$ from a scaled correlation was compared to directly determined $Cl_y$ …" is confusing, since the previous sentence states that not all source gases were measured during PGS. It needs to be made clear that this "validation" was performed for SouthTRAC.
- L12-13: The values (40%, 20%) here appear only in the abstract and conclusions, not in the main text. In my opinion, it is not appropriate to include "new information" in the abstract and summary sections that has not been thoroughly discussed in the body of the paper. Please add some corresponding statements in section 4.
- L13-14: Differences inside the respective vortex reaches up to 565 ppt more --> Differences inside the two vortices reach as much as 565 ppt, with more
- L15-16: As far as is known --> To our knowledge; within the respective polar vortex --> within the Antarctic and Arctic polar vortices
- L16-17: "the difference of $Cl_y$ inside the respective vortex is significant and larger than reported inter annual variations". I have a number of comments on this sentence:
  - The authors have not done a statistical analysis, so I do not think that "significant" is an appropriate word here.

- o This statement could be erroneously interpreted as implying that their study examines interannual variations. Moreover, the Strahan et al. [2014] paper on which this statement is based looked only at the Antarctic, not the Arctic.
- o The word "respectively" is used many times throughout the manuscript, sometimes (as here) incorrectly. I have recommended alternative wording in a few places.
- o Thus, I suggest instead: "the differences in $Cl_y$ inside the two vortices are substantial and larger than the interannual variations previously reported for the Antarctic".

**Section 1**
- L20: 1980-ies --> 1980s; also, pre-dominantly --> predominantly
- L23: substances, which are involved --> substances involved
- L23-24: OClO is a consequence of, and thus a good qualitative indicator of, halogen activation, but it does not itself participate in ozone destruction, as this sentence implies. Thus it is not normally considered part of $ClO_x$.
- L25: within --> through
- L26: While I applaud the recognition of some of the original papers, I think it would be good to also include some review articles (e.g., Solomon, Rev Geophys, 1999) and/or some more up-to-date citations (e.g., the most recent WMO Ozone Assessment).
- L30: Citing only the Newman et al. [2004] paper here gives the erroneous impression that it is the only relevant reference. At the very least, "e.g.," needs to be added to this citation. This is another instance where it might be appropriate to cite the WMO Report.
- L35: used again --> again used
- L39-40: Citations for the long-term trends in $Cl_y$ and $N_2O$ should be given.
- L45: between polar --> between the polar
- L46: Here and throughout the manuscript, when "mid-latitudes" is used as an adjective to modify a noun (e.g., stratosphere, profile, reference, etc.), it should be singular: "mid-latitude". When it is used as a noun itself (as in "at mid-latitudes"), then it is plural.
- L46: add "e.g.," in front of "Schoeberl"
- L48: add "e.g.," in front of "Hartmann"
- L62: SouhTRAC --> SouthTRAC
- L65: delete duplicate period after "4"

**Section 2**
- L70: capable to reach --> capable of reaching
- L72: Rio Grande (RGA), Argentina (53°S, 67°W) --> Rio Grande, Argentina (RGA, 53°S, 67°W); also, regions for --> regions of
- L74-75: The actual dates (not just "September" and "November") for the two phases of the campaign should be given.
- L77-80: 9 transfer flights + 10 Phase I flights + 3 Phase II flights = 22 total flights, not 23 as stated in L77
- L78: Rio Grande (RGA), Argentina --> RGA
- L84: Beside --> Besides (or, "in addition to"); also, delete comma after "instruments"
- L108-109: Given the in-flight conditions as described in this paragraph, it makes sense that the mean precisions during the campaign are poorer than those measured in the lab. So it is

puzzling that the mean precisions during the flights improved over those measured beforehand for methyl chloroform. Do the authors have an explanation?

- L108-112: It is stated that for CFC-113 "the amount of water in the analytical system should be kept as low as possible", but it is not clear whether that was actually done during the campaign. The implication is that in fact this was not done properly and that is why the in-flight precision of CFC-113 is so much worse than that determined in the lab, but this point needs to be clarified. I also think it is questionable whether the measurement of CFC-113 really stands out as an "exception" (L108) for its degraded performance. In fact, the precision estimated during the flights is even worse relative to the pre-campaign value (a factor of 5 difference) for CFC-11.
- L109: precision CFC-113 --> precision of CFC-113
- L109: It seems odd to me that the authors make the effort (L85) to define "payload", which is a widely known and not particularly technical term, but not "elutes", which many of their readers (including me) may not know. Also, should it be "CFC-113 is eluted by water" rather than "CFC-113 elutes near water"?
- L110: add a comma between "water" and "the amount of water"
- L111-113: Precision values are not given for GhOST-ECD $SF_6$ either in the text or in Table 1, yet $SF_6$ measurements are mentioned later in the paper.
- Table 1 caption: have been determined shortly before the SouthTRAC (ST) campaign and mean precisions during the flights --> have been determined in the laboratory shortly before the SouthTRAC (ST) campaign, and mean precisions were calculated during the flights
- L120: prior --> prior to
- L122: was of --> was
- L123: post-flight corrected --> corrected post-flight

Section 3
- L125: The occurrence of chlorine activation also depends on factors other than temperature (e.g., humidity, the availability of suitable aerosol particles) and has been observed outside the polar regions, so it would be better to say "tends to occur" rather than "occurs" here.
- L129: conclusion --> conclusions
- L134-135: Modern meteorological reanalyses have fairly high resolution these days [e.g., Fujiwara et al., ACP, 2017]; although they still do not resolve very small-scale features, it is not entirely fair to characterize them as having "rather coarse resolution".
- L138-145: This part of the paragraph is poorly written, repetitive, and hard to follow. It should be reorganized to improve the flow.
  - The sentence "It can be measured … atmosphere." is out of place, as it comes in between two sentences that say essentially the same thing. It should be moved and the other two sentences combined to reduce repetition.
  - Not only is the statement "the isolation inside the vortex benefits mixing on isentropic surfaces and therefore a small variability on isentropes (variability of about 6 ppb)" grammatically incorrect, but also it makes no sense. I'm not sure what "benefits mixing" means? Perhaps "inhibits" was meant? In any case, this sentence needs to be rewritten.
  - The sentence "The low mixing ratios inside the vortex … $N_2O$" again repeats the same information already stated twice above. (Also, descend --> descent)

- o the mid-latitudes vertical gradient is weak and more variable --> the vertical gradient in mid-latitude $N_2O$ is weak and more variable
- L146-147: "Towards tropopause altitudes, the $N_2O$ profiles of vortex and mid-latitudes merge and differentiation becomes difficult." Near the tropopause the vortex proper – and the transport barrier it represents – is no longer defined; the region in which chemical processing still takes place but confinement is weak (below ~350–380 K in SH, ~400–450 K in the NH, depending on the year) is often termed the "subvortex" [see Santee et al., JGR 2011, and numerous references therein]. So it is not appropriate to refer to "vortex profiles" in this region.
- L149: at best --> ideally
- L152-153: "Stratospheric transport and mixing is related to the isentropic surfaces whereas mixing at the extratropical tropopause affects the lowest 25 K relative to the local tropopause." This sentence needs work.
  - o The wording "is related to" is not clear. I assume that the first half of this sentence is referring to the fact that adiabatic flow in the stratosphere largely occurs along isentropic surfaces, but this should be clarified.
  - o is --> are; also, add a comma after "surfaces"
  - o References are needed, especially for the point that mixing affects the lowest 25 K above the tropopause (see below).
- flights, which --> flights that; contact to --> contact with
- L155-157: "Data from these flights were pre-filtered by taking only the measurements polewards of 60°S equivalent latitude and 20 K above the local tropopause." There are several issues with this sentence.
  - o The concept of equivalent latitude should be defined and a suitable reference for it provided (e.g., Butchart & Remsberg [JAS, 1986]).
  - o Presumably the EqL is being calculated based on PV from a meteorological reanalysis, but this information needs to be provided. The reanalysis being used in a study is typically identified in the "Data and Methods" section – here that section is entitled "The SouthTRAC Campaign", but the meteorological data is also an important component of this study and probably merits its own subsection. I have more to say on this point later.
  - o Similar to the above point, it needs to be made clear how the local tropopause is being determined. The Fig. 2 and 8 captions mention the "WMO tropopause", but more detailed information should be provided in the main text. In addition, it seems that the results may be highly sensitive to the exact definition of the tropopause used, and some discussion of the associated uncertainty in the results would be appropriate.
  - o The Antarctic vortex frequently extends to EqLs lower than 60°S EqL. Have the authors made sure that imposing the 60°S EqL cutoff has not eliminated vortex profiles in 2019?
  - o The previous paragraph states that mixing affects the region within 25 K of the tropopause, so it is not clear why the cutoff here was chosen to be 20 K.
- L157-158: This sentence ("The lowest levels …") is repetitive, unnecessary, and out of place – it should be combined with the similar sentence in L152-153.
- Fig. 2 caption: to the local --> from the local; criterion on --> criterion of; the the --> the
- L158-164: One general comment is that the creation of reference profiles is a key point on which much of the following analysis rests, and its description should not be relegated to the separate Supporting Information, which many readers will not make the effort to obtain. I

would prefer to see it in the main text, but it should at least be moved to an Appendix included at the end of the main paper file (unless ACP no longer allows such Appendices). Another general comment is that the main text, SI, and figure captions together fail to clearly describe the method, as specified in more detail in the points below.

- o L160: At this point, the reader has no idea what is meant by the term "vortex profile function." Also, Werner (2006) is a PhD thesis for which no download information is given, and thus it is not a suitable reference.
- o L161: Elsewhere the convention "60°S EqL" is used, so for consistency "-40° and -60° EqL" here should be "40° and 60°S EqL".
- o S1, L2-9: The first two paragraphs of the SI are fully redundant with the discussion in the main text. Apart from this repeated material, the description of the procedure in the SI is only one paragraph long, so again I would argue that it would be better to edit, merge, and rearrange the discussions in S1 L10-28 and L158-164 to produce a single compact paragraph in the body of the paper.
- o Fig. S1: I did not find the flowchart to be particularly helpful, so it could remain in the SI.
- o Fig. S2-S7: It would be much easier on the reader if all of the vortex figures (S2, S4, S6) were combined into one 3-panel figure, and the same for the mid-latitude figures (S3, S5, S7). In fact, it would probably work to combine them all into one 2-row, 3-column figure.
- o S1, L13: is calculated --> are calculated
- o S1, L14 and L21: pre-setted --> preset
- o S1, L14: measurements, which --> measurements that
- o S1, L17: Is this the case --> In this case
- o S1, L21: shows --> show
- o S1, L23: latiudes --> latitude
- o S1, L22-23: More discussion is needed on the 3% and 10% "outlier limits" for vortex and mid-latitude profiles, respectively. How were these preset outlier limits and $\Delta\theta$ bin sizes determined? What factors drove the differences between the values of these quantities for the vortex and mid-latitude profiles? How sensitive are the results to these choices?
- o Fig. S2: What causes the "staircase" pattern between the points discarded in iterations 1 and 2? Also, left over --> leftover, but "remaining" would probably be a better word here (also in the Fig. S3 caption)
- L167: decent --> descent
- L168-169: Greenblatt et al. [2002b] quantifies descent inside the Arctic vortex and thus has only limited applicability for SouthTRAC, since the characteristics and seasonal evolution of descent are somewhat different in the two hemispheres, as discussed by Manney et al. [JAS, 1994], which would be a more appropriate reference. Manney et al. note that parcels in the SH lower stratosphere generally cease to descend in mid-October, so descent might still be ongoing in September, contrary to the statement made here. Also, a further --> further.
- L173-174: I am confused about the "prescribed cutoff value" for vortex profiles and "associated variability" for mid-latitude profiles mentioned here and specified (20 ppb and 15 ppb, respectively) in the Fig. 2 caption. Where did these values come from? Why is one characterized as a "prescribed cutoff" and the other as an "associated variability"? Do these values have anything to do with the outlier limits discussed above? In addition, it seems that all points falling "below" (i.e., to the left of) the grey "cutoff" curve are deemed to be vortex points, even when they are "above" (i.e., to the right of) the black $N_2O_{vor}$ curve, but that is

not what is said in "if the mixing ratio is below the respective $N_2O_{vor}$ with a prescribed cutoff value, then it is assigned to the vortex" (and similarly for the mid-latitude case).

- L176-177: It is stated that measurements in the overlap region cannot be "fully assigned to one region". So what was done with them? Were they included in the analysis or discarded? This question is answered later in section 4.3, but the reader should not be left in suspense here. Also, delete the comma after "ratios", and "cannot" is one word, not two.
- L181: has been --> was
- L183: "timed" sounds intentional, whereas I believe that the campaign fortuitously took place shortly after the SSW. This sentence is also grammatically awkward. I suggest instead: "… November; thus they occurred shortly after the minor SSW event and captured the late winter evolution …"
- L187: Other flights besides the 11 September flight are omitted from Fig. 3, since the number shown does not add up to the total given in section 2.
- L189: It is not quite true that "flights sampled mostly inside the vortex or vortex boundary region" during Phase I – in fact, Fig. 3 shows that few or no such measurements were taken on nearly half of those flights. Also, extensive --> extensively; add a comma after "phase"
- L191: more than half of the … air 54% --> more than half (54%) of the … air

Section 4
- L194-197: These sentences about EESC seem out of place here. Perhaps they would fit better at the beginning of subsection 4.2, or in the Introduction. Also, matter --> manner
- L197-199: These two sentences are redundant with the paragraph at the start of section 4.1. It would be better to merge / edit to avoid such repetition from one paragraph to the next.
- L205-206: Measuring CFC-12 on both the ECD and MS channel of the instrument allows to up-sample the measurements of the organic source gases by using the higher resolved --> Measuring CFC-12 in both the ECD and MS channels of the instrument allows the measurements of the organic source gases to be up-sampled by using the better-resolved
- L206: CFC-12 on --> CFC-12 in; throughout the manuscript (including in figure captions), "measurements on" should be changed to "measurements in"
- L207: but also a better precision than on the MS channel --> but they also have better precision than data from the MS channel
- L209: add a comma after "ratios"
- L209: It might be good to add "linear or polynomial" in front of "fit function".
- S2, L30-31: For up-samling the GhOST-MS measurements, pre-required are good correlations between CFC-12 and the other --> Up-sampling the GhOST-MS measurements requires good correlations between CFC-12 and the other
- Fig. S8: It should be made clear in the caption that all of the data shown are from the GhOST-MS channel. Also, the small font makes the axis labels on these panels very hard to read.
- Fig. 5: In red, original data, whereas is black, measurements were up-sampled using CFC-12 measurements of the ECD channel --> Original data shown in red, measurements up-sampled using CFC-12 from the ECD channel in black
- L216-217: It is not that "the original data were not well captured"; rather, the original lower-resolution data did not capture well the abrupt transitions between regimes.

- L224-225: I find this wording unclear.  It would be better to rewrite as: "Organic chlorine (CCl$_y$) … up-sampled GhOST-MS measurements.  Thus, Cl$_y$ can be calculated from Eq. 1 if the mixing ratios of the major chlorine-containing substances at the stratospheric entry point (Cl$_{total}$) are known.  Air enters the stratosphere predominantly …".
- L227: can not --> cannot
- L228: times in the stratosphere since they entered the stratosphere --> times since they entered the stratosphere
- L236: previous --> previously
- L238: ratio, which --> ratio that; degradation and thus --> degradation, which thus
- L239: It would be appropriate to add "estimated" in front of "entry mixing ratios"
- L241: For the case no --> For the case where no; also, to make the distinction between the so-called "semi-direct" and "indirect" methods more clear, it would help to add "indirectly" between "calculated" and "based on".
- L244: add a comma after "trends"
- L246: where --> when
- L247: delete the commas after both "showed" and "tracers"
- L254: respective entry --> respective estimated entry
- L256-257: rations with --> ratios by
- L259: It is difficult for the reader to keep track of exactly what is meant by the "direct", "semi-direct", and "indirect" methods.  To help clarify this sentence, it would be good to add "based on previous balloon observations transferred to 2019" after "indirectly determined correlations" (assuming that I am interpreting the approaches correctly).
- L259: indirectly determined values are not only based on observations which have been performed about 10 years earlier but also are from the --> indirectly determined values are based on observations that were not only performed about 10 years earlier but that were also from the
- L261: This wording is unclear.  Replace "They" with "The balloon-based correlations".
- Fig. 6: I assume that the SouthTRAC (black) points in this figure show the up-sampled (not raw) GhOST-MS measurements for CFC-11, etc., but this should be stated explicitly.  Also, I may not be interpreting this figure correctly.  Do the red symbols represent the correlations between balloon CFC-11 (for example) and balloon CFC-12 data, or between balloon CFC-11 (for example) and GhOST-ECD CFC-12 data?  Please clarify.  In addition, the term "retrended" is used only in the figure caption and legend, not in the main text.  Although it is somewhat ambiguous, it is fine to use this term as long as it is defined in the body of the paper.
- L262-265: I am not following the logic here.  I understand that a subset of the components of CCl$_y$ "retrended" from earlier balloon data match well correlations with CFC-12 measured by SouthTRAC (Fig. 6).  But why does that necessarily mean that CCl$_y$ based on correlations with CFC-12 can be used as a good proxy for Cl$_y$?  I feel that a step is missing.  And why is it relevant to mention here again that Cl$_{total}$ can be derived to calculate Cl$_y$ – that information is not being used for Eq. 2 and the indirect method, is it?  In general, I feel that the relationship between Eq. 1 (at the heart of the semi-direct approach) and Eq. 2 (the basis of the indirect method) is not clearly explained.  Please clarify this discussion.
- L266-267: Since the previous sentences have been discussing balloon-based correlation functions, this sentence about the GhOST-ECD CFC-12 data seems out of place and confusing

to me – maybe it belongs at the end of the following paragraph rather than here, or perhaps I have misunderstood the role of those data in the foregoing discussion, as noted above.

- Table 2: To enhance clarity, it would be good to add "indirectly" in front of "derive".
- Fig. 7: Are the indirect results shown here from the "retrended" balloon data or from SouthTRAC? I assume the former but this should be stated explicitly. And why not show comparisons for both data sets, since you also provide SouthTRAC coefficients in Table 2? Also, in the caption: Indirectly and directly determined … (green and back) and --> Indirectly (green) and directly (black) determined … and
- L275: delete the comma after "Hemisphere"
- L276: $Cl_y$, where --> $Cl_y$ in cases where
- L277-278: "Since it was possible during SouthTRAC to measure the organic source gases, the $Cl_y$ from the direct measurements was used for further evaluation." In fact, while reading this section I wondered why the authors bothered to pursue the indirect approach when they actually have direct measurements of $CCl_y$. The discussion of the indirect $Cl_y$ calculation is particularly confusing, and it is not at all clear at this point what value it brings. So it would be helpful to add a pointer to section 4.4, where the indirect method is needed for the comparisons with $Cl_y$ in the Arctic, to better justify the inclusion of this discussion here. In addition, I think that, rather than "$Cl_y$ from the direct measurements", it would be more appropriate to say "$Cl_y$ determined semi-directly from the measurements".
- L279: Another thing that is not clear to me is why the fit coefficients for $N_2O$ are given in Table 2 if they are not being used here at all. On the other hand, several previous studies have used $N_2O$ to derive $Cl_y$, so I think it might be useful to expand the discussion of the $N_2O$ correlations. An obvious question that arises is: How well does the $Cl_y$ derived from fits with $N_2O$ agree with that based on the correlation with CFC-12, for both the balloon data and SouthTRAC? A figure similar to Fig. 7 could be added for $N_2O$.
- L284: only measurements were taken, which are polewards of 40° equivalent latitude --> only measurements polewards of 40° equivalent latitude are used
- L287: an air mass --> air mass
- L289-290: It seems to me that it might be better to exclude the measurements in the overlap region from the analysis rather than "double count" them. How many measurements fall into this category, and how would omitting them change the results?
- Fig. 8: I have a number of comments / questions about this figure.
  - It is stated that data are averaged over -40° to -90° – is the filtering of measurements obtained equatorward of 60° equivalent latitude (mentioned in section 3.1) only applied in calculating the vortex reference profile? Please clarify (here and in section 3.1).
  - The colors denoting the different regions have been changed. Previous figures used a consistent set of colors for these classifications, and it would be easier for readers if that same color scheme was used in all figures for which those classifications are relevant.
  - The lack of tick marks on the top and right-hand axes is annoying and makes it difficult to judge the values given in the text. In addition, the tick marks (especially the minor ones) that are present on the bottom and left-hand axes are too small to be easily seen.
  - Why does $Cl_{total}$ vary with altitude?
  - Elsewhere total chlorine was written "$Cl_{total}$", and that should be the case here too.
  - For consistency with the text, "-90° to -40°" should be "40°–90°S".
  - "mean averaged" is redundant

- L291: the measurements --> the SouthTRAC measurements
- L292: A reference is needed for the AGAGE results.
- L294-295: add commas after "330 and 390 K" and "390 and 400 K"
- L299-300: "at this altitude" – which altitude?  Where $Cl_y$ is maximum?  Please clarify.
- L301: Accompanying the minor SSW --> As a consequence of the minor SSW
- L301-302: It is not clear that the 16.4 million $km^2$ value quoted here refers to the maximum daily ozone hole area.  Moreover, it is not true that the 2019 hole was "the smallest since its discovery" – other holes in the mid-1980s had smaller maximum daily area values.  In any case, rather than quoting this value from the Ozone Watch web site, a better approach would be to reference the Wargan et al. [JGR, 2020] paper (already cited elsewhere in this manuscript); their Fig. 1d puts the area of the 2019 hole into climatological perspective.  Perhaps more importantly, it is not clear what the point of these sentences is.  Are the authors trying to imply that $Cl_y$ levels in the 2019 vortex played a role in the weak ozone hole that year?  Although $Cl_y$ abundances inside the vortex do vary from year to year as discussed previously by Strahan et al. [JGR, 2014], variations in lower stratospheric temperatures are the primary driver of variations in the strength of polar ozone depletion.
- L303: The title of this subsection suggests that the SouthTRAC and PGS comparison focuses on the polar region, but Fig. 10 and associated discussion includes mid-latitudes as well.  It may be true that comparison of $Cl_y$ in the Antarctic and Arctic polar vortices has not been done previously, as the authors assert, but it is not the case that no such comparisons have been performed in the midlatitudes.  In fact, total column $Cl_y$ (or, rather, $HCl+ClONO_2$) in the NH and SH mid-latitudes (Jungfraujoch and Lauder, respectively) and the trends therein are compared in the Ozone Assessment (e.g., Fig. 1-13 of WMO 2018).  It would be good to place their findings into the context of these (and possibly other) midlatitude results.
- L310: A separation --> The separation
- L311: is based on the above mentioned method based on --> is based on the above-mentioned method using
- S3, L34: the vortex and the mid-latitude profile during PGS is needed --> the vortex and the mid-latitude profiles during PGS are needed
- S3, L35: Phase --> phase
- L312-314: As shown in section 4.2, the indirect method … possible, proves to be comparable --> In section 4.2, the indirect method … possible, was shown to provide results comparable to those obtained by the semi-direct method
- L317: I assume that "2019" is a typo and that the same balloon data from 2009 as used for SouthTRAC were again used for PGS?
- S4, L37: Correlations function --> Correlation function
- L319 and L326: $Cl_{tot}$ --> $Cl_{total}$
- L319-320: This sentence ("The vertical coordinate of the classification was selected according to the displayed vertical coordinate.") is confusing and, if I understand it correctly, completely unnecessary, as the very next sentence makes clear that the two panels display the results as a function of θ and Δθ.  Also, I am curious why both vertical coordinates are shown here but not in Fig. 8.  Then in Fig. 10 only the tropopause-relevant coordinate is shown, with the argument that it allows for better comparison of $Cl_y$ in the two hemispheres.  It seems to me that it might have made more sense to show both views in Fig. 8, and then use only the tropopause-relevant coordinate in Figs. 9 and 10 for the reasons stated.

- L322: as it was done --> as was done
- L323-325: Several issues arise in this sentence.
  - As I noted above, information about the meteorological data on which this study depends needs to be provided much earlier in the manuscript, ideally in section 2.
  - I was surprised to discover that the analysis is based on NCEP reanalyses. Insufficient information is provided here to identify exactly which NCEP reanalysis is being used (NCEP-NCAR R1, NCEP-DOE R2, CFSR, or CFSv2), but that needs to be specified and the corresponding reference (i.e., published journal article) cited.
  - I have concerns if either NCEP R1 or R2 have been used for this analysis. Although both are still in widespread use, these reanalyses have been shown in several studies, including some recent papers stemming from the SPARC Reanalysis Intercomparison Project (S-RIP), to be unsuitable for most stratospheric studies (as noted in the S-RIP overview paper by Fujiwara et al. [ACP, 2017]).
  - What exactly is meant by "climatological" here? That is, how many years have been considered in the averages? Were the climatological means also calculated over the days covered by the respective campaigns, or are they monthly averages, or …? Are the climatological tropopauses being calculated over 40° to 90° (this latitude range is stated in the caption, but it is not clear exactly what it is referring to)?
- L326-327: "… abundance of total chlorine (Cltot) was lower in the stratosphere from the time of PGS (2015/2016) to the time of SouthTRAC (2019)" – the wording of this sentence is unclear. It should be rewritten to state that the abundance of total chlorine in the stratosphere decreased between the two campaign periods. It is very difficult for the reader to precisely judge the magnitude of this decline from the figure. Is the difference in the estimated PGS and SouthTRAC values of $Cl_{total}$ consistent with expectation given the known decreasing trend in stratospheric chlorine loading? This is a key point.
- L329-330: the SouthTRAC profile increased stronger and values become more than 435 ppt larger than during PGS within the vortex at equal potential temperatures --> the SouthTRAC profile increased more steeply, reaching values more than 435 ppt larger than those during PGS at the same potential temperatures
- L330: Differences become --> Differences are
- L331-332: This sentence ("Inside the vortex … during PGS.") is entirely redundant with the second sentence of this paragraph and should be deleted.
- L332-333: Although close together between 20 and 25 K Δθ, the difference of $Cl_y$ increased to 565 ppt at 65 K Δθ --> Although the two $Cl_y$ profiles lie close together between 20 and 25 K Δθ, the differences between them increase to 565 ppt at 65 K Δθ
- Fig. 9:
  - Again, please add tick marks on the top and right-hand axes.
  - 40° to 90° --> 40° to 90° equivalent latitude
  - Delete "and as a function of potential temperature difference to the local tropopause" in line 3 – this information is provided in the description of panel (b) below.
  - "mean averaged" is redundant
  - SouhTRAC --> SouthTRAC
- L335: the latitude --> the geographic latitude
- L337-338: and better allow for --> and allows for better

- L339-340: It might be interesting to know how many points contribute to each latitude-altitude bin in both hemispheres. Is there a minimum threshold for the number of points in each bin? Perhaps bins with very disparate numbers of points contributing in the NH and the SH could be marked in some manner.
- L341: add a comma after "latitudes"
- L344: Highest levels of $Cl_y$ reach 386 ppt more $Cl_y$ during PGS --> The highest values of $Cl_y$ reached are 386 ppt greater during PGS
- L345: vortex of each hemisphere is --> vortices of the two hemispheres are
- L347: I do not believe that "sporadic", which means "infrequent" or "intermittent", is the right word here, especially as no time information is conveyed in this plot. Perhaps "weak" or "moderate" would work, if I have understood the point the authors wish to make.
- L348: it is not clear what "it" is referring to here -- $Cl_y$?
- L349: for both --> in both; add a comma after "hemispheres"; there is no need to introduce the acronym for ExTL since it is not used again in the manuscript
- Fig. 10: Please add tick marks on the top and right-hand axes as well as minor tick marks. Also, the color bar label should indicate that these are differences, not raw $Cl_y$ values.

Section 5
- L352: Using an extended method according to Greenblatt --> Extending the method of Greenblatt
- L353-355: It is stated that, compared to coarser-resolution PV, the method to define the vortex used here allows small structures such as filaments to be resolved. First, as noted earlier, modern meteorological reanalyses provide PV at fairly fine resolution. Second, no evidence is presented in this paper that any such filaments were actually resolved using their approach. So I am not convinced that a PV-based definition would not have been adequate.
- L358-360: The authors are correct when they point out that the dynamical tropopause would be more appropriate for this kind of study than the thermal tropopause. Unfortunately, the use of the WMO tropopause raises questions about the value of this investigation. I do not really understand how the authors can say that "no dynamical PV tropopause data is yet available for the SouthTRAC campaign". In fact, high-resolution PV fields are available from multiple reanalyses. There is abundant literature discussing which PV values are most appropriate for defining the dynamical tropopause, depending on the hemisphere and isentropic surface, etc. So it is not clear to me why the authors could not have chosen representative PV values and performed their own interpolations to the in situ measurement locations to determine the local tropopause. But even if the authors are not set up for those calculations, they could still do more to reassure readers that use of the thermal tropopause does not substantially affect their conclusions. Keber et al. used the dynamical (2 PVU) tropopause, so that information is readily available for PGS. Some simple comparisons between the WMO and PV tropopauses for the period of the PGS campaign and examination of the impact the differences between them have for Figs. 9 and 10 would be informative.
- L364: CFC-12 on --> CFC-12 in
- L365: channel --> channels
- L372: add a comma after "SouthTRAC"
- L374: add a comma after "2015/2016"

- L375: "At the time of publication, it is not known that such a comparison has already been made". First, this statement is somewhat ambiguous. I think the authors mean "To our knowledge, such a comparison has not been published previously." Second, they should be a bit more precise in the language here, focusing on $Cl_y$ in the polar vortices, given the discussion of mid-latitude $Cl_y$ in the WMO Ozone Assessment as noted above.
- L382: would be negative to about --> are estimated to be negative at about
- L382-383: The difference of $Cl_y$ inside the respective vortex is significant and even larger than the inter annual variations reported by Strahan et al. (2014) --> The differences in $Cl_y$ values inside the two vortices are substantial and even larger than the interannual variations reported by Strahan et al. (2014) for the Antarctic. (See earlier comments on a similar statement in the abstract.)
- L384: of the respective --> in each
- L385: respective campaign only shows a section of the respective winter seasons. These sections do not match --> respective campaigns only show a portion of the winter seasons. These intervals do not correspond
- L386: the respective polar vortex --> the two polar vortices
- L389: add a comma after "SouthTrac"
- L391: First, citations need to be added to support this statement about the BDC being stronger during NH winter than during SH winter. Second, I think it would be more appropriate to move the conjecture about a possible cause for the interhemispheric disparity in $Cl_y$ to section 4.4, where these results are discussed, rather than have it in the "Summary and Conclusions" section. In addition, I'd like to see the discussion of the discrepancy and its possible causes developed a bit more, and put into context of the midlatitude results in WMO 2018 (mentioned above). Also, I suggest some wording changes: on the northern winter hemisphere than in the southern winter hemisphere due to stronger Brewer-Dobson circulation --> during winter in the Northern Hemisphere than during winter in the Southern Hemisphere due to the stronger Brewer-Dobson circulation.
- L393: in higher --> at higher
- L394: exhibits a larger variability as it is more effected --> exhibits larger variability as it is more affected; also, capitalize "Southern Hemisphere"
- L395": side -- > hand; is less effected --> is typically less affected

---

## Author Comment (AC1)

We would like to thank both reviewers for their constructive comment on our manuscript. In the following, we address the respective proposals for improvement. Changes are explained in detail, answering each referee point by point. Reviewer comments are in normal font. Our answers are in italic and changes to the manuscript in blue.

**Response to Referee #1**

**Main topic areas to address in revision**

Source of tropopause data, lines 155-165. Each profile is analyzed with respect to its height above the tropopause yet nowhere is it stated where this tropopause information comes from. What is the source of the 'local tropopause'? Is the aircraft doing frequent profiling to identify a tropopause (through a temperature minimum)? This must be explained. Later in the paper (line 322) there is talk of using a thermal tropopause, but this doesn't note the data source. Surely the climatological mean tropopause is not used for the constituent analyses. This issue is very important as the results depend on what information is used to determine this coordinate.

The unknown source of your tropopause data leads right into 3 related issues: using the thermal tropopause for your analysis, the decision not to use potential vorticity from a reanalysis, and the use (and source of) of equivalent latitude.

We addressed this issue by including a new subsection "Meteorological data" in the section "The SouthTRAC campaign". In this subsection, we shortly explain what kind of local tropopause we used in this analysis. Regarding the following comment, we switch to the more appropriate dynamical tropopause. We go into more detail on the next comment.

"[...]

**2.1 Meteorological data**

[...]

The local tropopause information along the flight tracks of HALO was created using the Chemical Lagrangian Model of the Stratosphere (CLaMS) (e.g., Grooß et al., 2014). The underlying meteorological data are taken from ECMWF ERA-5 data (Hersbach et al., 2020). In this work, the potential vorticity (PV) based dynamic tropopause is used (e.g., Gettelman et al., 2011), taking the 2 PVU (potential vorticity unit) for the dynamical tropopause. Since the PV tropopause is not physically meaningful in the tropics, potential temperature level of 380 K was taken as tropopause if the 2 PVU level lies above. [...]"

Use of thermal tropopause. The thermal tropopause is inappropriate in polar winter because the temperature profile is often isothermal – the dynamical (PV) definition is needed. See the analysis of dynamical v. thermal tropopause in Zaengl and Hoinka (J. Climate 2001). This means you need PV from reanalysis data (ERA5, MERRA2...whatever). These fields are available in fairly high resolution (0.5 degree or

better) and even with interpolation they may more accurately identify the tropopause than does temperature in an atmosphere with weak vertical temperature gradients. Whatever your final analysis method is, you will need to justify it based on 1) showing that reanalysis PV doesn't give sensible results, or 2) proof that the temperature tropopause actually makes sense at high latitudes in winter.

We fully agree that the thermal tropopause is inappropriate in polar winter. Therefore, the analysis was carried out again using the dynamical tropopause, based on PV from ERA-5 data. Where we have listed the necessary information for dynamic tropopause in the manuscript can be found in the previous comment.

Source of equivalent latitude. Around line 156 equivalent latitude is said to be used to sort the flight data: where does your equivalent latitude come from? Just 20 lines earlier it is stated why use of reanalysis data and its coarse resolution is a drawback to the analysis, but where to you think the equivalent latitude information comes from? It is calculated based on global PV fields which, by necessity, come from a reanalysis. So although you haven't explained the source of either the tropopause or equivalent latitude data used, it seems clear that you are using reanalysis info. This should be acknowledged. It's fine if you want to use the Greenblatt method for identifying profiles, but I'm not sure it's accurate to say that the reanalysis PV isn't good enough for your analysis. (Have you tested this?)

We are aware that the equivalent latitude information is based on global PV fields, which come from the reanalysis, mentioned in the answer to the previous comment. At this point, however, we only us the equivalent latitude to pre-filter the measurements which helps the filter procedure to find the lower and upper envelope. The vortex and mid-latitude profiles are still based on the N2O measurements. We added "(Butchart and Remsberg , 1986)" as reference to the equivalent latitude. Earlier in this section, we replaced the statement about the resolution of PV with the following:

"[...] PV is a model-derived quantity. Although the underlying meteorological reanalysis have a fairly high resolution these days (e.g., Hersbach et al., 2020), small-scale features like vortex filaments with different chemical compositions may not be well resolved [...]"

Antarctic and Arctic vortex size differences. These play a role in whether Figure 10 is meaningful. The Antarctic vortex mean edge is at 60S equivalent latitude – it's a large vortex. (Sep avg ~35 million km2). Even in 2019 the Antarctic vortex at 360K had an average size until the last third of September. I'm not certain what the Arctic vortex mean edge is but it's probably closer to 70N equivalent latitude (avg March vortex <20 million km2). Because of this the hemispheric difference plot using equivalent latitude coordinate doesn't make physical sense in the 60-70 degree range. In Fig. 10 the difference at 65 degrees will be a comparison of the Antarctic vortex with the northern midlatitudes. Since the hemispheric vortex profiles are already compared in Fig. 9, perhaps add panels to that figure showing the NH/SH midlatitude differences on the 2 vertical coordinates. I don't think Fig. 10 is very useful and could be eliminated.

We agree that the different sizes of the Artic and Antarctic vortices limit the hemispheric comparison. In the text we have already indicated that this comparison is limited in the polar region and vortex edge region due to different sizes, strength, and transport barrier of the vortices. However, the figure also shows a comparison of lowermost stratosphere of the mid-latitudes of the Northern Hemisphere and Southern Hemisphere. We extended the discussion about the differences regarding mid-latitudes Cly values. Thus, we would like to keep figure 10 in this study. Section 4.4 was extended by:

"[...] It must be noted that the polar vortices of the two hemispheres are different in size, stability and strength of the transport barrier. The comparison on equivalent latitude is therefore only possible to a limited extent. Nevertheless, possible reasons for the observed differences can be derived from the hemispheric difference of the Brewer-Dobson circulation, using the age of air as a common metric for transport. Konopka et al. (2015) showed, that north of 60 °N, age of air is always younger than south of 60 °S in the same season, implying a stronger residual circulation in the Northern Hemisphere. Analysis of Haenel et al. (2015) revealed differences in age of air trends in the lowermost stratosphere of the mid-latitudes of Northern and Southern Hemisphere with a positive trend in the Northern Hemisphere and a negative trend in the Southern Hemisphere. In addition, Mahieu et al. (2014) reported long-term total column data for HCl and ClONO2 (representing Clv) in the stratosphere, at Jungfraujoch (46.5 °N) and at Lauder(45 °S), though the end of 2016. A negative trend of Clv is observed at both stations but with a non-significant trend for the Jungfraujoch data over the last decade and a slightly larger negative trend from the Lauder data. Furthermore, lower-stratosphere HCl from the Global Ozone Chemistry And Related trace gas Data records for the Stratosphere (GOZCARDS)shows larger decreases at southern latitudes and increases at northern mid-latitudes (Froidevaux et al., 2015). Thus, higher values of Cly in the mid-latitudes during PGS seems to be plausible. [...]"

Figure 7 discussion (I. 270). What data are used to calculate the mean age shown in Figure 7? There is discussion just prior to this about mean age and the 'arrival time' – is this what's plotted? Maybe I'm missing something but I cannot see what observations or information are used to produce mean ages. But a bigger problem is that mean age values of 5 years are shown for Cly of 1500 ppt. This can't be right. The best estimate is closer to 3000 ppt at 5 years. See for example Newman et al (ACP 2007) or Strahan et al (2014, JGR) or compare the N2O values you observed with the ACE N2O/mean age mapping in Strahan et al (JGR 2011). No data have been presented that demonstrate that SouthTrac data, which are entirely from 390K and below, have such old age. It's more likely the maximum age there is near 3 years.

We used SF6 measurements of the GhOST-ECD channel in a time resolution of around 1 minute and a precision of 0.64%. The usage of SF6 to calculate mean age was already mentioned in the manuscript in section 4.2 in the paragraph after equation 1: "[...] The concept of the age spectrum can be used to determine mean age values based on observations of chemically inert tracers in the stratosphere. For this purpose, in addition to the age spectrum, tropospheric time series of the inert tracers

are required (Engel et al., 2002). This was done for the SouthTRAC campaign by using SF6 measurements of the GhOST-ECD and tropospheric time trends taken from the AGAGE (Advanced Global Atmospheric Gases Experiment) Network (Prinn et al., 2018). [...]".

 $SF_6$  is a commonly used tracer for age of air, but recent research suggests, that its lifetime has been overestimated and thus it may be giving higher mean ages. (e.g., Ray et al., 2017 and Leedham Elvidge et al., 2018). Leedham Elvidge et al., 2018 also provides correction functions of SF6 derived mean age. We compared  $N_2O$ /mean age mapping during SouthTRAC with ACE  $N_2O$ /mean age mapping in Strahan et al., 2011. Without SF6-based mean age correction, toward older air, the difference reaches roughly one year, whereas with correction, difference becomes roughly half a year. It must be mentioned that we compare ACE annual mean N2O from mid-latitude and tropical observations with mostly mid-latitude and polar observations from September to November 2019 during the SouthTRAC campaign. In addition, Konopka et al. 2015 showed, that signatures of old air within the Antarctic vortex propagate down to 340K of potential temperature and in the polar regions around 380K, oldest air anywhere can be found in September south of 60°S. The maximum  $Cl_{\rm v}$  value during SouthTRAC was 1668 ppt at 4.9 years of mean age, 4.2 years when corrected by a correction function from Leedham Elvidge et al., 2018. The corrected mean age seems to fit better for the given  $Cl_v$  value, although not fully comparable with Strahan et. Al., 2014. Furthermore, Clv calculation using N2O instead of CFC-12 as the reference substance leads to similar values (see new Figure S7 in the supporting information). To alert the reader that the mean age shown here may be somewhat overestimated, we have added a second x-axis in Figure 7 showing the profiles at corrected mean age. However, this is only intended as a guideline, as it represents only one possible correction of the mean age. in the paragraph to figure 7, we have thus included the following:

"[...] Recent research suggests that SF6-based mean age is biased because the suggested lifetime has been overestimated (e.g., Ray et al., 2017). As a guideline, Figure 7 additionally shows a corrected mean age of air using one of the linear fit functions from Leedham Elvidge et al. (2018), based on a comparison of SF6-based mean age with a combined mean age based on five alternative age tracers. In this study, however, the uncorrected mean age of air is used. [...]"

The vortex Cly profile differences (Fig. 9) imply interhemispheric (IH) differences in mean age (and age spectrum) in the lower branch of the Brewer Dobson Circulation (BDC). These are presumably driven by transport and indicate that the NH lowermost stratosphere is younger than the SH. I believe such differences are expected – see for example Birner and Boenisch, ACP 2011. Simulations driven by reanalyses may reproduce these differences (as well as the midlatitude differences), but what about chemistry climate models (CCMs)? It would strengthen this paper to put your measurements in the context of what they tell us about IH differences in the lower BDC. These measurements help confirm our thinking about the stratospheric circulation. You might comment on whether and why it's important for CCMs to reproduce similar hemispheric differences.

We agree, that including a discussion with chemistry climate models would additionally strengthen this manuscript. However, it would go beyond the scope of this evaluation and cannot be dealt within a few sentences in section 4.4. As an outlook, we included a statement regarding chemical climate models in the summary and conclusion to pinpoint, that such interhemispheric difference should be captured therein:

"[...] These hemispheric differences can also be found in simulations based on reanalysis, e.g., Konopka et al. (2015). A comparison of the available data with chemical transport models should be subject to further studies. Furthermore, such interhemispheric differences should also be captured by chemistry climate models, which are not only used to understand past changes but also predict future changes in chemical composition [...]"

 Lines 220-240. Isn't the semi-direct Cly calculation nearly identical to Schauffler's method (JGR 2003)? While that was referenced much earlier it seems far more relevant here. If this is true then you can reference Schauffler here and shorten the description, only describing any way your method differs.

The methods are quite similar but with the difference, that Schauffler et al., 2003 did not use relevant age.

This paper uses measurements to calculate Cly from only the long-lived Cl-containing species, but there are contributions from short lived (VSL) Cl species too (e.g., CH2Cl2, C2Cl4). It should be explicitly stated that such species are excluded from this study. The estimated size of this neglected contribution could be noted. See Hossaini et al., JGR 2019 for an observational and modeling study that estimates the VSL Cly impacts.

You are correct. The contribution from the short-lived chlorinated substances is small but significant, as revealed in Hossaini et al., 2019. We included the information, by focusing only on the long-lived substance in this analysis, in the last paragraph of the introduction. In addition, we listed the contribution from the very-short lived chlorinated substances to the total stratospheric chlorine as well as the contribution in inorganic form from Hossaini et al., 2019 and from the WMO Report 2018.

"[...] Calculations of  $Cl_v$  are based only on long-lived chlorinated substances. There is an additional contribution to total stratospheric chlorine from the very short-lived chlorinated substances. Engel and Rigby (2018) estimated a contribution of 115 (75 – 160) ppt from very short-lived chlorinated substances for 2016. Hossaini et al. (2019) estimated a contribution of about 111 ± 22 ppt, of which 13 ± 4.6 ppt are already in inorganic form, which is not considered in this analysis. [...]"

**Minor Comments**

- Title: It doesn't make sense to say 'comparison' without saying what you're comparing with. The abstract reveals it is Cly in the Arctic LMS a few years earlier.

Perhaps 'Comparison of Cly in the Arctic and Antarctic lowermost stratospheric vortices'?

We changed the title for more clarity.

"[...] Comparison of Inorganic Chlorine in the Antarctic and Arctic lowermost stratospheres by separate Late Winter aircraft measurements [...]"

First sentence of the abstract. You're really talking about stratospheric inorganic chlorine so please say so. And the strat inorganic chlorine comes from all chlorine containing source gases with a lifetime of more than 5 months (see Hossaini et al JGR 2019), so that's the long-lived and many of the VSL species.

We extended this sentence to clarify that we were considering stratospheric inorganic chlorine. We also add to the sentence the information that chlorinated very-short lived substances contribute to the inorganic chlorine.

"[...] Stratospheric inorganic chlorine ( $Cl_y$ ) is predominantly released from long-lived chlorinated source gases and, to a small extend, very short-lived chlorinated substances. [...]"

line 16. "Based on the results of these two campaigns, the difference of Cly inside the respective vortex is significant and larger than reported inter annual variations." Each campaign was a single winter – there is no information on interannual variability. I realize you are citing another paper on Cly variability in Antarctic lower stratosphere, but what about Arctic variability? Unknown? As written this statement is misleading and not supported.

Our results are based on one winter each in the Artic and Antarctic. Therefore, interannual variations are not examined in this study, as the referee mentioned correctly. Referee #2 also marked the word "significant" as not applicable in this context. We have taken up the suggestion of referee #2 for improvement and adjusted the sentence accordingly.

"[...], the differences in  $Cl_y$  inside the two vortices are substantial and larger than the inter-annual variations previously reported for the Antarctic. [...]"

- line 20: '1980-ies' is 1980s

Done.

- line 23 OCIO isn't involved in depletion. Null cycle.

OCIO has been removed from the list of substances.

"[...] Chlorine substances involved in rapid ozone depletion are Cl,  $Cl_2$ , ClO, and ClOOCl, and can be summarized as  $ClO_x$ . [...]"

- line 45. In both hemispheres, polar winter temperatures are above radiative equilibrium because of dynamical (wave-driven) heating. It's not just the absence of insolation.

You are fully correct. Although temperatures within the polar vortex are basically driven by radiative processes, they are also determined by dynamics and the transport of heat by atmospheric motions. But we don't want to go into too much detail here about how the temperatures inside the vortex come about. For this reason, we will shorten the statement in the manuscript by excluding the statement about the ultraviolet heating. The new sentence is:

"[...] Due to a temperature difference and consequently to a latitudinal pressure gradient between the polar and mid-latitude stratosphere (e.g., Schoeberl and Hartmann, 1991), a state with a strong westerly wind in the stratosphere is established (polar night jet). This jet acts as a transport barrier, leading to strong latitudinal gradients of potential vorticity and long-lived substances like N2O results (e.g., Hartmann et al., 1989) [...]"

 Since the paper is comparing Cly in the 2 vortices, do you have any comments/conclusions about differences in maximum potential O3 depletion in each LMS vortex?

Comments and conclusions about differences in maximum potential  $O_3$  depletion in the respective LWS vortex is beyond the scope of this manuscript. Beside the comparing  $Cl_y$  in the two vortices, mostly important is the chlorine activation e.g. the production of active chlorine from the reservoir species on polar stratospheric clouds.

- line 85. You don't need to define payload

The definition of payload has been removed.

- line142. I would emphasize that you mean mixing within the vortex. E.g., "...benefits mixing on isentropic surfaces inside the vortex..."

We have rewritten this sentence for more clarity.

"[...] In addition, N2O has a small variability inside the vortex on constant isentropic surfaces (variability of about 6 ppb (Greenblatt et al., 2002a)). This is an indication of well mixed air inside the polar vortex due to the long isolation in polar winter. [...]"

- line 143. 'descent' not 'descend'

The sentence containing this typo was removed due to repetition.

- line 146. What you're describing is that as you approach the tropopause the vortex ceases to exist so there is no longer a barrier to mixing. There is nothing to distinguish.

We have rewritten this sentence for more clarity, that the vortex barrier vanishes towards the tropopause.

"[...] Towards tropopause altitudes, the transport barrier of the polar vortex disappears, and a classification is not possible. [...]"

- line 152. By "Stratospheric transport and mixing is related to the isentropic surfaces" do you mean that transport and mixing occur on isentropic surfaces? This is unclear, please rephrase.

We do mean quasi-isentropic mixing. As this is not the only effect on the composition of air in the UTLS, we further extended this sentence.

"[...] The composition of the lowermost stratosphere is affected by the diabatic decent inside and outside the polar vortex and quasi-isentropic mixing with air from lower latitudes. [...]"

 line 155 'had contact to the vortex core'? This is awkward and unclear. Is the intended meaning that the reference profile was entirely inside the vortex, away from the edge and mixing at the edge?

This sentence was rewritten for more clarity.

"[...] The vortex reference profile (see Fig. 2) was generated from all flights that are assumed to contain measurements within vortex air. [...]"

- line 189, 'extensively'

By rewording this sentence, this word is omitted.

 lines 194-7. "The metric describing the combined effect of all ozone depleting substances (ODS) as an equivalent amount of inorganic chlorine in the stratosphere, related to tropospheric source gases in a simple, is the equivalent effective stratospheric chlorine (EESC)". Awkward sentence. I suggest: "Equivalent effective stratospheric chlorine (EESC) is a simple metric that sums the effect of all ozone depleting substances (ODS) as an equivalent amount of inorganic chlorine in the stratosphere..."

We have rewritten this sentence according to the suggestion. In addition, this sentence, and the following regarding the EESC was included in the introduction and removed at this point.

- General comment on 'pre-filtered', 'pre-required'. Drop the 'pre', it's not needed.

The data is filtered only because the subsequent procedure for the upper and lower envelope works well programmatically. We would like to keep "pre-filtered".

- To clarify the meaning, I'd suggest a slight rewording (line 220): "Cly can be calculated as the difference between total chlorine entering the stratosphere and the organic chlorine that remains bound in chlorinated halocarbons"

To make this part of the section clearer, we have rewritten it.

"[...] Organic chlorine (CCly) can be calculated directly from the up-sampled GhOST-MS measurements. Thus, Cly can be calculated from Eq. 1 if the mixing ratios of the major chlorine-containing substances at the stratospheric entry point (Cltotal) are known. [...]"

- line 256: 'rations'...you meant 'ratios'

Done.

 line 239: "in the following..." Move this statement to the beginning of the next paragraph where you actually describe the semi-direct method and then reword. For example you can begin the next paragraph with: "The semi-direct Cly calculation is used in the case where no measurements of chlorine containing substances are available. This method is based on [trace gas?] correlations found in previous measurement campaigns."

The following paragraph does not describe the semi-direct Cly calculation. Instead, it describes the indirect Cly calculation based on scaled correlation from previous measurement campaigns, as stated at the beginning of the paragraph. The semidirect Cly calculation is described in the first paragraph of this section. The sentence mentioned here serves to introduce the term "semi-direct Cly" as a term for Cly from in-situ measurements, which reappears later in the paper. We would leave this sentence at this point in the text.

- Figure 9. It would be more useful to give titles to each panel other than 'a' and 'b'. Those labels normally go inside the panel.

A figure earlier in the paper also uses (a) and (b). For the sake of consistency, we would like to leave it at that, with a detailed description in the figure caption.

- Fig. 9 shows that the NH data reaches 405K while in the SH 385K is the maximum. Do these represent the same maximum altitude for flights, and does this difference indicate that the SH LMS vortex is much colder than the NH vortex?

Figure 9 shows only the maximum potential temperature captured inside the respective vortex. However, this also depends on the flight patterns. Flights during the PGS campaign were operated from Kiruna (68° N). In contrast, flights during the SouthTRAC campaign were operated from Rio Grande (53° S). The longer distance to and from the vortex had to be considered for the flight planning and has an influence

on the maximum potential temperature that could be reached inside the respective vortex. The maximum height of the flights cannot be taken from this.

- Fig. 10 is saying that the NH midlatitudes are older than the SH. Anything to say about that?

We extended the discussion on the differences of NH and SH mid-latitudes. The extended passage can be found in the comment "Antarctic and Arctic vortex size difference".

Lines 298-302: Since you are identifying profiles as vortex, midlatitude, or edge already, I imagine the effect of the SSW is that you measured more edge and midlatitude profiles in November than you might have in another year. But you've pointed out that vortex descent has essentially ceased by September, so as long as you are sampling vortex air that hasn't mixed with midlatitudes, I would expect that the Cly profiles you measure aren't affected by the SSW. In other words, the mean age profile for air masses that are truly vortex air masses might well look similar to other years. Thus, the statement implying that the fraction of CCly found as Cly being affected by the SSW right may not be right. On the other hand, there aren't data from other years and maybe this point should be made. There is no information on interannual variability.

Thank you for mentioning this point. We do not want to create a link between the amount of  $Cl_y$  and the size of the ozone hole or the minor SSW event. Instead, we wanted to note that the minor SSW event led to an early chlorine deactivation. We further use a more appropriate reference for the size of the Antarctic ozone hole.

"[...] Inorganic chlorine within the vortex could be obtained from  $\Theta$  between 330 to 385 K. Cly inside the vortex increases significantly up to a value of 1687±19 ppt. Thus, in late winter and early spring at this altitude about half of the recorded chlorine is found in inorganic form. Despite this amount of inorganic chlorine in the lower stratosphere, the total polar ozone column was higher than usual in September 2019. As a result of the minor SSW event, chlorine deactivation began earlier in 2019 and the ozone hole was about 10 x 106 km2 in size, thus only 20 % of that in 2018 mid-September (Wargan et al., 2020). [...]"

In general, 'data' is a plural noun, thus, 'data are...' not 'data is'.

Done.

**Response to Referee #2**

**Title**

- The comparison being made in this paper is not clear from the title. It would be better to craft a different title capturing the idea that in situ measurements from separate aircraft campaigns are being used to compare Cly abundances in the Antarctic and the Arctic LMS.

**We changed the title for more clarity.**

"[...] Comparison of Inorganic Chlorine in the Antarctic and Arctic lowermost stratospheres by separate Late Winter aircraft measurements [...]"

**Abstract**

- L1: The wording "... long-lived chlorinated source gases. These include the reservoir species" seems to imply that HCl and ClONO2 are chlorinated source gases. Thus, These include --> Cly includes

Done.

- L5: in late winter --> in austral late winter

Done.

 L8: The sentence "Cly from a scaled correlation was compared to directly determined Cly ..." is confusing, since the previous sentence states that not all source gases were measured during PGS. It needs to be made clear that this "validation" was performed for SouthTRAC.

We have rewritten this sentence to create less confusion and to show that SouthTRAC data were used to validate Cly from a scaled correlation.

"[...] Using SouthTRAC data, Cly from a scaled correlation was compared to directly determined Cly and agreed well. [...]"

L12-13: The values (40%, 20%) here appear only in the abstract and conclusions, not in the main text. In my opinion, it is not appropriate to include "new information" in the abstract and summary sections that has not been thoroughly discussed in the body of the paper. Please add some corresponding statements in section 4.

**We include this information into section 4.4 after the discussion of Figure 9.**

"[...] The fractions of total chlorine which are in the form of  $Cl_y$  inside the vortex and in the mid-latitudes during PGS at the same distance from the local tropopause as for the highest values within the vortex during SouthTRAC, are about 40% within the vortex and about 20% in the mid-latitudes. [...]" - L13-14: Differences inside the respective vortex reaches up to 565 ppt more --> Differences inside the two vortices reach as much as 565 ppt, with more

We have rewritten this sentence according to the proposal.

- L15-16: As far as is known --> To our knowledge; within the respective polar vortex --> within the Antarctic and Arctic polar vortices

**Done; Done.**

- L16-17: "the difference of Cly inside the respective vortex is significant and larger than reported inter annual variations". I have a number of comments on this sentence:
  - The authors have not done a statistical analysis, so I do not think that "significant" is an appropriate word here.
  - This statement could be erroneously interpreted as implying that their study examines interannual variations. Moreover, the Strahan et al. [2014] paper on which this statement is based looked only at the Antarctic, not the Arctic.
  - The word "respectively" is used many times throughout the manuscript, sometimes (as here) incorrectly. I have recommended alternative wording in a few places.
  - Thus, I suggest instead: "the differences in Cly inside the two vortices are substantial and larger than the interannual variations previously reported for the Antarctic".

We agree that "significant" should not be used in this context. Furthermore, our results are based on one winter each in the Artic and Antarctic. Therefore, interannual variations are not examined in this study. We are happy to use the appropriate suggestion and change the sentence accordingly.

"[...], the differences in Cly inside the two vortices are substantial and larger than the inter-annual variations previously reported for the Antarctic. [...]"

**Section 1**

- L20: 1980-ies --> 1980s; also, pre-dominantly --> predominantly

Done.

- L23: substances, which are involved --> substances involved

Done.

- L23-24: OCIO is a consequence of, and thus a good qualitative indicator of, halogen activation, but it does not itself participate in ozone destruction, as this sentence implies. Thus it is not normally considered part of CIOx.

OCIO has been removed from the list of substances.

"[...] Chlorine substances involved in rapid ozone depletion are Cl,  $Cl_2$ , ClO, and ClOOCl, and can be summarized as  $ClO_x$ . [...]"

- L25: within --> through

Done.

 L26: While I applaud the recognition of some of the original papers, I think it would be good to also include some review articles (e.g., Solomon, Rev Geophys, 1999) and/or some more up-to-date citations (e.g., the most recent WMO Ozone Assessment).

We added Solomon et al. 1999 as a reference because this paper clearly illustrated the activation of chlorine form the reservoir species.

- L30: Citing only the Newman et al. [2004] paper here gives the erroneous impression that it is the only relevant reference. At the very least, "e.g.," needs to be added to this citation. This is another instance where it might be appropriate to cite the WMO Report.

"e.g." was added for this citation.

- L35: used again --> again used

Done.

- L39-40: Citations for the long-term trends in Cly and N2O should be given.

As a reference for negative Cly trends we included Newman et al. 2007. As a reference for a positive trend of N2O, we included chapter 1 of the 2018 Ozone Assessment Report.

- L45: between polar --> between the polar

Done.

 L46: Here and throughout the manuscript, when "mid-latitudes" is used as an adjective to modify a noun (e.g., stratosphere, profile, reference, etc.), it should be singular: "mid-latitude". When it is used as a noun itself (as in "at mid-latitudes"), then it is plural.

"Mid-latitudes" was changed to "mid-latitude". In the following, this term is adapted, according to its use.

L46: add "e.g.," in front of "Schoeberl"

Done.

- L48: add "e.g.," in front of "Hartmann"

Done.

- L62: SouhTRAC --> SouthTRAC

Done.

- L65: delete duplicate period after "4"

Done.

**Section 2**

- L70: capable to reach --> capable of reaching

**Done.**

L72: Rio Grande (RGA), Argentina (53°S, 67°W) --> Rio Grande, Argentina (RGA, 53°S, 67°W); also, regions for --> regions of

Done; Done.

- L74-75: The actual dates (not just "September" and "November") for the two phases of the campaign should be given.

The two sentences about the campaign phases were supplemented with the start and end dates of the respective phase.

"[...] The first phase took place from September 6th to October 9th, 2019 to target the dynamical objectives (e.g., Rapp et al. 2020). The second phase took place from November 2nd to 15th, 2019 to, among others, sample polar vortex remnants. [...]"

L77-80: 9 transfer flights + 10 Phase I flights + 3 Phase II flights = 22 total flights, not
23 as stated in L77

There was a short local flight on Sal during the first transfer from Oberpfaffenhofen to Rio Grande, which I did not list in the text. I apologize for the inconsistency and add the following sentence:

"[...] Within the first transfer from EDMO to RGA, there was an additional local flight operated from SID. [...]"

- L78: Rio Grande (RGA), Argentina --> RGA

As I mention the three-letter codes of the other airports at this point, I would keep the current wording for Rio Grande.

- L84: Beside --> Besides (or, "in addition to"); also, delete comma after "instruments"

Replaced "Beside" with "In addition to"; Done.

 L108-109: Given the in-flight conditions as described in this paragraph, it makes sense that the mean precisions during the campaign are poorer than those measured in the lab. So it is 2 puzzling that the mean precisions during the flights improved over those measured beforehand for methyl chloroform. Do the authors have an explanation?

Unfortunately, no explanation could be found. The chromatographic peaks of methyl chloroform were examined closely without finding any problem or abnormality.

- L108-112: It is stated that for CFC-113 "the amount of water in the analytical system should be kept as low as possible", but it is not clear whether that was actually done during the campaign. The implication is that in fact this was not done properly and that is why the in- flight precision of CFC-113 is so much worse than that determined in the lab, but this point needs to be clarified. I also think it is questionable whether the measurement of CFC-113 really stands out as an "exception" (L108) for its degraded performance. In fact, the precision estimated during the flights is even worse relative to the pre-campaign value (a factor of 5 difference) for CFC-11.

We have rewritten the section about the poorer precisions of CFC-113 and included CFC-11 to this section.

We also mention in the text, that we dry the air before pre-concentration. More detailed information about drying can be found in the already listed publication about the GhOST-MS.

"[...] The exceptions are CFC-11, CFC-113, and methyl chloroform. [...] It is difficult to determine exactly what the poorer precisions of these substances can be attributed to. The chromatographic peak of CFC-11 is very narrow and variable environmental conditions (due to changes in altitude, pressure, and temperature in the cabin) have an influence on the peak shape. The amount of water in the analysis system is also important and is kept as low as possible by drying before pre-concentration. As the chromatographic peak of CFC-113 is close to the chromatographic peak of water, small changes in water can affect the chromatographic peak of CFC-113. [...]"

- L109: precision CFC-113 --> precision of CFC-113
  - Done.
- L109: It seems odd to me that the authors make the effort (L85) to define "payload", which is a widely known and not particularly technical term, but not "elutes", which many of their readers (including me) may not know. Also, should it be "CFC-113 is eluted by water" rather than "CFC-113 elutes near water"?

The definition of payload was removed from the manuscript.

To avoid having to introduce and explain the term "elute" unnecessarily, the sentence was rewritten, and "chromatographic peak" was used for the description. Changes can be seen in comment to L108-112.

- L110: add a comma between "water" and "the amount of water"

Done.

- L111-113: Precision values are not given for GhOST-ECD SF6 either in the text or in Table 1, yet SF6 measurements are mentioned later in the paper.

We included the precision of  $SF_6$  in the text in section 2.2 in addition to the precision of CFC-12 with the GhOST-ECD.

"[...] CFC-12 and SF6 with the GhOST-ECD channel were measured with a precision of 0.2 % and 0.64 %. [...]"

 Table 1 caption: have been determined shortly before the SouthTRAC (ST) campaign and mean precisions during the flights --> have been determined in the laboratory shortly before the SouthTRAC (ST) campaign, and mean precisions were calculated during the flights

We have rewritten this part of the sentence according to the proposal.

- L120: prior --> prior to

Done.

- L122: was of --> was

Done.

- L123: post-flight corrected --> corrected post-flight

Done.

**Section 3**

- L125: The occurrence of chlorine activation also depends on factors other than temperature (e.g., humidity, the availability of suitable aerosol particles) and has been observed outside the polar regions, so it would be better to say "tends to occur" rather than "occurs" here.

We have taken the suggestion from the reviewer.

- L129: conclusion --> conclusions

Done.

- L134-135: Modern meteorological reanalyses have fairly high resolution these days [e.g., Fujiwara et al., ACP, 2017]; although they still do not resolve very small-scale features, it is not entirely fair to characterize them as having "rather coarse resolution".

We agree that the wording "rather coarse resolution" is not appropriate. Therefore, we have rephrased the sentences regarding PV for air mass classification:

"[...] PV is a model-derived quantity. Although the underlying meteorological reanalysis have a fairly high resolution these days (e.g., Hersbach et al., 2020), small scale features like vortex filaments with different chemical compositions may not be well resolved. [...]"

- L138-145: This part of the paragraph is poorly written, repetitive, and hard to follow. It should be reorganized to improve the flow.
  - The sentence "It can be measured ... atmosphere." is out of place, as it comes in between two sentences that say essentially the same thing. It should be moved and the other two sentences combined to reduce repetition.
  - Not only is the statement "the isolation inside the vortex benefits mixing on isentropic surfaces and therefore a small variability on isentropes (variability of about 6 ppb)" grammatically incorrect, but also it makes no sense. I'm not sure what "benefits mixing" means? Perhaps "inhibits" was meant? In any case, this sentence needs to be rewritten.
  - The sentence "The low mixing ratios inside the vortex ... N2O" again repeats the same information already stated twice above. (Also, descend --> descent)

**Regarding these three comments, we have rewritten this part of the paragraph.**

The sentence "It can be measured ... atmosphere." was moved to the top, followed by a small description of the polar vortex. Subsequently,  $N_2O$  inside and outside the vortex is described. The sentence about the variability of  $N_2O$  inside the vortex is rewritten for more clarity:

"[...] A tracer like N2O can be measured in situ with a sufficiently high time resolution to reveal small structures in the atmosphere. During the dark polar winter, stratospheric temperatures are below those of the mid-latitude stratosphere. The associated pressure gradient between the pole and mid-latitudes, as well as the Earth's rotation, leads to enhanced circumpolar winds, also known as polar night jet or polar vortex (e.g., Schoeberl and Hartmann, 1991). Furthermore, the decreasing polar temperatures lead to a subsidence of polar air, also known as diabatic decent (e.g., Schoeberl and Hartmann, 1991; Bauer et al., 1994). A tracer like N2O exhibits a horizontal gradient across the vortex edge in the stratosphere with lower mixing ratios inside the vortex and higher mixing ratios outside the vortex. In addition, N2O has a small variability inside the vortex on constant isentropic surfaces (variability of about 6 ppb (Greenblatt et al., 2002a)). This is an indication of well mixed air inside the polar vortex due to the long isolation in polar winter. [...]"

• the mid-latitudes vertical gradient is weak and more variable --> the vertical gradient in mid-latitude N2O is weak and more variable

We have taken the suggestion from the reviewer.

L146-147: "Towards tropopause altitudes, the N2O profiles of vortex and midlatitudes merge and differentiation becomes difficult." Near the tropopause the vortex proper – and the transport barrier it represents – is no longer defined; the region in which chemical processing still takes place but confinement is weak (below ~350–380 K in SH, ~400–450 K in the NH, depending on the year) is often termed the "subvortex" [see Santee et al., JGR 2011, and numerous references therein]. So it is not appropriate to refer to "vortex profiles" in this region.

We have rewritten this sentence for more clarity, that the vortex barrier vanishes towards the tropopause and no vortex profile can be defined.

"[...] Towards tropopause altitudes, the transport barrier of the polar vortex disappears, and a classification is not possible. [...]"

- L149: at best --> ideally

Done.

- L152-153: "Stratospheric transport and mixing is related to the isentropic surfaces whereas mixing at the extratropical tropopause affects the lowest 25 K relative to the local tropopause." This sentence needs work.
  - The wording "is related to" is not clear. I assume that the first half of this sentence is referring to the fact that adiabatic flow in the stratosphere largely occurs along isentropic surfaces, but this should be clarified.
  - is --> are; also, add a comma after "surfaces"
  - References are needed, especially for the point that mixing affects the lowest 25 K above the tropopause (see below).

**We have reworded the sentence into two sentence and added necessary information on stratospheric transport and mixing. References were added for mixing at the extratropical tropopause.**

"[...] The composition of the lowermost stratosphere is affected by the diabatic decent inside and outside the polar vortex and quasi-isentropic mixing with air from lower latitudes. In addition, mixing at the extratropical tropopause affects the lowest 20-25 K above the local tropopause (Hoor et al., 2004, 2005) [...]"

- flights, which --> flights that; contact to --> contact with

This sentence was rewritten for more clarity.

"[...] The vortex reference profile (see Fig. 2) was generated from all flights that are assumed to contain measurements within vortex air. [...]"

- L155-157: "Data from these flights were pre-filtered by taking only the measurements polewards of 60°S equivalent latitude and 20 K above the local tropopause." There are several issues with this sentence.
  - The concept of equivalent latitude should be defined and a suitable reference for it provided (e.g., Butchart & Remsberg [JAS, 1986]).

We have included the suggested reference. However, we will refrain from defining the equivalent latitude at this point in the text, given that it is a well-known concept in stratospheric research.

 Presumably the EqL is being calculated based on PV from a meteorological reanalysis, but this information needs to be provided. The reanalysis being used in a study is typically identified in the "Data and Methods" section – here that section is entitled "The SouthTRAC Campaign", but the meteorological data is also an important component of this study and probably merits its own subsection. I have more to say on this point later.

We support this statement. It is not well captured, where the meteorological data come from, although they are important for the further analysis in this manuscript. Therefore, we include a new subsection "Meteorological data" in the section "The SouthTRAC campaign" with the necessary information.

**"[...]**

**2.1 Meteorological data**

HALO was equipped with a wide range of in situ and remote-sensing instruments. In addition to the scientific instruments installed for the measurement campaign, the Basic Halo Measurements and Sensor System (BAHAMAS) is part of HALO. BAHAMAS is installed permanently and provides meteorological and aircraft parameters along the flight trajectory (DLR, 2020). The local tropopause information along the flight tracks of HALO was created using the Chemical Lagrangian Model of the Stratosphere (CLaMS) (e.g., Grooß et al., 2014). The underlying meteorological data are taken from ECMWF ERA-5 data (Hersbach et al., 2020). In this work, the potential vorticity (PV) based dynamic tropopause is used (e.g., Gettelman et al.,2011), taking the 2 PVU (potential vorticity unit) for the dynamical tropopause. Since the PV tropopause is not physically meaningful in the tropics, potential temperature level of 380 K was taken as tropopause if the 2 PVU level lies above. [...]"

 Similar to the above point, it needs to be made clear how the local tropopause is being determined. The Fig. 2 and 8 captions mention the "WMO tropopause", but more detailed information should be provided in the main text. In addition, it seems that the results may be highly sensitive to the exact definition of the tropopause used, and some discussion of the associated uncertainty in the results would be appropriate. *Information about the local tropopause can be taken from the newly introduced subsection 2.1 (see point before).*

• The Antarctic vortex frequently extends to EqLs lower than 60°S EqL. Have the authors made sure that imposing the 60°S EqL cutoff has not eliminated vortex profiles in 2019?

In fact, filtering to 60°S equivalent latitude can remove some measurements that may be counted as vortex. This pre-filtering is necessary for the iterative filtering procedure to find the lower envelope. The loss of these few data points should not affect the final profile. We checked graphically if the cut at 60° leads to a substantial loss of data points at the lower envelope. This was not the case.

• The previous paragraph states that mixing affects the region within 25 K of the tropopause, so it is not clear why the cutoff here was chosen to be 20 K.

The previous paragraph has been changed with updated information, that mixing affects the region within 20 -25 K above the local tropopause (e.g. Hoor et al., 2004, 2005).

In the Southern Hemisphere, the extratropical transition layer seems to be shallower (Hegglin et al., 2009). We therefore chose the lower value of 20 K for the cutoff.

- L157-158: This sentence ("The lowest levels ...") is repetitive, unnecessary, and out of place – it should be combined with the similar sentence in L152-153.

This sentence has been removed and the necessary information and reference has been provided earlier in the text as suggested.

Fig. 2 caption: to the local --> from the local; criterion on --> criterion of; the the --> the

Done; Done; Done.

- L158-164: One general comment is that the creation of reference profiles is a key point on which much of the following analysis rests, and its description should not be relegated to the separate Supporting Information, which many readers will not make the effort to obtain. I would prefer to see it in the main text, but it should at least be moved to an Appendix included at the end of the main paper file (unless ACP no longer allows such Appendices). Another general comment is that the main text, SI, and figure captions together fail to clearly describe the method, as specified in more detail in the points below.
  - L160: At this point, the reader has no idea what is meant by the term "vortex profile function." Also, Werner (2006) is a PhD thesis for which no download information is given, and thus it is not a suitable reference.

We have changed the reference to a more suitable one: Werner et al., 2010.

 L161: Elsewhere the convention "60°S EqL" is used, so for consistency "-40° and -60° EqL" here should be "40° and 60°S EqL".

**We changed it accordingly.**

S1, L2-9: The first two paragraphs of the SI are fully redundant with the discussion in the main text. Apart from this repeated material, the description of the procedure in the SI is only one paragraph long, so again I would argue that it would be better to edit, merge, and rearrange the discussions in S1 L10-28 and L158-164 to produce a single compact paragraph in the body of the paper.

We removed the first two paragraphs, which are indeed repeated material. The description of the filter procedure was moved to the main manuscript. We added an appendix A describing the filter procedure:

**"[...]**

**Appendix A. Filter procedure for vortex and mid-latitude profiles**

A filter procedure was used to derive the lower envelope for the vortex profile and the upper envelope for the mid-latitude profile. Figure S1 in the supporting information displays the procedure for the task using either  $\Delta \Theta$  or  $\Theta$  as the vertical coordinate. The process is initialized by binning the N2O measurements into intervals of e.g.,  $\Delta \Theta$ . The bin size must be adjusted to the number of measurements available for the vortex and mid-latitude profile to make the filter procedure work properly. For every bin, the mean value, standard deviation, and relative standard deviation are calculated. This is necessary as the condition for the filter needs a binned profile to begin with. While the maximum relative standard deviation is larger than the preset outlier limit, the measurements that are not flagged as outliers are binned in intervals of  $\Delta \Theta$  (this is done twice in the first iteration step, since the binned profile is already needed for the initialization and no outliers are set for the beginning of the filtering process). Every bin is checked whether the relative standard deviation is larger than the outlier limit. In this case, all measurements of  $N_2O$  which are higher (or lower, if the upper envelope is requested) than the mean of the respective bin are flagged as outliers and removed from further iterations. The iteration process stops when the maximum relative standard deviation is below the preset outlier limit. For the vortex profile, bin size was set to 5 Kelvin. The variability of N2O on a constant Θ surface inside the vortex is about 6 ppb (Greenblatt et al., 2002). For the range of  $N_2O$  mixing ratios in this work, this corresponds roughly to about 3 % and was thus set as the outlier limit. Four iterations were done to get the lower envelope (grey samples in Figure S 2 a) and b) in the supporting information). For the mid-latitude profile, the bin size was set to 2 Kelvin. Strahan et al. (1999) showed that the variability of N2O in the Southern

Hemisphere lower stratosphere of the mid-latitudes is approximately between 5 and 15 % (see plate 6 therein). Therefore, a value of about 10 % is set for the outlier limit, which leads to two iteration steps for the remaining measurements. For the profiles only those measurements are used which are not marked as outlier [...]"

In the main text, we replaced the sentence "A more detailed description of the creation of reference profiles can be found in the supporting information." With the following information from the supporting information:

"[...] As an intermediate step to the final profiles the measurements of the lower and upper envelope are binned in 5 Kelvin intervals of  $\Theta$  or  $\Delta\Theta$  (see Figure S 2).

Mean values of the binned profiles are then used to generate a polynomial fit function for the vortex profile and the mid-latitude profile (Figure S 3). [...]"

• Fig. S1: I did not find the flowchart to be particularly helpful, so it could remain in the SI.

Flowchart remains unchanged.

• Fig. S2-S7: It would be much easier on the reader if all of the vortex figures (S2, S4, S6) were combined into one 3-panel figure, and the same for the midlatitude figures (S3, S5, S7). In fact, it would probably work to combine them all into one 2-row, 3-column figure. o S1, L13: is calculated --> are calculated

The figures are rearranged. We combined Figure S2 to S5 into one 2-row, 2column figure. Therefore, it is more compact, and the reader gets a good overview. Figure S6 and S7 are combined to one 2-column figure. We have refrained from a 2-row, 3-columns figure, as the individual figures become much smaller, and the reader may have difficulties looking at them.

• S1, L14 and L21: pre-setted --> preset

Done.

• S1, L14: measurements, which --> measurements that

Done.

• S1, L17: Is this the case --> In this case

Done.

• S1, L21: shows --> show

Done.

• S1, L23: latiudes --> latitude

Done.

S1, L22-23: More discussion is needed on the 3% and 10% "outlier limits" for vortex and mid-latitude profiles, respectively. How were these preset outlier limits and Δθ bin sizes determined? What factors drove the differences between the values of these quantities for the vortex and mid-latitude profiles? How sensitive are the results to these choices?

Greenblatt et al. (2002) shows that the variability of N2O inside the vortex is about 6 ppb. By looking at the lowest N2O mixing ratios within the potential temperature range (see Figure S2), 6 ppb refers to roughly up to 3% variability. For the mid-latitudes, Strahan et al. (1999) investigated a variability in the lowermost stratosphere ranging from around 5% to 15%. We chose the mean of 10% as representative of the variability of the midlatitudes. The size of  $\Delta \theta$  was chosen accordingly the amount of measurements to make the filter procedure work. We have added this information at appropriate places in the text.

**Regarding $\Delta \theta$ :**

"[...] The bin sizes must be adjusted to the number of measurements available for the vortex and mid-latitude profile to make the filter procedure work properly. [...]"

**Regarding the 3% outlier limit:**

"[...] For the vortex profile, bin size was set to 5 Kelvin. The variability of  $N_2O$  on a constant  $\Theta$  surface inside the vortex is about 6 ppb (Greenblatt et al., 2002). For the range of  $N_2O$  mixing ratios in this work, this corresponds roughly to about 3 % and was thus set as the outlier limit. [...]"

**Regarding the 10% outlier limit:**

"[...] For the mid-latitude profile, the bin size was set to 2 Kelvin. Strahan et al. (1999) shows that the variability of  $N_2O$  in the Southern Hemisphere lower stratosphere of the middle latitudes is approximately between 5 and 15% (see plate 6 therein). Therefore, a value of about 10% is set for the outlier limit, which leads to two iteration steps for the remaining measurements. [...]"

• Fig. S2: What causes the "staircase" pattern between the points discarded in iterations 1 and 2? Also, left over --> leftover, but "remaining" would probably be a better word here (also in the Fig. S3 caption)

The "staircase" pattern is a result of the filtering procedure and is always as large as the bin size. For every bin, all measurements which are higher (for vortex profile) or lower (for mid-latitude profile) than the  $N_20$  mean value of the bin are flagged, in case the relative standard deviation of the bin is larger than the outlier limit.

- L167: decent --> descent

Done.

L168-169: Greenblatt et al. [2002b] quantifies descent inside the Arctic vortex and thus has only limited applicability for SouthTRAC, since the characteristics and seasonal evolution of descent are somewhat different in the two hemispheres, as discussed by Manney et al. [JAS, 1994], which would be a more appropriate reference. Manney et al. note that parcels in the SH lower stratosphere generally cease to descend in mid-October, so descent might still be ongoing in September, contrary to the statement made here. Also, a further --> further.

The referee is correct. Greenblatt et al., 2002b is not a suitable reference for the diabatic descending of the Southern Hemisphere since it differs in this aspect from the Northern Hemisphere. We therefore rearranged this paragraph and used the suggested reference instead.

"[...] In the lower stratosphere of the Southern Hemisphere, the descending stops around mid-October (Manney et al., 1994). However, N2O data of the SouthTRAC flights did not reveal strong diabatic descent during the time of the campaign (below  $\theta$ = 400 K). Therefore, only one reference vortex profile was generated for the campaign. [...]"

L173-174: I am confused about the "prescribed cutoff value" for vortex profiles and "associated variability" for mid-latitude profiles mentioned here and specified (20 ppb and 15 ppb, respectively) in the Fig. 2 caption. Where did these values come from? Why is one characterized as a "prescribed cutoff" and the other as an "associated variability"? Do these values have anything to do with the outlier limits discussed above? In addition, it seems that all points falling "below" (i.e., to the left of) the grey "cutoff" curve are deemed to be vortex points, even when they are "above" (i.e., to the right of) the black N2Ovor curve, but that is not what is said in "if the mixing ratio is below the respective N2Ovor with a prescribed cutoff value, then it is assigned to the vortex" (and similarly for the mid-latitude case).

The prescribed cutoff was adopted from Greenblatt et al. (2002). Ivanova et al. (2008) also used the value 20 ppb by Greenblatt et al. (2002) for measurements in the Antarctic, thus we assumed a usage of the value for both Arctic and Antarctic. We derived the associated variability from the 10% variability of the mid-latitude measurements, leading to a  $\pm$ 15 ppb variability. In the text, we include theses information into the last paragraph of section 3.1:

"[...] For the prescribed cutoff value, the value of 20 ppb proposed by Greenblatt et al. (2002) was used. The associated variability of the mid-latitude profile was set to 15 ppb, as the variability  $N_2O$  in the mid-latitudes is roughly 10% (see supporting information and reference therein). [...]"

For more clarity regarding the classification procedure, we changed some wording in the following same paragraph:

"[...] if the mixing ratio is below the respective  $N_2O_{vor}$  added by a prescribed cutoff value, then it is assigned to the vortex. Otherwise, if the mixing ratio is above the respective  $N_2O_{mid}$  minus an associated variability, then it is assigned to mid-latitudes. Mixing ratios above the respective  $N_2O_{vor}$  added by a prescribed cutoff value and below the respective  $N_2O_{mid}$  minus an associated variability are assigned to the boundary region. For the mixing ratios where  $N_2O_{vor}$  added by a prescribed cutoff value and  $N_2O_{mid}$  minus an associated variability overlap, these measurements cannot be assigned to one region. [...]"

- L176-177: It is stated that measurements in the overlap region cannot be "fully assigned to one region". So what was done with them? Were they included in the analysis or discarded? This question is answered later in section 4.3, but the reader should not be left in suspense here. Also, delete the comma after "ratios", and "cannot" is one word, not two.

We added a sentence to make clear, that these measurements were not discarded. Further, we delete the comma after "ratios" and changed "can not" to "cannot".

"[...] For this reason, these measurements are assigned to both the vortex and the mid-latitudes in later analysis. [...]"

- L181: has been --> was

Done.

L183: "timed" sounds intentional, whereas I believe that the campaign fortuitously took place shortly after the SSW. This sentence is also grammatically awkward. I suggest instead: "... November; thus they occurred shortly after the minor SSW event and captured the late winter evolution ..."

We agree that "timed" is misleading and changed the sentence similar to the suggestion.

"[...] The SouthTRAC campaign flights took place from early September to early October and in the first half of November; thus they took place shortly after the minor SSW event and captured the late winter evolution of the Antarctic polar vortex. [...]"

- L187: Other flights besides the 11 September flight are omitted from Fig. 3, since the number shown does not add up to the total given in section 2.

That is correct. Figure 3 only shows the local flights for which the classification was possible. We added this in the text.

"[...] Figure 3 displays an overview of air mass classification of the local flights of the SouthTRAC campaign (classification in  $\theta$ -coordinates) [...]"

 L189: It is not quite true that "flights sampled mostly inside the vortex or vortex boundary region" during Phase I – in fact, Fig. 3 shows that few or no such measurements were taken on nearly half of those flights. Also, extensive --> extensively; add a comma after "phase"

The meaning of this sentence is that there were flights in the first phase, which measured predominantly vortex and vortex boundary region. To show this better, we have divided the sentence and added this information:

"[...] Vortex and boundary region were sampled in both phases of the campaign. The first phase contains some flights that have predominantly sampled vortex or vortex boundary region (e.g., flight ST15 on 29 September or flight ST16 on 30 September). [...]"

- L191: more than half of the ... air 54% --> more than half (54%) of the ... air

Done.

**Section 4**

- L194-197: These sentences about EESC seem out of place here. Perhaps they would fit better at the beginning of subsection 4.2, or in the Introduction. Also, matter --> manner

We agree with the Referee and moved the sentences about EESC to the introduction. In addition, we reworded the first sentence at the advice of the Referee #1.

"[...] Equivalent effective stratospheric chlorine (EESC) is a simple metric that sums the effect of ozone depleting substances (ODS) as an equivalent amount of inorganic chlorine in the stratosphere (Newman et al., 2007; Daniel et al., 1995). Changes to the EESC are mainly due to Cly, as Bry makes up a smaller fraction (Strahan et al., 2014) [...]"

- L197-199: These two sentences are redundant with the paragraph at the start of section 4.1. It would be better to merge / edit to avoid such repetition from one paragraph to the next.

As this information are redundant with the first paragraph of section 4.1, we removed these sentences and part of one sentence was included in the first paragraph of section 4.1.

"[...] This could lead to a rather coarse resolution where fine structures like filaments and small-scale dynamical perturbations are sometimes not well resolved. [...]"

 L205-206: Measuring CFC-12 on both the ECD and MS channel of the instrument allows to up-sample the measurements of the organic source gases by using the higher resolved --> Measuring CFC-12 in both the ECD and MS channels of the instrument allows the measurements of the organic source gases to be up-sampled by using the better-resolved Done.

- L206: CFC-12 on --> CFC-12 in; throughout the manuscript (including in figure captions), "measurements on" should be changed to "measurements in"

We changed "CFC-12 on" to "CFC-12 in"; There is no "measurements on" in the text to change.

- L207: but also a better precision than on the MS channel --> but they also have better precision than data from the MS channel

Done.

- L209: add a comma after "ratios"

Done.

- L209: It might be good to add "linear or polynomial" in front of "fit function".

Done.

 S2, L30-31: For up-samling the GhOST-MS measurements, pre-required are good correlations between CFC-12 and the other --> Up-sampling the GhOST-MS measurements requires good correlations between CFC-12 and the other

Done.

 Fig. S8: It should be made clear in the caption that all of the data shown are from the GhOST-MS channel. Also, the small font makes the axis labels on these panels very hard to read.

We filled in the information, that the data are from the GhOST-MS channel. In addition, we have created two figures from this one figure so that the plots are easier to look at.

 Fig. 5: In red, original data, whereas is black, measurements were up-sampled using CFC-12 measurements of the ECD channel --> Original data shown in red, measurements up-sampled using CFC-12 from the ECD channel in black

Done.

- L216-217: It is not that "the original data were not well captured"; rather, the original lower-resolution data did not capture well the abrupt transitions between regimes.

We rewritten this sentence for more clarity.

"[...] Especially with the sharp gradients, e.g. at 04:10 UTC and at 05:50 UTC in Fig 5, the original lower-resolution data did not capture well the transitions between the regimes, compared to the up-sampled [...]"

L224-225: I find this wording unclear. It would be better to rewrite as: "Organic chlorine (CCly) ... up-sampled GhOST-MS measurements. Thus, Cly can be calculated from Eq. 1 if the mixing ratios of the major chlorine-containing substances at the stratospheric entry point (Cltotal) are known. Air enters the stratosphere predominantly ...".

We rewritten this part of the section accordingly.

- L227: can not --> cannot

Done.

- L228: times in the stratosphere since they entered the stratosphere --> times since they entered the stratosphere

Done.

- L236: previous --> previously

Done.

- L238: ratio, which --> ratio that; degradation and thus --> degradation, which thus

Done; Done.

- L239: It would be appropriate to add "estimated" in front of "entry mixing ratios"

We have inserted "estimated" as suggested.

 L241: For the case no --> For the case where no; also, to make the distinction between the so-called "semi-direct" and "indirect" methods more clear, it would help to add "indirectly" between "calculated" and "based on".

Done; included "indirectly" between "calculated" and "based on".

- L244: add a comma after "trends"

Done.

- L246: where --> when

Done.

- L247: delete the commas after both "showed" and "tracers"

Done.

- L254: respective entry --> respective estimated entry

Done.

- L256-257: rations with --> ratios by

Done.

 L259: It is difficult for the reader to keep track of exactly what is meant by the "direct", "semi-direct", and "indirect" methods. To help clarify this sentence, it would be good to add "based on previous balloon observations transferred to 2019" after "indirectly determined correlations" (assuming that I am interpreting the approaches correctly).

We have attached the proposed additional information to this sentence.

 L259: indirectly determined values are not only based on observations which have been performed about 10 years earlier but also are from the --> indirectly determined values are based on observations that were not only performed about 10 years earlier but that were also from the

Done.

- L261: This wording is unclear. Replace "They" with "The balloon-based correlations".

Done.

 Fig. 6: I assume that the SouthTRAC (black) points in this figure show the up-sampled (not raw) GhOST-MS measurements for CFC-11, etc., but this should be stated explicitly. Also, I may not be interpreting this figure correctly. Do the red symbols represent the correlations between balloon CFC-11 (for example) and balloon CFC-12 data, or between balloon CFC-11 (for example) and GhOST-ECD CFC-12 data? Please clarify. In addition, the term "retrended" is used only in the figure caption and legend, not in the main text. Although it is somewhat ambiguous, it is fine to use this term as long as it is defined in the body of the paper.

Figure 6 displays the raw GhOST-MS measurements of CFC-12, CFC-11, CH3Cl, and HCFC-142b in black. The red symbols represent the balloon observations in 2009 and 2011 transferred to the time of the SouthTRAC campaign in 2019, thus not using GhOST-ECD CFC-12 data. We want to show that despite the different hemispheres and the time interval of 10 year, the correlations of the long-lived chlorinated substance are comparable. These correlations can be used to determine organic chlorine (CCly) from CFC-12 alone. For more clarification, we extended the description in the caption and avoid the term "re-trended":

"[...] Correlation between CFC-12 and CFC-11, CFC-12 and CH3Cl, and between CFC-12 and HCFC-142b. In black the raw measurements by GhOST-MS, in red the balloon observations scaled to the time of the SouthTRAC campaign using mean arrival time. [...]"

**We also extended the sentence in the text, where figure 6 is mentioned:**

"[...] Fig. 6 displays scaled correlations from the balloon observations (red) and correlations form the SouthTRAC data (black) of three long-lived substances against CFC-12. [...]"

L262-265: I am not following the logic here. I understand that a subset of the components of CCly "retrended" from earlier balloon data match well correlations with CFC-12 measured by SouthTRAC (Fig. 6). But why does that necessarily mean that CCly based on correlations with CFC-12 can be used as a good proxy for Cly? I feel that a step is missing. And why is it relevant to mention here again that Cltotal can be derived to calculate Cly – that information is not being used for Eq. 2 and the indirect method, is it? In general, I feel that the relationship between Eq. 1 (at the heart of the semi-direct approach) and Eq. 2 (the basis of the indirect method) is not clearly explained. Please clarify this discussion.

We agree that there is a missing step in between the explanation on how to get to equation 2. Thus, we expanded the description, why we need  $Cl_{total}$  again for the indirect  $Cl_y$  and how we get  $Cl_{total}$ . The following changes are done in the manuscript:

"[...] The balloon-based correlations correspond well to the correlations measured during the SouthTRAC campaign. Thus, the balloon-based correlations can be used to determine CCly from CFC-12 alone. As already mentioned earlier, Cltotal is also needed for the calculation of Cly. For this, the mean age values derived for the balloon measurements are used and Cltotal is calculated for the conditions during SouthTRAC hereafter. Cly is then derived as the difference between Cltotal and CCly. A correlation function for the conditions during Antarctic late winter 2019 has then been derived for the indirect calculation of Cly as a function of CFC-12 mixing ratios (Eq. 2). [...]"

 L266-267: Since the previous sentences have been discussing balloon-based correlation functions, this sentence about the GhOST-ECD CFC-12 data seems out of place and confusing to me – maybe it belongs at the end of the following paragraph rather than here, or perhaps I have misunderstood the role of those data in the foregoing discussion, as noted above.

With the before mentioned method, we generated the correlation function for Cly with the reference substance CFC-12 based on the balloon data. We now need CFC-12 measurements during the SouthTRAC campaign to calculate Cly. This was done using CFC-12 from the GhOST-ECD channel. We slightly changed the sentence to make clear, that CFC-12 of GhOST-ECD was used for indirect Cly.

"[...] In the following, CFC-12 from the GhOST-ECD channel is used for the indirect determination of inorganic chlorine. [...]"

- Table 2: To enhance clarity, it would be good to add "indirectly" in front of "derive".

Done.

 Fig. 7: Are the indirect results shown here from the "retrended" balloon data or from SouthTRAC? I assume the former but this should be stated explicitly. And why not show comparisons for both data sets, since you also provide SouthTRAC coefficients in Table 2? Also, in the caption: Indirectly and directly determined ... (green and back) and --> Indirectly (green) and directly (black) determined ... and

*This figure shows results of the indirect method from the "re-trended" balloon data. We clarify this in the caption of the figure.*

"[...] Figure 7. Indirectly (green) determined Cly based on balloon observations in 2009 and 2011 and semi-directly (black) determined Cly as a function of age of air. In red, the absolute difference between these methods. [...]"

- L275: delete the comma after "Hemisphere"

Done.

- L276: Cly, where --> Cly in cases where

Done.

- L277-278: "Since it was possible during SouthTRAC to measure the organic source gases, the Cly from the direct measurements was used for further evaluation." In fact, while reading this section I wondered why the authors bothered to pursue the indirect approach when they actually have direct measurements of CCly. The discussion of the indirect Cly calculation is particularly confusing, and it is not at all clear at this point what value it brings. So it would be helpful to add a pointer to section 4.4, where the indirect method is needed for the comparisons with Cly in the Arctic, to better justify the inclusion of this discussion here. In addition, I think that, rather than "Cly from the direct measurements", it would be more appropriate to say "Cly determined semi-directly from the measurements".

You are right. For SouthTRAC the indirect method is not needed. However, the measurements during SouthTRAC offer to compare the semi-direct method and the indirect method to show that the indirect method leads to comparable results, which is crucial later. We added a pointer to section 4.4 to justify the comparison.

We also changed " $Cl_y$  from the direct measurements" to " $Cl_y$  determined semi-directly from the measurements".

"[...] However, the good comparability of the two methods offers the possibility to compare  $Cl_y$  from different measurement campaigns, which differ regarding the number of measured chlorinated substances (see section 4.4). [...]"

L279: Another thing that is not clear to me is why the fit coefficients for N2O are given in Table 2 if they are not being used here at all. On the other hand, several previous studies have used N2O to derive Cly, so I think it might be useful to expand the discussion of the N2O correlations. An obvious question that arises is: How well does the Cly derived from fits with N2O agree with that based on the correlation with CFC-12, for both the balloon data and SouthTRAC? A figure similar to Fig. 7 could be added for N2O.

We included the fit coefficients for  $N_2O$  as  $N_2O$  was used to calculate  $Cl_y$  in several publications before. Thus, if one wants to calculate  $Cl_y$  for this time using  $N_2O$  measurements, the given correlation coefficients can be used. As they are already mentioned in the paragraph before (before equation 2) we expand the discussion of the  $N_2O$  correlation at this point by the following (including a similar figure to Fig 7, as suggested by the referee):

"[...]  $N_2O$  shows a compact correlation to long-lived chlorinated substances and has been used in many publications for the determination of  $Cl_y$  (e.g., Schauffler et al., 2003; Strahan et al., 2014; Strahan and Douglass, 2018). Using CFC-12 from the GhOST-ECD channel and  $N_2O$  from the UMAQS instrument, we obtain comparable values for  $Cl_y$  (see figure S 7 in the supporting information). In the following, CFC-12 from the GhOST-ECD channel is used for the indirect determination of inorganic chlorine. [...]"

We have refrained from a figure for the coefficients from the SouthTRAC data, since in corresponding coefficients in the table are derived from the same semi-directly determined Cly values.

- L284: only measurements were taken, which are polewards of 40° equivalent latitude --> only measurements polewards of 40° equivalent latitude are used

Done.

- L287: an air mass --> air mass

Done.

 L289-290: It seems to me that it might be better to exclude the measurements in the overlap region from the analysis rather than "double count" them. How many measurements fall into this category, and how would omitting them change the results?

When using the classification in  $\Theta$ -coordinates, there are 263 Cly measurements in the overlap region for the SouthTRAC campaign. The number of measurements is even larger with 592 measurements using  $\Delta\Theta$  as the vertical coordinate. Omitting these measurements would affect the variability of the profiles in the range of potential temperature, where the vortex cutoff and variability of the mid-latitude profile overlaps. Since, for example, the cutoff of 20 ppb is no longer considered, a narrow

range of mixing ratios is considered for the vortex profile, and it shifts toward higher Cly values. The measurements in the overlap area can, however, originate from the vortex or the mid-latitudes. Thus, we would like to include them for both profiles.

- Fig. 8: I have a number of comments / questions about this figure.
  - It is stated that data are averaged over -40° to -90° is the filtering of measurements obtained equatorward of 60° equivalent latitude (mentioned in section 3.1) only applied in calculating the vortex reference profile? Please clarify (here and in section 3.1).

Indeed, the filtering of measurements poleward of 60° equiv. latitude is only applied to generate the vortex reference profile as well as the filtering between 60°S and 40° S equivalent latitude for the mid-latitude profile. The consideration of measurements south of 40° S equiv. latitude serves on the one hand to ensure that the classification was only made in this range and to filter out the data of the transfer flights to the Northern Hemisphere. The information given in the text, both in section 3.1 and in section 4.3, should already make clear that only for the vortex profile, there is a limited number of flights and the cut at 60°S, whereas in the analysis in 4.3 all flights are used and from this all measurements south of 40°S are taken.

• The colors denoting the different regions have been changed. Previous figures used a consistent set of colors for these classifications, and it would be easier for readers if that same color scheme was used in all figures for which those classifications are relevant.

The colors have been changed to match the colors of the previous figures in terms of classifications.

• The lack of tick marks on the top and right-hand axes is annoying and makes it difficult to judge the values given in the text. In addition, the tick marks (especially the minor ones) that are present on the bottom and left-hand axes are too small to be easily seen.

Tick marks were added to the top and right-hand axes. In addition, ticks were made larger, both major and minor ticks.

• Why does Cltotal vary with altitude?

The mean value of Cltotal between around 280 and 400 K potential temperature in figure 8 is 3074 ppt with a maximum of 3097 ppt and a minimum of 3054 ppt. Thus, the range of Cltotal is about 43 ppt. Furthermore, the relative standard deviation of all bins is around 0.41%. The standard deviation of each individual bin is of the same order of magnitude. The variability of Cltotal is therefore very small. Nevertheless, a very small increase in Cltotal with altitude in the stratosphere can be expected, considering the temporal delay (the age of air) with which Cltotal propagates into the stratosphere (e.g., Engel et al., 2002). • Elsewhere total chlorine was written "Cltotal", and that should be the case here too.

*Cl*tot was changed to *Cl*total in the figure legend.

• For consistency with the text, "-90° to -40°" should be "40°–90°S".

The figure title was changed to:

"[...] Chlorine partitioning between 40 and 90°S eq. lat [°] [...]"

• "mean averaged" is redundant

We deleted "mean".

- L291: the measurements --> the SouthTRAC measurements

Done.

- L292: A reference is needed for the AGAGE results.

For not misleading at this point, we have rewritten this sentence, as this is not at statement based on AGAGE result but based on the chlorine input values which are based on the AGAGE time trends.

"[...] The SouthTRAC measurements of the long-lived chlorinated substances are consistent with the total chlorine of these substances, based on time trends from the AGAGE Network. [...]"

- L294-295: add commas after "330 and 390 K" and "390 and 400 K"

Done.

- L299-300: "at this altitude" – which altitude? Where Cly is maximum? Please clarify.

We changed "at this altitude" to "at highest measured potential temperatures"

- L301: Accompanying the minor SSW --> As a consequence of the minor SSW

Done.

L301-302: It is not clear that the 16.4 million km2 value quoted here refers to the maximum daily ozone hole area. Moreover, it is not true that the 2019 hole was "the smallest since its discovery" – other holes in the mid-1980s had smaller maximum daily area values. In any case, rather than quoting this value from the Ozone Watch web site, a better approach would be to reference the Wargan et al. [JGR, 2020] paper (already cited elsewhere in this manuscript); their Fig. 1d puts the area of the

2019 hole into climatological perspective. Perhaps more importantly, it is not clear what the point of these sentences is. Are the authors trying to imply that Cly levels in the 2019 vortex played a role in the weak ozone hole that year? Although Cly abundances inside the vortex do vary from year to year as discussed previously by Strahan et al. [JGR, 2014], variations in lower stratospheric temperatures are the primary driver of variations in the strength of polar ozone depletion.

We took Wargan et al. 2020 as the reference for the size of the Antarctic ozone hole in 2019. In addition, we have rewritten the sentences. We do not want to create a link between the amount of  $Cl_y$  and the size of the ozone hole. Instead, we wanted to note that the minor SSW event led to an early chlorine deactivation.

"[...] Thus, in late winter and early spring at this altitude about half of the recorded chlorine is found in inorganic form. Despite this amount of inorganic chlorine in the lower stratosphere, the total polar ozone column was higher than usual in September 2019. As a result of the minor SSW event, chlorine deactivation began earlier in 2019 and the ozone hole was about  $10 \times 10^6$  km2 in size, thus only 20 % of that in 2018 mid-September (Wargan et al., 2020). [...]"

L303: The title of this subsection suggests that the SouthTRAC and PGS comparison focuses on the polar region, but Fig. 10 and associated discussion includes midlatitudes as well. It may be true that comparison of Cly in the Antarctic and Arctic polar vortices has not been done previously, as the authors assert, but it is not the case that no such comparisons have been performed in the midlatitudes. In fact, total column Cly (or, rather, HCl+ClONO2) in the NH and SH mid-latitudes (Jungfraujoch and Lauder, respectively) and the trends therein are compared in the Ozone Assessment (e.g., Fig. 1-13 of WMO 2018). It would be good to place their findings into the context of these (and possibly other) midlatitude results.

**We included mid-latitudes comparisons from the Ozone Assessment 2018 to section 4.4. of our manuscript.**

"[...] In addition, Mahieu et al. (2014) reported long-term total column data for HCl and ClONO2 (representing Cly) in the stratosphere, at Jungfraujoch (46.5°N) and at Lauder(45°S), though the end of 2016. A negative trend of Cly is observed at both stations but with a non-significant trend for the Jungfraujoch data over the last decade and a slightly larger negative trend from the Lauder data. Furthermore, lower-stratosphere HCl from the Global Ozone Chemistry And Related trace gas Data records for the Stratosphere (GOZCARDS) shows larger decreases at southern latitudes and increases at northern mid-latitudes (Froidevaux et al., 2015). Thus, higher values of Cly in the mid-latitudes during PGS seems to be plausible. [...]"

**- L310: A separation --> The separation**

Done.

- L311: is based on the above mentioned method based on --> is based on the abovementioned method using Done.

- S3, L34: the vortex and the mid-latitude profile during PGS is needed --> the vortex and the mid-latitude profiles during PGS are needed

Done.

- S3, L35: Phase --> phase

Done.

- L312-314: As shown in section 4.2, the indirect method ... possible, proves to be comparable --> In section 4.2, the indirect method ... possible, was shown to provide results comparable to those obtained by the semi-direct method

The sentence was rewritten as suggested.

- L317: I assume that "2019" is a typo and that the same balloon data from 2009 as used for SouthTRAC were again used for PGS?

Yes indeed, this is a typo and we changed it to 2009.

- S4, L37: Correlations function --> Correlation function
- L319 and L326: Cltot --> Cltotal

Done.

- L319-320: This sentence ("The vertical coordinate of the classification was selected according to the displayed vertical coordinate.") is confusing and, if I understand it correctly, completely unnecessary, as the very next sentence makes clear that the two panels display the results as a function of  $\theta$  and  $\Delta \theta$ . Also, I am curious why both vertical coordinates are shown here but not in Fig. 8. Then in Fig. 10 only the tropopause-relevant coordinate is shown, with the argument that it allows for better comparison of Cly in the two hemispheres. It seems to me that it might have made more sense to show both views in Fig. 8, and then use only the tropopause-relevant coordinate in Fig. 9 and 10 for the reasons stated.

**We agree that this sentence is unnecessary and removed it. Regarding then use of $\Delta\theta$ as a vertical coordinate: Only when comparing the two campaigns $\Delta\theta$ is used as by using $\Delta\theta$ instead of $\theta$ the comparison of the campaigns leads to slightly different results especially in the lowest stratosphere. We would therefore leave the vertical coordinates as they are for the figures 8 to 10.**

- L322: as it was done --> as was done
  - Done.
- L323-325: Several issues arise in this sentence.
  - As I noted above, information about the meteorological data on which this study depends needs to be provided much earlier in the manuscript, ideally in section 2.

As mentioned earlier, a new subsection "Meteorological data" was included in the section "The SouthTRAC campaign" with the information about the meteorological data for this study.

- I was surprised to discover that the analysis is based on NCEP reanalyses. Insufficient information is provided here to identify exactly which NCEP reanalysis is being used (NCEP-NCAR R1, NCEP-DOE R2, CFSR, or CFSv2), but that needs to be specified and the corresponding reference (i.e., published journal article) cited.
- I have concerns if either NCEP R1 or R2 have been used for this analysis. Although both are still in widespread use, these reanalyses have been shown in several studies, including some recent papers stemming from the SPARC Reanalysis Intercomparison Project (S-RIP), to be unsuitable for most stratospheric studies (as noted in the S-RIP overview paper by Fujiwara et al. [ACP, 2017]).
- What exactly is meant by "climatological" here? That is, how many years have been considered in the averages? Were the climatological means also calculated over the days covered by the respective campaigns, or are they monthly averages, or ...? Are the climatological tropopauses being calculated over 40° to 90° (this latitude range is stated in the caption, but it is not clear exactly what it is referring to)?

Regarding the three points, we excluded the comparison with climatological mean tropopause values for the SouthTRAC and PGS campaign. The NCEP/NCAR Reanalysis 1 (R1) was used with monthly means. Due to the change to PV-based tropopause and the revision of the manuscript, we no longer performed a comparison with climatological tropopause values. This information is no longer included in the revised script.

 L326-327: "... abundance of total chlorine (Cltot) was lower in the stratosphere from the time of PGS (2015/2016) to the time of SouthTRAC (2019)" – the wording of this sentence is unclear. It should be rewritten to state that the abundance of total chlorine in the stratosphere decreased between the two campaign periods. It is very difficult for the reader to precisely judge the magnitude of this decline from the figure. Is the difference in the estimated PGS and SouthTRAC values of Cltotal consistent with expectation given the known decreasing trend in stratospheric chlorine loading? This is a key point.

**We slightly changed the wording of this sentence to the following:**

"[...] Independent of the vertical coordinate, total chlorine (Cltotal) in the lower stratosphere decreased from the time of PGS (2015/2016) to the time of SouthTRAC (2019) [...]"

**We further expanded the discussion about the decrease of Cltotal between the two campaigns by including results from the WMO Report 2018:**

"[...] The difference is on average about 60±9.6 ppt, thus roughly a rate of change of -16±2.6 ppt year-1. This rate is higher than the average rate of change of -12.7±0.9 ppt year-1given by Engel et al. (2018b) between 2012 and 2016, considering the longlived chlorinated substances. [...]"

- L329-330: the SouthTRAC profile increased stronger and values become more than 435 ppt larger than during PGS within the vortex at equal potential temperatures --> the SouthTRAC profile increased more steeply, reaching values more than 435 ppt larger than those during PGS at the same potential temperatures

Done.

- L330: Differences become --> Differences are

Done.

- L331-332: This sentence ("Inside the vortex ... during PGS.") is entirely redundant with the second sentence of this paragraph and should be deleted.

We excluded this sentence.

L332-333: Although close together between 20 and 25 K Δθ, the difference of Cly increased to 565 ppt at 65 K Δθ --> Although the two Cly profiles lie close together between 20 and 25 K Δθ, the differences between them increase to 565 ppt at 65 K Δθ

Done.

- Fig. 9:
  - Again, please add tick marks on the top and right-hand axes.

Done.

• 40° to 90° --> 40° to 90° equivalent latitude

Done.

Delete "and as a function of potential temperature difference to the local tropopause" in line 3 – this information is provided in the description of panel (b) below.

Done.

• "mean averaged" is redundant

Done.

• SouhTRAC --> SouthTRAC

Done.

- L335: the latitude --> the geographic latitude

Done.

- L337-338: and better allow for --> and allows for better

Done.

- L339-340: It might be interesting to know how many points contribute to each latitude- altitude bin in both hemispheres. Is there a minimum threshold for the number of points in each bin? Perhaps bins with very disparate numbers of points contributing in the NH and the SH could be marked in some manner.

The threshold for the number of measurements in each bin is five measurements. If this is not reached, the bin is not used for the evaluation. We add this information in the text as followed:

"[...] Only bins which contain at least five data points were considered in this analysis. [...]"

- L341: add a comma after "latitudes"

Done.

- L344: Highest levels of Cly reach 386 ppt more Cly during PGS --> The highest values of Cly reached are 386 ppt greater during PGS

Done.

- L345: vortex of each hemisphere is --> vortices of the two hemispheres are

Done.

 L347: I do not believe that "sporadic", which means "infrequent" or "intermittent", is the right word here, especially as no time information is conveyed in this plot.
 Perhaps "weak" or "moderate" would work, if I have understood the point the authors wish to make.

We changed "sporadic" to "weak".

- L348: it is not clear what "it" is referring to here -- Cly?

"It" is referred to the potential temperature range of up to 20 K above the local tropopause. As the sentence before begins with this range of potential temperature, we change "it" to "this range".

- L349: for both --> in both; add a comma after "hemispheres"; there is no need to introduce the acronym for ExTL since it is not used again in the manuscript

Done; Done; "(ExTL)" was removed from the sentence.

- Fig. 10: Please add tick marks on the top and right-hand axes as well as minor tick marks. Also, the color bar label should indicate that these are differences, not raw Cly values.

We added the tick marks and changed the label of the color bar.

**Section 5**

- L352: Using an extended method according to Greenblatt --> Extending the method of Greenblatt

Done.

 L353-355: It is stated that, compared to coarser-resolution PV, the method to define the vortex used here allows small structures such as filaments to be resolved. First, as noted earlier, modern meteorological reanalyses provide PV at fairly fine resolution. Second, no evidence is presented in this paper that any such filaments were actually resolved using their approach. So I am not convinced that a PV-based definition would not have been adequate.

We agree that a comparison with the PV-based classification, which has not been done in this manuscript, is not appropriate. Although it is not clear whether vortex filaments are detected, the classification based on N2O measurement does indeed reveal small-scale structures, as can be seen in Figure 5. We replaced this statement with a more suitable one:

"[...] The classification of air masses based on high-resolution in-situ measurements of  $N_2O$  offers the possibility to detect and account for even small structures and follows well the sharp gradient between the regimes. [...]"

L358-360: The authors are correct when they point out that the dynamical tropopause would be more appropriate for this kind of study than the thermal tropopause. Unfortunately, the use of the WMO tropopause raises questions about the value of this investigation. I do not really understand how the authors can say that "no dynamical PV tropopause data is yet available for the SouthTRAC campaign". In fact, high-resolution PV fields are available from multiple reanalyses. There is abundant literature discussing which PV values are most appropriate for defining the

dynamical tropopause, depending on the hemisphere and isentropic surface, etc. So it is not clear to me why the authors could not have chosen representative PV values and performed their own interpolations to the in situ measurement locations to determine the local tropopause. But even if the authors are not set up for those calculations, they could still do more to reassure readers that use of the thermal tropopause does not substantially affect their conclusions. Keber et al. used the dynamical (2 PVU) tropopause, so that information is readily available for PGS. Some simple comparisons between the WMO and PV tropopauses for the period of the PGS campaign and examination of the impact the differences between them have for Figs. 9 and 10 would be informative.

We have consulted the working group that determines the local tropopause height in potential temperature along the flight trajectory for the HALO measurement campaigns PGS and SouthTRAC. We now have the PV-based dynamic tropopause height. Accordingly, the evaluation is now carried out with these values. The sentences regarding the missing PV tropopause values will be removed from the manuscript.

- L364: CFC-12 on --> CFC-12 in

Done.

- L365: channel --> channels

Done.

- L372: add a comma after "SouthTRAC"

Done.

- L374: add a comma after "2015/2016"

Done.

 L375: "At the time of publication, it is not known that such a comparison has already been made". First, this statement is somewhat ambiguous. I think the authors mean "To our knowledge, such a comparison has not been published previously." Second, they should be a bit more precise in the language here, focusing on Cly in the polar vortices, given the discussion of mid-latitude Cly in the WMO Ozone Assessment as noted above.

We now use the suggested wording "To our knowledge". Furthermore, we have rewritten this sentence to point out, that we did not find a comparison of vortices Cly in previous publications.

"[...] To our knowledge, a comparison of  $Cl_y$  of the Arctic and Antarctic vortex has not been published was previously. [...]"

- L382: would be negative to about --> are estimated to be negative at about

Done.

 L382-383: The difference of Cly inside the respective vortex is significant and even larger than the inter annual variations reported by Strahan et al. (2014) --> The differences in Cly values inside the two vortices are substantial and even larger than the interannual variations reported by Strahan et al. (2014) for the Antarctic. (See earlier comments on a similar statement in the abstract.)

Done.

- L384: of the respective --> in each

Done.

L385: respective campaign only shows a section of the respective winter seasons.
 These sections do not match --> respective campaigns only show a portion of the winter seasons. These intervals do not correspond

This sentence was partly rewritten as suggested.

"[...] Furthermore, the respective campaigns only show a part of the winter seasons. These intervals do not correspond completely [...]".

- L386: the respective polar vortex --> the two polar vortices

Done.

- L389: add a comma after "SouthTrac"

Done.

- L391: First, citations need to be added to support this statement about the BDC being stronger during NH winter than during SH winter. Second, I think it would be more appropriate to move the conjecture about a possible cause for the interhemispheric disparity in Cly to section 4.4, where these results are discussed, rather than have it in the "Summary and Conclusions" section. In addition, I'd like to see the discussion of the discrepancy and its possible causes developed a bit more, and put into context of the midlatitude results in WMO 2018 (mentioned above). Also, I suggest some wording changes: on the northern winter hemisphere than in the southern winter hemisphere due to stronger Brewer-Dobson circulation --> during winter in the Northern Hemisphere than during winter in the Southern Hemisphere due to the stronger Brewer-Dobson circulation.

On the advice of the referee, the presumption is included in Section 4.4. The necessary source for the hemispheric difference of the Brewer-Dobson circulation was added. In addition, the discussion regarding the difference of Cly in the mid-latitudes of SH and

NH has been extended. This also includes results from the WMO report 2018, mentioned in a previous comment. It was no longer possible to change the wording because a change was made to the sentence.

"[...] Nevertheless, possible reasons for the observed differences can be derived from the hemispheric difference of the Brewer-Dobson circulation, using the age of air as a common metric for transport. Konopka et al. (2015) showed, that north of 60 °N, age of air is always younger than south of 60 °S in the same season, implying a stronger residual circulation in the Northern Hemisphere. Analysis of Haenel et al. (2015) revealed differences in age of air trends in the lowermost stratosphere of the midlatitudes of Northern and Southern Hemisphere with a positive trend in the Northern Hemisphere and a negative trend in the Southern Hemisphere. [...]".

- L393: in higher --> at higher

Done.

- L394: exhibits a larger variability as it is more effected --> exhibits larger variability as it is more affected; also, capitalize "Southern Hemisphere"

Done; Done.

- L395": side -- > hand; is less effected --> is typically less affected

Done; Done.

**Comparison of Inorganic Chlorine in the Southern Hemispheric Antarctic and Arctic lowermost stratosphere during by separate**

**Late Winter 2019aircraft measurements**

Markus Jesswein1, Heiko Bozem2, Hans-Christoph Lachnitt2, Peter Hoor2, Thomas Wagenhäuser1, Timo Keber1, Tanja Schuck1, and Andreas Engel1

1University of Frankfurt, Institute for Atmospheric and Environmental Sciences, Frankfurt, Germany 2Johannes Gutenberg University of Mainz, Institute for Atmospheric Physics, Mainz, Germany

Correspondence: Markus Jesswein (jesswein@iau.uni-frankurt.de)

Abstract. Inorganic Stratospheric inorganic chlorine ( $Cl_y$ ) is the sum of the degradation products of predominantly released from long-lived chlorinated source gases . These include and, to a small extend, very short-lived chlorinated substances.  $Cl_x$ includes the reservoir species (HCl and ClONO2) and active chlorine species (i.e.  $ClO_x$ ). The active chlorine species drive catalytic cycles that deplete ozone in the polar winter stratosphere. This work presents calculations of inorganic chlorine ( $Cl_y$ )

- 5 derived from chlorinated source gas measurements on board the High Altitude and Long Range Research Aircraft (HALO) during the Southern hemisphere Transport, Dynamic and Chemistry (SouthTRAC) campaign in austral late winter and early spring 2019. Results are compared to Cly of the Northern Hemisphere derived from measurements of the POLSTRACC-GW-LCYCLE-SALSA (PGS) campaign in the Arctic winter of 2015/2016. A scaled correlation was used for PGS data, since not all source gases were measured. Using the SouthTRAC data, Cly from a scaled correlation was compared to directly determined
- 10 Cly and agreed well. An air mass classification based on in situ N2O measurements allocates the measurements to the vortex, the vortex boundary region, and mid-latitudes. Although the Antarctic vortex was weakened in 2019 compared to previous years, Cly reached 1687±20±19 ppt at 385 K, therefore up to around 50 % of total chlorine could be found in inorganic form inside the Antarctic vortex, whereas only 15 % of total chlorine could be found in inorganic form in the southern mid-latitudes. In contrast, only 40 % of total chlorine could be found in inorganic form in the Arctic vortex during PGS and roughly 20 % in
- 15 the northern mid-latitudes. Differences inside the respective vortex reaches up to 565two vortices reach as much as 540 ppt, with more Cly in the Antarctic vortex 2019 than in the Arctic vortex 2016 (at comparable distance to the local tropopause). As far as is knownTo our knowledge, this is the first comparison of inorganic chlorine within the respective polar vortex. Antarctic and Arctic polar vortices. Based on the results of these two campaigns, the difference of differences in Cly inside the respective vortex is significant two vortices are substantial and larger than reported inter annual variations the inter-annual variations.
- 20 previously reported for the Antarctic.

**1 Introduction**

The Antarctic ozone hole is a recurring event, which was first documented by Farman et al. (1985) and has been observed anually since the 1980-iesannually since the 1980s. Polar ozone depletion is pre-dominantly predominantly driven by anthropogenic chlorine and bromine from long-lived halogenated species (Molina and Rowland, 1974; Engel et al., 2018b). (Molina and Rowland

- 25 , 1974; Engel and Rigby, 2018). The primary mechanisms for the depletion of ozone  $(O_3)$  in the polar stratosphere are the catalytic cycles with halogen-containing free radicals as chain carriers (Molina et al., 1987). Chlorine substances, which are involved in rapid ozone depletion are Cl, Cl2, ClO, and ClOOCl, and  $\frac{1}{2}$ , and can be summarized as ClOx. Additionally, hydrogen chloride (HCl) and chlorine nitrate (ClONO2) contribute to ozone depletion as they enable the production of active chlorine within through heterogeneous reactions on polar stratospheric clouds (PSC) during polar winter with low temperatures
- (e.g., Crutzen and Arnold, 1986; Molina et al., 1987) (e.g., Crutzen and Arnold, 1986; Molina et al., 1987; Solomon, 1999). They 30 are therefore called reservoir species. Chemically active chlorine  $(ClO_x)$  and the reservoir gases together form the total inorganic chlorine ( $Cl_v$ ), also called available chlorine. Equivalent effective stratospheric chlorine (EESC) is a simple metric that sums the effect of ozone depleting substances (ODS) as an equivalent amount of inorganic chlorine in the stratosphere (Newman et al., 2007; Daniel et al., 1995). Changes to the EESC are mainly due to  $Cl_v$ , as  $Br_v$  makes up a smaller fraction
- (Strahan et al., 2014). 35

The size of the Antarctic ozone hole varies and depends on the amount of  $Cl_v$  and on stratospheric temperature and dynamics (Newman et al., 2004)(e.g., Newman et al., 2004). However,  $Cl_v$  data is are sparse in the polar stratosphere and likewise the amount of measurements of total organic chlorine ( $CCl_v$ ). In contrast, there are many more observations from e. g. remote sensing instruments of nitrous oxide (N2O), which can be used to determine  $Cl_v$ . A common tool to determine  $Cl_v$  
[revised manuscript text omitted]
 390380 K, with values between  $190\pm99171\pm78$  ppt and  $270260\pm117$  ppt, whereas the in-
- 360 crease is stronger between  $390_{380}$  and  $400 \text{ K}_{\star}$  reaching a value of  $435\pm143446\pm124$  ppt. The profile of the vortex boundary region increases in the range from  $484\pm109502\pm110$  ppt up to  $977\pm3201090\pm377$  ppt in the  $\Theta$  interval of 360 to 395 K. The variability of the vortex boundary profile increases with height. This is partly due to the air mass classification, since the range of values in the vortex boundary region increases with increasing potential temperature (see Fig 2). Inorganic chlorine within the vortex could be obtained from  $\Theta$  between 330 to 385 K. Clv inside the vortex increases significantly up to a value of
- 365 1687±20±19 ppt. Thus, in late winter and early spring at this altitude highest measured potential temperatures, about half of the recorded chlorine is found in inorganic form, although from a meteorological point of view this year is exceptional with a significantly weakened vortex due to. Despite this amount of inorganic chlorine in the lower stratosphere, the total polar ozone column was higher than usual in September 2019. As a result of the minor SSW Accompanying the minor SSW event, the event, chlorine deactivation began earlier in 2019 and the ozone hole was about 16.4 million 10 x 106 km2 in size(NASA
- 370 Ozone Watch), making it the smallest since its discovery, thus only 20% of that in 2018 mid-September (Wargan et al., 2020).

**4.4 Comparison of Clv in the Antarctic (SouthTRAC) and Arctic (PGS) polar winter**

To compare Cly in the Antarctic polar vortex and in the Arctic polar vortex, measurements performed on the HALO aircraft during the PGS campaign were used. PGS consisted of the three partial missions POLSTRACC (Polar Stratosphere in a Changing Climate), GW-LCYCLE (Investigation of the Life cycle of gravity waves) and SALSA (Seasonality of Air mass

375 transport and origin in the Lowermost Stratosphere) to probe stratospheric air during the Arctic winter in 2015/2016 (Oelhaf et al., 2019). Flights of the PGS campaign were conducted from 17 December 2015 until 18 March 2016 and can be separated into two main phases. For this study, only the flights from the second main phase from 26 February to 18 March are investigated, since they took place in a comparable period (later winter). A The separation into vortex, vortex boundary, and mid-latitudes

mid-latitude measurements is based on the above mentioned method based on using  $N_2O$  measurements performed by the

- TRIHOP instrument on board of HALO during PGS (Krause et al., 2018) (see Figure S. 9-6 in the supporting information). As 380 shown in In section 4.2, the indirect method for the determination of  $Cl_v$ , where a direct measurement of all relevant chlorinated substances is not possible, proves to be comparable was shown to provide comparable to those obtained by the semi-direct method. During PGS, CFC-12 was measured with the ECD channel but the MS channel was in NCI mode and could not measure all chlorinated substances. Clv was therefore calculated using the indirect method and CFC-12 measurements from
- the GhOST-ECD channel during PGS. As for SouthTRAC, the scaled correlations from observations of the cryogenic whole-385 air sampling on two balloon flights inside the Arctic polar vortex in 2019 2009 and 2011 were used (correlation function for the Arctic winter 2015/2016 can be taken from the supporting information).

Fig. 9 displays the mean vertical profiles of Clv inside the vortex and Cltotical of the respective hemisphere. The vertical coordinate of the classification was selected according to the displayed vertical coordinate. Measurements from the individual

- 390 campaigns have been binned into 5 K potential temperature (a) and potential temperature difference to the local tropopause (b). The thermal tropopause according to WMO PV-based dynamical tropopause was used for PGS, as it was done for the SouthTRAC analysis. The PGS campaign averaged tropopause for the time period was at 315With a mean PV-based tropopause at 306 Kand slightly above the climatological tropopause of 309, during PGS, it is only slightly lower than during SouthTRAC with 308 K (NCEP Reanalysis Derived data provided by the NOAA/OAR/ESRL PSL, Boulder, Colorado, USA)
- 395 whereas the campaign averaged tropopause during SouthTRAC was found to be at 320 Kand therefore below the climatological tropopause of 327 K. Independent of the vertical coordinate, abundance of total chlorine () was lower in the stratosphere Cltotal) in the lower stratosphere decreased from the time of PGS (2015/2016) to the time of SouthTRAC (2019). The difference is on average about  $60\pm9.6$  ppt, thus roughly a rate of change of  $-16\pm2.6$  ppt year-1. This rate is higher than the average rate of change of -12.7±0.9 ppt year-1 given by Engel and Rigby (2018) between 2012 and 2016, considering the long-lived 400
- chlorinated substances.

Using  $\Theta$  as the vertical coordinate (Fig. 9a), vertical profiles of vortex classified Clv of PGS and SouthTRAC show different results. Although the Clv vortex profiles are similar until around 350 K, the SouthTRAC profile increased stronger and values become more than 435 more steeply, reaching values more than 444 ppt larger than during PGS within the vortex at equal those during PGS at the same potential temperatures. Differences become Differences are slightly larger when using  $\Delta \Theta$  as

- 405 the vertical coordinate (Fig 9b). Inside the vortex of the respective hemisphere,  $Cl_v$  increased stronger with height above the tropopause during SouthTRAC than during PGS. Although Although the two Cly profiles lie close together between 20 and  $2550 \text{ K} \Delta \Theta$ , the difference of Clv increased to 565 differences between them increase to 540 ppt at 65 K  $\Delta \Theta$ . The fractions of total chlorine which are in the form of Clv inside the vortex and in the mid-latitudes during PGS at the same distance from the local tropppause as for the highest values within the vortex during SouthTRAC, are about 40% within the vortex and about
- 410 20% in the mid-latitudes.

Fig. 10 shows the difference between PGS and SouthTRAC Clv in a latitude-altitude cross section. As a horizontal coordinate, equivalent latitude\* was used, i. e. the geographic latitude for all tropospheric observations and equivalent latitude for all stratospheric ones (Keber et al., 2020). As a vertical coordinate,  $\Delta \Theta$  was used. Since the tropopause height of each respective hemisphere is different and changes with the season, the tropopause relative coordinate  $\Delta \Theta$  accounts for tropopause variabil-

- 415 ity and better allows for better comparison of Cly. The data have been binned in 5° latitude and 5K of potential temperature relative to the local tropopause. Only bins which contain at least five data points were considered in this analysis. The difference was calculated by subtracting each Southern Hemispheric latitude-altitude bin from the equivalent Northern Hemispheric latitude-altitude bin. Values in the troposphere differ only slightly. In the lower stratosphere, a separation into two areas can be seen. In the lower stratosphere at higher latitudes, overall higher mixing ratios were derived during South-
- 420 TRAC in comparison to PGS. Differences between SouthTRAC and PGS amount to 601There are two bins with substantially higher values during SouthTRAC with around 900 ppt at maximum between 65 and 70more Cly between 80 and 90 K of  $\Delta\Theta$  and 65 to 70° equivalent latitude. This difference is much larger than the difference when comparing vortex classified measurements. Thus, it is very likely that vortex and outer vortex values are compared due to the different Arctic and Antarctic vortex size. The stratosphere of the mid-latitudes shows consistently higher Cly values during PGS. Highest levels The highest

[revised manuscript text omitted]

---

## Referee Report (RR1)

**Re-review of "Comparison of Inorganic Chlorine in the Antarctic and Arctic lowermost stratosphere by separate Late Winter aircraft measurements" by Jesswein et al.**

The manuscript has been substantially revised in response to the comments of the two referees. In general, the authors have done a good job in responding to the points raised by the reviewers, and the manuscript has been much improved. However, some of the concerns from the original reviews have not been adequately addressed, and new issues have been introduced through the revision process. Thus I feel that further corrections and clarifications, as detailed below, are necessary before the paper can be accepted for publication.

As before, both major substantive issues and minor points of clarification, wording suggestions, and grammar / typo corrections are listed together for each Section (including Supporting Information) in sequential order through the manuscript.

Respectfully,
Michelle Santee

Abstract & Section 1:
- L1: extend --> extent
- L12-13: could be --> was (in all three places in these two lines)
- L50: delete "results"

Section 2:
- L78: No need to define "RGA" here as the airport codes are given in L84-85
- L90: type --> types
- L98: ERA-5 data --> ERA-5
- L99: dynamic --> dynamical
- L100: the 2 PVU --> 2 PVU
- L101: potential temperature … as tropopause --> the potential temperature … as the tropopause
- L110: the spelling of "ionisation" here is inconsistent with that in L111 and L144
- L122: precision … measurement were --> precision … measurements was
- L128: 0.2% and 0.64% --> 0.2% and 0.64%, respectively

Section 3:
- L146: region --> region of origin
- L150: reanalysis --> reanalyses
- L153: theta has already been defined in L53 and is again defined in L174, so it does not need to be defined here
- L154-160: Much of this discussion repeats concepts that have already been covered adequately in the Introduction (L47-54). Since the audience for this paper is likely to be quite familiar with this background material, it is arguably not necessary to include it at all, but in any case it certainly does not need to be reiterated. I suggest merging and streamlining the description in these two places. In addition, it would be better not to have two sentences in one paragraph starting with "A tracer like $N_2O$ …".

- L158: a subsidence --> subsidence
- L160-161: a small variability --> small variability
- L162-163: "the vertical gradient in mid-latitude $N_2O$" – is vertical gradient really meant here, or horizontal gradient?
- L163: it would be clearer to say "both tropical and polar air"
- L171-172: affected by the diabatic decent --> affected by diabatic descent [typo: "decent"]
- L176-202: Overall, the description of the reference profile derivation has been improved in the revised manuscript. However, additional points of clarification are needed:
  - L183: I am confused about the binning in $\theta$ (or $\Delta\theta$). It is stated here that the measurements are binned in 5 K intervals. But according to the Appendix, that is true only for the vortex reference; for mid-latitudes, a 2 K bin was used. But then Figure S2 and its caption suggest that both 2 K and 5 K bins are used for both vortex and midlatitude profiles at different stages of the process. Please clarify.
  - L189: the descending stops --> descent generally stops
  - L192: I do not think that "calculated" is the right word here. Measurement points are being categorized to segregate them into the $N_2O_{vor}$ and $N_2O_{mid}$ data sets, but saying that these data sets are "calculated" gives the impression that some other manipulation is being done.
  - L194-199: These sentences are particularly badly written and the English is very confusing. Words like "added" are misused, and, unless I have misunderstood, the $N_2O$ data sets are being mixed up with the fit functions. Assuming this is an accurate description, I suggest rewriting these lines as: "The following then applies for each $N_2O$ measurement: if the mixing ratio is below the vortex fit function plus the prescribed cutoff value, then it is assigned to the $N_2O_{vor}$ data set. Otherwise, if the mixing ratio is above the mid-latitude fit function minus the associated variability, then it is assigned to the $N_2O_{mid}$ data set. Mixing ratios above the vortex fit function plus the prescribed cutoff value and below the mid-latitude fit function minus the associated variability are assigned to the boundary region. Measurements for which the vortex fit function plus the prescribed cutoff value and the mid-latitude fit function minus the associated variability overlap cannot be uniquely classified and are assigned to both the vortex and the mid-latitudes in later analysis."
  - L200: the prescribed cutoff value --> the prescribed vortex cutoff
  - L201: variability $N_2O$ --> variability in $N_2O$
  - L201-202: The reader is referred to the supporting information "and reference therein". However, no references are given in the SI, and in fact this discussion has been moved from the SI to the Appendix. But rather than refer to the Appendix, it would probably be better to simply cite Strahan et al. (1999) here for this point.
  - Figure 2 caption: Cutoff criterion --> The vortex cutoff criterion (see main text for details); where cutoff and mid-latitude variability crosses --> where the vortex cutoff and mid-latitude variability cross
- L213: Vortex and boundary region --> The vortex and boundary regions
- L214: The first phase contains some flights that have predominantly sampled vortex --> The first phase includes some flights that predominantly sampled the vortex
- L216: vortex boundary layer --> vortex boundary air

Section 4:

- L222: took --> take; this means sampling air along --> this means that air is sampled along
- L228: both channels --> the two channels
- Figure 4 caption: measured on … and on --> measured in … and in
- L230: measurements on --> measurements in
- L235: show a higher --> show higher
- L238: up-sampled --> up-sampled data
- Figure 5 caption: The first sentence of the caption needs to be re-written to make it clear what the comparison is with.  Also, measured on --> measured in
- L244: Air enters predominantly --> Air enters the stratosphere predominantly
- Section 4.2: The discussion of the semi-direct and indirect calculations of $Cl_y$ remains quite confusing.
  - Unless I am mistaken, what is referred to here as the "semi-direct" calculation – that is, Eqn. (1) – is performed twice: once with $CCl_y$ obtained from up-sampled GhOST-MS measurements, and a second time with $CCl_y$ calculated from CFC-12 alone.  This is not immediately clear from the current text, but it should be spelled out explicitly.  For example, the distinction can be drawn between the calculations mentioned in L257-258 and those mentioned in L282-284, with Eqn. (1) referenced in both places.
  - L275-277: These lines talk about comparing the semi-direct $Cl_y$ values from SouthTRAC to "indirectly determined" values, but Figure 6 shows only the correlations of various species with CFC-12, not $Cl_y$.  Also, indirectly determine --> indirectly determined
  - L280: form --> from
  - L284: delete "hereafter"
  - Then, again assuming that I have understood the process, once $CCl_y$ has been derived just from CFC-12 data (the second step above), a curve is fit to those values to obtain the coefficients for the "indirect" method of Eqn. (2).  Is that correct?  The problem is, at this point the reader has no idea why this extra step is needed.  So far only SouthTRAC measurements have been discussed, and for them $Cl_y$ can be obtained semi-directly from Eqn. (1) with the in situ $CCl_y$ data (the first step above).  The justification of the need for the indirect approach should come \*before\* it is presented, not after.
  - In addition, the paragraph in L259-294 is extremely long and tiring to read.  Thus I suggest inserting a paragraph break in L284, after "between $Cl_{total}$ and $CCl_y$".  The new paragraph should begin with a brief sentence of explanation as to why it is necessary to go through the effort of implementing the indirect approach, maybe something along the lines of: "With the good agreement between the two semi-direct $Cl_y$ calculations, we explore whether $Cl_y$ can be successfully estimated from CFC-12 for situations in which measurements of $CCl_y$ are not available.  That is, a correlation function …".
  - L288: correlation to --> correlation with
  - L291: Just to be really clear: CFC-12 … is used for --> CFC-12 … is used as the reference in Eqn. (2) for
- L297: Very young air also shows larger differences between the two $Cl_y$ estimates.
- L298: the suggested lifetime --> its suggested lifetime
- L301: I think it would be clearer to say "… age tracers.  The fundamental picture does not change, however, hence we use the uncorrected mean age of air."
- L314: delete "of $\theta$"

- L315: errorbars --> error bars
- L316-317: $Cl_y$ is given for all measurements --> the $Cl_y$ is estimated based on all measurements
- L317-318: $Cl_y$ is given according to the region --> $Cl_y$ is estimated separately in each region
- L320-322: First, Figure 8 should be referenced again at the beginning of this paragraph. Second, it is stated that the SouthTRAC measurements of long-lived chlorinated substances are consistent with the $Cl_{total}$ from AGAGE, but I'm not sure exactly what that means. Is this statement referring specifically to the troposphere?
- Figure 8 caption: -90° to -40° --> 40°-90° S
- L329: at highest --> at the highest
- L330: that in 2018 mid-September --> that in mid-September 2018
- L344-345: provide comparable --> provide values comparable
- L352-353: With a mean PV-based tropopause at 306 K, during PGS, it is only slightly lower than during SouthTRAC with 308 K --> The mean PV-based tropopause was at 306 K during PGS, only slightly lower than that during SouthTRAC at 308 K
- L355-357: First, I'm not sure what "considering the long-lived chlorinated substances" means. Does this statement refer to WMO (2018)? More importantly, the difference between the rate of decline seen in this study and that reported in WMO (2018) (~16 vs. 12.7 ppt/yr) is nonnegligible (~25%). Can the authors speculate on what is giving rise to this discrepancy? Does it imply something about the accuracy of their estimated $Cl_y$ values?
- L362-364: This sentence is very difficult to read and confusing. Assuming that I have interpreted it correctly, I suggest re-writing as: "The maximum fraction of total chlorine in the form of $Cl_y$ during PGS at the same distance from the local tropopause as the maximum SouthTRAC $Cl_y$ fraction is about 20% in the mid-latitudes (not shown) and about 40% inside the vortex (Fig. 9b)." Note that the original sentence suggested that the comparison was being made at the same $\Delta\theta$ as the largest values *inside the vortex* during SouthTRAC, but I assume that that was not what was actually done for the mid-latitudes. Also, if the NH mid-latitude result has been shown in this paper, then a specific pointer to it should be added.
- Figure 9 caption: It would be better to turn the fourth sentence around: PV tropopauses for PGS (black) and SouthTRAC (green) are displayed as dashed horizontal lines with the 1σ variability as shaded areas.
- L367-368: of each respective hemisphere --> of the two hemispheres
- L376: For maximum clarity, add "(Fig. 9)" after "measurements". Also, rather than "vortex and outer vortex", it would be better to say "vortex core and vortex edge".
- L376-379: It doesn't make sense that two sentences about the mid-latitudes (L377-378) are interposed between sentences (L376 and L379) talking about the vortex comparisons. Thus the sentence "It must be noted that … transport barrier." should be moved up and edited/merged with the sentence in L376.
- L380: The comparison on equivalent latitude is therefore only possible to a limited extent --> Therefore performing the comparison in equivalent latitude / potential temperature coordinates removes only some of the sources of discrepancy
- L380-395: The entire discussion of the interhemispheric differences in Fig. 10 is poorly written and not well thought-through. The authors have placed on the reader the burden of figuring out how previously reported BDC and age of air differences may be related to the

differences in $Cl_y$ that they observe.  The linkage between differences in age of air (and the trends therein) and $Cl_y$ differences must be drawn much more explicitly in the paper.  The discussion of how their results relate to previous mid-latitude trends is also muddled.

- o L382: Konopka et al. (2015) showed, that north of 60 °N, age of air is always younger than south of 60 °S in the same season --> Konopka et al. (2015) showed that the age of air is always younger north of 60 °N than south of 60 °S in the corresponding season
- o Since air is younger north of 60N than it is south of 60S, it seems to me that lower values of $Cl_y$ are expected at NH high latitudes than at SH high latitudes, as seen in Figs. 9 and 10. I do not believe that such a statement is made in the manuscript.
- o How do the differences in $Cl_{total}$ between the two campaigns – which are not well explained as noted above – interact with / affect the interhemispheric Cly differences?
- o The authors mention recent work quantifying interhemispheric differences in the trends in AoA, but again the connection to their $Cl_y$ results is not made explicitly.  If there is a positive trend in AoA in the NH (i.e., air is getting older) and a negative trend in AoA in the SH (i.e., air is getting younger), then wouldn't that mean that $Cl_y$ should be getting slightly larger in NH mid-latitudes and smaller in SH mid-latitudes?  But surely these trends are very small and could not be expected to be evident above interannual variability over the three-year interval between the campaigns.  Thus I am not convinced that AoA trends have any relevance for the observed $Cl_y$ differences.
- o L385-390: The discussion of long-term trends in mid-latitude HCl and $ClONO_2$ has been largely lifted from WMO (2018) without attribution.  While I applaud their desire to go back to original sources, the authors have been sloppy and careless in presenting this material.  For example, they state that Mahieu et al. (2014) reported data "through the end of 2016".  In fact, Mahieu et al. (2014) only show data through 2011; the plots are updated through 2016 in the WMO Report.  Then it is stated that GOZCARDS lower stratospheric HCl shows "larger" decreases at SH latitudes – larger than what?  Than the increases seen in the NH?  The citation for this statement is Froidevaux et al. (2015), but a more recent paper by Froidevaux et al. published in 2019 would be a more up-to-date reference.  Most importantly, nowhere in these lines in the manuscript are the actual trends for HCl and $ClONO_2$ quoted.  According to WMO (2018), for the period 1997–2016, total column HCl decreased by $0.42\pm0.23\%yr^{-1}$ and total column $ClONO_2$ decreased by $0.60\pm0.39\%yr^{-1}$.  It doesn't seem to me that trends of that magnitude could account for the >200 ppt difference in estimated mid-latitude $Cl_y$ between the PGS and SouthTRAC campaigns seen in Fig. 10.  Therefore the statement that "higher values of $Cl_y$ in the mid-latitudes during PGS [seem] to be plausible" is not supported.
- L391-392: It is not clear to me what is meant by "the lowest 20 K above the local tropopause show weak impact of the stronger Antarctic".  Is this sentence referring to the Antarctic vortex?  Even if it is, its meaning is unclear. Moreover, in Fig. 10 bins in the lowest 20 K above the tropopause are not filled at the highest EqLs, so the relevance of the vortex is not clear.
- L392: Two exceptions are singled out for discussion, but other bins in the vicinity show differences nearly as large.
- L394: this range --> this $\theta$ range; almost zero --> generally small

Section 5:
- L411: Since a reference is included for the Arctic winter of 2015/2016, it would be appropriate to include one here for the Antarctic winter of 2019 as well.
- L422: can be --> was (twice)
- L423: vortex --> vortices
- L425: Elsewhere "inter annual" has been written as "inter-annual" (L18) and "interannual" (L37).  Please be consistent.
- L431: can be --> were (twice)
- L432-433: The statement "These hemispheric differences can also be found in simulations based on reanalysis, e.g., Konopka et al. (2015)" is a bit misleading here because the preceding sentence was about $Cl_y$, whereas Konopka et al. (2015) did not discuss $Cl_y$.  Again, the connection between AoA and the $Cl_y$ calculated in this work needs to be made directly.
- L434: should be subject to --> should be the subject of
- L435: which are not only used to --> which are used to not only
- L436: reveal --> reflect

Appendix, References, & Supplemental Material:
- L455: this corresponds --> 6 ppb corresponds
- L460: outlier --> outliers
- L628: implivations --> implications
- L632: Arctiv --> Arctic
- SI, L4: requieres --> requires
- SI, L5: Figure S 5 --> Figure S 4
- SI, L10: Fig. S 7 --> Figure S 6
- Figure S 7: It needs to be made clear in the caption that the displayed curves were derived from CFC-12 from GhOST-ECD and $N_2O$ from UMAQS.

---

## Referee Report (RR2)

**Third review of "Comparison of Inorganic Chlorine in the Antarctic and Arctic lowermost stratosphere by separate Late Winter aircraft measurements" by Jesswein et al.**

The manuscript has again been revised and substantially improved. Only a few minor wording issues, either overlooked by the authors in their latest revision or introduced in the newly added text, remain to be addressed. After that the paper is ready for publication.

Respectfully,
Michelle Santee

- L186: the descending generally stops --> descent generally stops
- L196: unique --> uniquely
- L254: where no measurements of all major chlorine containing substances are available --> where measurements of all major chlorine-containing substances are not available
- L273: indirectly determine --> indirectly determined
- L355: difference between $Cl_{total}$ from controlled substances during PGS and SouthTRAC --> difference between $Cl_{total}$ from controlled substances during PGS and that during SouthTRAC
- L383: SouhTRAC --> SouthTRAC
- L384-386: This wording is confusing. I think this would be clearer: "Taking into account the difference in $Cl_{total}$ between the two campaigns would reduce the observed differences in the mid-latitudes, where mixing ratios of $Cl_y$ were higher during PGS, but would increase the differences in the high latitudes, where higher $Cl_y$ was derived for SouthTRAC."
- L387: than they can --> than can; difference of $Cl_{total}$ --> difference in $Cl_{total}$
- L390: "transported" is not the right word; "converted" or "transformed" would be better
- L391-392: It would be more appropriate to state that the results of this work are consistent with those of Konopka et al. (2015), since that study came first: "Therefore, the observed differences in $Cl_y$, with higher $Cl_y$ values in the southern high latitudes than in the northern high latitudes (see Fig. 9 and 10), are consistent with the differences in the age of air found by Konopka et al. (2015)."
- L393-394: showed an updated report to the long-term total column data for HCl and $ClONO_2$ (representing $Cl_y$) by Mahieu et al. (2014) --> updated the long-term total column HCl and $ClONO_2$ (representing $Cl_y$) record reported by Mahieu et al. (2014); also: though --> through
- L396: add a comma after "stations"
- L401: I do not understand what is meant by "the trends given are indicative of the interhemispheric differences in $Cl_y$". Why would the rates of change in $Cl_y$ necessarily reflect its absolute abundances in the two hemispheres?
- L402: of about 200 ppt and above --> of 200 ppt or more; also, why is the word "explicitly" used here? It does not seem appropriate and probably should be deleted.
- L442: was found --> were found (in two places)
- L444-445: reanalyses data --> reanalysis data

---

## Author Response (AR2)

**Comparison of Inorganic Chlorine in the Antarctic and Arctic lowermost stratosphere by separate Late Winter aircraft measurements**

We would like to thank again for the constructive comments on our manuscript. In the following, we address the respective proposals for improvement. The changes are explained in detail and answered point by point. Reviewer comments are in normal font. Our answers are in italic and changes to the manuscript in blue.

Response to Referee #2

Abstract & Section 1:

- L1: extend --> extent

  *Done.*

- L12-13: could be --> was (in all three places in these two lines)

  *Done.*

- L50: delete "results"

  *Done.*

Section 2:

- L78: No need to define "RGA" here as the airport codes are given in L84-85

  *RGA was deleted at this point.*

- L90: type --> types

  *Done.*

- L98: ERA-5 data --> ERA-5

  *Data was deleted.*

- L99: dynamic --> dynamical

  *Done.*

- L100: the 2 PVU --> 2 PVU

  *Done.*

- L101: potential temperature ... as tropopause --> the potential temperature ... as the tropopause

  *Done.*

- L110: the spelling of "ionisation" here is inconsistent with that in L111 and L144

  *We changed "ionisation" to "ionization" to be consistent with L111. L144 does not contain information about the ionization.*

- L122: precision ... measurement were --> precision ... measurements was

  *Done.*

- L128: 0.2% and 0.64% --> 0.2% and 0.64%, respectively

  *Done.*

Section 3:

- L146: region --> region of origin

  *Done.*

- L150: reanalysis --> reanalyses

  *Done.*

- L153: theta has already been defined in L53 and is again defined in L174, so it does not need to be defined here

  *We excluded the ($\theta$) in this line.*

- L154-160: Much of this discussion repeats concepts that have already been covered adequately in the Introduction (L47-54). Since the audience for this paper is likely to be quite familiar with this background material, it is arguably not necessary to include it at all, but in any case it certainly does not need to be reiterated. I suggest merging and streamlining the description in these two places. In addition, it would be better not to have two sentences in one paragraph starting with "A tracer like $N_2O$ ...".

  *We agree that the discussion regarding the polar vortex is repetitive in this section. We thus refer to the previous description of the polar vortex with the important insight for us that the trace gas mixing ratios inside and outside the vortex differ. Furthermore, as $N_2O$ was introduced a few sentences before, the first "A trace like $N_2O$..." was rewritten. The new wording is as follows:*

"[…] N$_2$O can be measured in situ with a high time resolution to reveal small scale structures in the atmosphere. As already mentioned in the introduction, air inside the polar vortex has a substantially different composition regarding trace gases than air outside the vortex. A tracer like N$_2$O … […]"

- L158: a subsidence --> subsidence

*By rewording, this expression is omitted.*

- L160-161: a small variability --> small variability

*Done.*

- L162-163: "the vertical gradient in mid-latitude N$_2$O" – is vertical gradient really meant here, or horizontal gradient?

*Yes, the vertical gradient is meant here. The vertical profile of N$_2$O in the mid-latitudes shows a smaller decrease with altitude but the variability of the profile is larger due to the influence of both, polar and tropical air. For less confusion, we have rewritten this sentence as follows:*

"[…] The vertical profile of N$_2$O in the mid-latitudes shows a weak gradient but a high variability as it is influenced by both tropical and polar air […]"

- L163: it would be clearer to say "both tropical and polar air"

*Done, see comment before.*

- L171-172: affected by the diabatic decent --> affected by diabatic descent [typo: "decent"]

*Done.*

- L176-202: Overall, the description of the reference profile derivation has been improved in the revised manuscript. However, additional points of clarification are needed:

  o L183: I am confused about the binning in θ (or Δθ). It is stated here that the measurements are binned in 5 K intervals. But according to the Appendix, that is true only for the vortex reference; for mid-latitudes, a 2 K bin was used. But then Figure S2 and its caption suggest that both 2 K and 5 K bins are used for both vortex and midlatitude profiles at different stages of the process. Please clarify.

  *I am very sorry for the confusion. The 2 K bins are used for the iterative process mentioned in the appendix (5 K bin was a remnant from the first draft. We apologize and correct it). After the iterative process, as an intermediate step for both profiles, the remaining measurements are binned in 5 K bins as described in the text. This can be seen in Figure S 2 c) and d). Since we have only written "(see Figure S 2)" this can lead to confusion. Therefore, we added c) and d).*

"[…] (see Figure S 2 c and d) […]"

o   L189: the descending stops --> descent generally stops

*Done.*

o   L192: I do not think that "calculated" is the right word here. Measurement points are being categorized to segregate them into the $N_2O_{vor}$ and $N_2O_{mid}$ data sets but saying that these data sets are "calculated" gives the impression that some other manipulation is being done.

*The essential information is missing. We apologize for this inaccuracy. The $N_2O_{vor}$ and $N_2O_{mid}$ data set are calculated from $\theta$ or $\Delta\theta$ using the fit function for every $N_2O$ measurement point of the UMAQS instrument for all flights. We have added this information with the following reformulation:*

"[…] A vortex and mid-latitude reference $N_2O$ data set ($N_2O_{vor}$ and $N_2O_{mid}$) can be calculated from $\theta$ or $\Delta\theta$ by using the fit function for the vortex and mid-latitude profiles for every measurement point of the UMAQS instrument for all flights […]"

o   L194-199: These sentences are particularly badly written and the English is very confusing. Words like "added" are misused, and, unless I have misunderstood, the $N_2O$ data sets are being mixed up with the fit functions. Assuming this is an accurate description, I suggest rewriting these lines as: "The following then applies for each $N_2O$ measurement: if the mixing ratio is below the vortex fit function plus the prescribed cutoff value, then it is assigned to the $N_2O_{vor}$ data set. Otherwise, if the mixing ratio is above the mid-latitude fit function minus the associated variability, then it is assigned to the $N_2O_{mid}$ data set. Mixing ratios above the vortex fit function plus the prescribed cutoff value and below the mid- latitude fit function minus the associated variability are assigned to the boundary region. Measurements for which the vortex fit function plus the prescribed cutoff value and the mid-latitude fit function minus the associated variability overlap cannot be uniquely classified and are assigned to both the vortex and the mid-latitudes in later analysis."

*In the previous point we clarified that the $N_2O$ measurements are not mixed up with the fit functions. Nevertheless, we have largely rewritten this section according to suggestions given.*

 "[…] The following then applies for each $N_2O$ measurement: if the mixing ratio is below the respective $N_2O_{vor}$ plus the prescribed vortex cutoff, then it is assigned to the vortex. Otherwise, if the mixing ratio is above the respective $N_2O_{mid}$ minus the associated variability, then it is assigned to the mid-latitudes. Mixing ratios above the respective $N_2O_{vor}$ plus the prescribed vortex cutoff and below the respective $N_2O_{mid}$ minus the associated variability are assigned to the boundary region. Measurements for which the respective $N_2O_{vor}$ plus the prescribed vortex cutoff and the $N_2O_{mid}$ minus the associated variability overlap cannot be unique classified and are assigned to both the vortex and the mid-latitudes in later analysis. […]"

o L200: the prescribed cutoff value --> the prescribed vortex cutoff

*Done.*

o L201: variability $N_2O$ --> variability in $N_2O$

*Done.*

o L201-202: The reader is referred to the supporting information "and reference therein". However, no references are given in the SI, and in fact this discussion has been moved from the SI to the Appendix. But rather than refer to the Appendix, it would probably be better to simply cite Strahan et al. (1999) here for this point.

*We apologize for this, which is a remnant of the first submission. We now cite Strahan et al. (1999) here as suggested.*

o Figure 2 caption: Cutoff criterion --> The vortex cutoff criterion (see main text for details); where cutoff and mid-latitude variability crosses --> where the vortex cutoff and mid- latitude variability cross

*Done; Done.*

- L213: Vortex and boundary region --> The vortex and boundary regions

  *Done.*

- L214: The first phase contains some flights that have predominantly sampled vortex --> The first phase includes some flights that predominantly sampled the vortex

  *Done.*

- L216: vortex boundary layer --> vortex boundary air

  *Done.*

Section 4:

- L222: took --> take; this means sampling air along --> this means that air is sampled along

  *Took was changed to takes; Done.*

- L228: both channels --> the two channels

  *Done.*

- Figure 4 caption: measured on ... and on --> measured in ... and in

  *Done.*

- L230: measurements on --> measurements in

  *Done.*

- L235: show a higher --> show higher

  *Done.*

- L238: up-sampled --> up-sampled data

  *Done.*

- Figure 5 caption: The first sentence of the caption needs to be re-written to make it clear what the comparison is with. Also, measured on --> measured in

  *The second sentence explains the difference between the red and black profile. We deleted "Comparison" at the beginning so that it is clear that only CFC-11 is shown in this Figure; Done.*

- L244: Air enters predominantly --> Air enters the stratosphere predominantly

  *Done.*

- Section 4.2: The discussion of the semi-direct and indirect calculations of $Cl_y$ remains quite confusing.

  o Unless I am mistaken, what is referred to here as the "semi-direct" calculation – that is, Eqn. (1) – is performed twice: once with $CCl_y$ obtained from up-sampled GhOST-MS measurements, and a second time with $CCl_y$ calculated from CFC-12 alone. This is not immediately clear from the current text, but it should be spelled out explicitly. For example, the distinction can be drawn between the calculations mentioned in L257-258 and those mentioned in L282-284, with Eqn. (1) referenced in both places.

  *We have taken the suggested changes and referred to Eq. 1 at proposed places:*

  "[…] In the following, $Cl_y$ derived from the difference between the estimated entry mixing ratios and observed $CCl_y$ from the in situ measurements (see Eq. 1) is referred to as the semi-direct calculation of $Cl_y$. […] As already mentioned earlier, $Cl_{total}$ is also needed for the calculation of $Cl_y$ (see Eq. 1). […]"

  o L275-277: These lines talk about comparing the semi-direct $Cl_y$ values from SouthTRAC to "indirectly determined" values, but Figure 6 shows only the correlations of various species with CFC-12, not $Cl_y$. Also, indirectly determine --> indirectly determined

*This sentence is meant to introduce the reader to what follows in the rest of this subsection. After carefully reading this sentence again, it is indeed confusing that only the semi-direct $Cl_y$ is mentioned at this point. It would have been better to mention both semi-direct and indirect $Cl_y$ or to delete both from the sentence. For more clarity, we have deleted "semi-direct Cly" at this point in the text, as the next lines only show the comparison of the correlations and the comparison of $Cl_y$ appears later in this subsection.*

o L280: form --> from

*Done.*

o L284: delete "hereafter"

*Done.*

o Then, again assuming that I have understood the process, once $CCl_y$ has been derived just from CFC-12 data (the second step above), a curve is fit to those values to obtain the coefficients for the "indirect" method of Eqn. (2). Is that correct? The problem is, at this point the reader has no idea why this extra step is needed. So far only SouthTRAC measurements have been discussed, and for them $Cl_y$ can be obtained semi-directly from Eqn. (1) with the in situ $CCl_y$ data (the first step above). The justification of the need for the indirect approach should come *before* it is presented, not after.

*It is correct, that at this point $CCl_y$ is derived from CFC-12 alone. Additionally, $Cl_{total}$ is also derived from the mean age of air from the balloon measurements, already mentioned in the paragraph. The justification of the indirect method can be found at the beginning of this paragraph with "For the case where no measurements of chlorine containing substances …". We now complete the introductory sentence of this paragraph with an outlook that the indirect method is needed for the comparison with the NH.*

"[…] For the case where no measurements of all major chlorine containing substances are available, $Cl_y$ has in the past been calculated indirectly based on correlations derived from previous measurement campaigns. This also applies to $Cl_y$ from the Northern Hemisphere, later in the analyses (see section 4.4). […]"

o In addition, the paragraph in L259-294 is extremely long and tiring to read. Thus I suggest inserting a paragraph break in L284, after "between $Cl_{total}$ and $CCl_y$". The new paragraph should begin with a brief sentence of explanation as to why it is necessary to go through the effort of implementing the indirect approach, maybe something along the lines of: "With the good agreement between the two semi-direct $Cl_y$ calculations, we explore whether $Cl_y$ can be successfully estimated from CFC-12 for situations in which measurements of $CCl_y$ are not available. That is, a correlation function …".

*We acknowledge that this paragraph is very long, and potential readers can thereby lose track. We further therefore now start a new paragraph after "between Cl$_{total}$ and CCl$_y$". The beginning of the new paragraph was changed slightly as follows:*

"[…] With the good agreement between observed correlations and scaled correlations from the balloon measurements and the previously described determination of Cl$_y$, we explore whether Cl$_y$ can be successfully estimated from CFC-12 alone. That is, a correlation function… […]"

○ L288: correlation to --> correlation with

*Done.*

○ L291: Just to be really clear: CFC-12 … is used for --> CFC-12 … is used as the reference in Eqn. (2) for

*Done.*

- L297: Very young air also shows larger differences between the two Cl$_y$ estimates.

*To be more precisely, we have rewritten the following sentence:*

"[…] The difference between the two methods is rather small, with less than around 30 ppt difference between 1 and 4 years of mean age and a maximum difference of about 65 ppt at 5 years of mean age. […]"

- L298: the suggested lifetime --> its suggested lifetime

*Done.*

- L301: I think it would be clearer to say "... age tracers. The fundamental picture does not change, however, hence we use the uncorrected mean age of air."

*Thanks for the more appropriate description. Changes were done as suggested.*

- L314: delete "of θ"

*Done.*

- L315: errorbars --> error bars

*Done.*

- L316-317: Cl$_y$ is given for all measurements --> the Cl$_y$ is estimated based on all measurements

*Done.*

- L317-318: Cl$_y$ is given according to the region --> Cl$_y$ is estimated separately in each region

  *Done.*

- L320-322: First, Figure 8 should be referenced again at the beginning of this paragraph. Second, it is stated that the SouthTRAC measurements of long-lived chlorinated substances are consistent with the Cl$_{total}$ from AGAGE, but I'm not sure exactly what that means. Is this statement referring specifically to the troposphere?

  *We included "In Figure 8 …" at the beginning of this paragraph. Yes, this statement is referring to the troposphere. The near zero Cl$_y$ throughout the troposphere indicates, that the observations during the SouthTRAC campaign is very close to Cl$_{total}$ that we have determined using the AGAGE time series. However, one can also draw the conclusion that the calibration of our stratospheric measurements of the chlorinated substances is consistent with the AGAGE calibration scales. We have rewritten the sentence regarding the consistency with the AGAGE measurements as follows:*

  "[…] The tropospheric measurements during SouthTRAC are thus consistent with Cl$_{total}$ derived from ground based AGAGE measurements. […]"

- Figure 8 caption: -90° to -40° --> 40°-90° S

  *Done.*

- L329: at highest --> at the highest

  *Done.*

- L330: that in 2018 mid-September --> that in mid-September 2018

  *Done (L332 not L330).*

- L344-345: provide comparable --> provide values comparable

  *Done.*

- L352-353: With a mean PV-based tropopause at 306 K, during PGS, it is only slightly lower than during SouthTRAC with 308 K --> The mean PV-based tropopause was at 306 K during PGS, only slightly lower than that during SouthTRAC at 308 K

  *Done.*

- L355-357: First, I'm not sure what "considering the long-lived chlorinated substances" means. Does this statement refer to WMO (2018)? More importantly, the difference between the rate of decline seen in this study and that reported in WMO (2018) (~16 vs. 12.7 ppt/yr) is nonnegligible (~25%). Can the authors speculate on what is giving rise to this discrepancy? Does it imply something about the accuracy of their estimated Cl$_y$ values?

*The value of 12.7 ppt/yr from the WMO (2018) represents the total controlled chlorine, e.g. all CFCs, CCl$_4$, HCFCs, CH$_3$CCl$_3$ and halon-1211. Instead of saying "considering the long-lived chlorinated substances", the more correct expression is "for the controlled substances", which we included in our manuscript.*

*We have deleted the decline rate derived from the difference in Cl$_{total}$ during PGS and SouthTRAC (the given 16±2.6 ppt/year). Instead, we use the decline rate from the WMO Report (2018) to show that around 45 ppt of the difference in Cl$_{total}$ between the two campaigns (60 ppt) can be explained by the decline rate of -12.7±0.9 ppt derived in WMO (2018). The remaining 15 ppt can be explained by the higher Cl$_{total}$ in the Northern Hemisphere. We added the following information to the manuscript as follows:*

"[…] The difference between Cl$_{total}$ from controlled substances during PGS and SouthTRAC is about 60±9.6 ppt. This difference can be explained by a combination of temporal trends of controlled substances and interhemispheric gradients. Using the rate of decline from Engel and Rigby (2018) of -12.7±0.9 ppt year$^{-1}$ for the controlled substances, a difference of about 45 ppt is expected due to the time difference between the two campaigns. The remaining difference of about 15 ppt can be explained by the higher Cl$_{total}$ in the Northern Hemisphere […]"

- L362-364: This sentence is very difficult to read and confusing. Assuming that I have interpreted it correctly, I suggest re-writing as: "The maximum fraction of total chlorine in the form of Cl$_y$ during PGS at the same distance from the local tropopause as the maximum SouthTRAC Cl$_y$ fraction is about 20% in the mid-latitudes (not shown) and about 40% inside the vortex (Fig. 9b)." Note that the original sentence suggested that the comparison was being made at the same Δθ as the largest values *inside the vortex* during SouthTRAC, but I assume that that was not what was actually done for the mid-latitudes. Also, if the NH mid- latitude result has been shown in this paper, then a specific pointer to it should be added.

*We thank you for the suggestion, which we almost completely used in the manuscript. We get these rough percentages at the same Δθ as the largest values inside the vortex during SouthTRAC, as you suggested.*

"[…] The fraction of total chlorine in the form of Cl$_y$ during PGS at the same distance from the local tropopause as the maximum SouthTRAC Cl$_y$ fraction is about 20% in the mid-latitudes (not shown) and about 40% inside the vortex (Fig. 9b). […]"

- Figure 9 caption: It would be better to turn the fourth sentence around: PV tropopauses for PGS (black) and SouthTRAC (green) are displayed as dashed horizontal lines with the 1σ variability as shaded areas.

*Done.*

- L367-368: of each respective hemisphere --> of the two hemispheres

*Done.*

- L376: For maximum clarity, add "(Fig. 9)" after "measurements". Also, rather than "vortex and outer vortex", it would be better to say "vortex core and vortex edge".

  *We added "(Fig. 9)" at the suggested position and changed "vortex and outer vortex" to "vortex core and vortex edge".*

- L376-379: It doesn't make sense that two sentences about the mid-latitudes (L377-378) are interposed between sentences (L376 and L379) talking about the vortex comparisons. Thus the sentence "It must be noted that ... transport barrier." should be moved up and edited/merged with the sentence in L376.

  *We rearranged theses lines according to the suggestion as follows:*

  "[...] Thus, it is very likely that vortex core and vortex edge values are compared due to the different Arctic and Antarctic vortex size, stability and strength of the transport barrier. Therefore, performing the comparison in equivalent latitude/potential temperature coordinates removes only some of the sources of discrepancy. The stratosphere of the mid-latitudes shows consistently higher $Cl_y$ values during PGS. The highest values of $Cl_y$ reached are 315 ppt greater during PGS between 65 and 70 K of $\Delta\theta$ and 40 to 45 ° equivalent latitude. [...]"

- L380: The comparison on equivalent latitude is therefore only possible to a limited extent --> Therefore performing the comparison in equivalent latitude / potential temperature coordinates removes only some of the sources of discrepancy

  *Done (see point above).*

- L380-395: The entire discussion of the interhemispheric differences in Fig. 10 is poorly written and not well thought-through. The authors have placed on the reader the burden of figuring out how previously reported BDC and age of air differences may be related to the differences in $Cl_y$ that they observe. The linkage between differences in age of air (and the trends therein) and $Cl_y$ differences must be drawn much more explicitly in the paper. The discussion of how their results relate to previous mid-latitude trends is also muddled.

  *We address this discussion by responding to the following comments that relate to it.*

  o L382: Konopka et al. (2015) showed, that north of 60 °N, age of air is always younger than south of 60 °S in the same season --> Konopka et al. (2015) showed that the age of air is always younger north of 60 °N than south of 60 °S in the corresponding season

  *Done.*

  o Since air is younger north of 60N than it is south of 60S, it seems to me that lower values of $Cl_y$ are expected at NH high latitudes than at SH high latitudes, as seen in Figs. 9 and 10. I do not believe that such a statement is made in the manuscript.

*We have linked to our results in this regard following the findings of Konopka et al. (2015). The following was added right after presenting the result of Konopka et al. (2015):*

"[…] Older air will be higher in Cly as a larger fraction of total chlorine has already been transformed to the inorganic form. The differences in age of air found by Konopka et al. (2015) are therefore consistent with the observed differences in $Cl_y$ with higher $Cl_y$ values in the southern high latitudes than in the northern high latitudes (see Fig. 9 and 10). […]"

o   How do the differences in $Cl_{total}$ between the two campaigns – which are not well explained as noted above – interact with / affect the interhemispheric Cly differences?

*We further explained the difference in $Cl_{total}$, as can be seen above in the comments. As explained above, the difference of around 60±9.6 ppt can be explained by a combination of temporal decline and interhemispheric differences and is consistent with tropospheric observations. This maximum difference of 60±9.6 ppt would only propagate completely to $Cl_y$ when all chlorine is converted to the inorganic form. Taking into account the difference in $Cl_{total}$ between the two campaigns will reduce the observed differences in the mid-latitudes with higher MR during PGS and increase the differences in the high latitudes with higher MR of $Cl_y$ during SouthTRAC. However, the effects should be small, and the differences shown between NH and SH $Cl_y$ are considerably larger than can be explained by the differences in $Cl_{total}$. We have tried to briefly mention the effect of $Cl_{total}$ decline in the text with the following statement in section 4.4:*

"[…] The already mentioned difference of $Cl_{total}$ between PGS and SouhTRAC amounts to 60±9.6 ppt. This maximum difference of 60±9.6 ppt will only propagate completely to $Cl_y$ when all chlorine is converted to the inorganic form. Taking into account the difference in $Cl_{total}$ between the two campaigns will reduce the observed differences in the mid-latitudes with higher mixing ratios of $Cl_y$ during PGS and increase the differences in the high latitudes with higher Cly derived for the SouthTRAC campaign. The observed differences in $Cl_y$ are thus clearly larger than they can be explained by the temporal difference of $Cl_{total}$. […]"

o   The authors mention recent work quantifying interhemispheric differences in the trends in AoA, but again the connection to their $Cl_y$ results is not made explicitly. If there is a positive trend in AoA in the NH (i.e., air is getting older) and a negative trend in AoA in the SH (i.e., air is getting younger), then wouldn't that mean that $Cl_y$ should be getting slightly larger in NH mid-latitudes and smaller in SH mid-latitudes? But surely these trends are very small and could not be expected to be evident above interannual variability over the three-year interval between the campaigns. Thus I am not convinced that AoA trends have any relevance for the observed $Cl_y$ differences.

*Assuming that you are referring to the set of results from Heanel et al. (2015). We agree that the given age of air trends have no relevance for our observed Cly differences. Therefore, we deleted this sentence.*

o  L385-390: The discussion of long-term trends in mid-latitude HCl and ClONO$_2$ has been largely lifted from WMO (2018) without attribution. While I applaud their desire to go back to original sources, the authors have been sloppy and careless in presenting this material. For example, they state that Mahieu et al. (2014) reported data "through the end of 2016". In fact, Mahieu et al. (2014) only show data through 2011; the plots are updated through 2016 in the WMO Report. Then it is stated that GOZCARDS lower stratospheric HCl shows "larger" decreases at SH latitudes – larger than what? Than the increases seen in the NH? The citation for this statement is Froidevaux et al. (2015), but a more recent paper by Froidevaux et al. published in 2019 would be a more up-to-date reference. Most importantly, nowhere in these lines in the manuscript are the actual trends for HCl and ClONO$_2$ quoted. According to WMO (2018), for the period 1997–2016, total column HCl decreased by 0.42±0.23%yr$^{-1}$ and total column ClONO$_2$ decreased by 0.60±0.39%yr$^{-1}$. It doesn't seem to me that trends of that magnitude could account for the >200 ppt difference in estimated mid-latitude Cl$_y$ between the PGS and SouthTRAC campaigns seen in Fig. 10. Therefore the statement that "higher values of Cl$_y$ in the mid- latitudes during PGS [seem] to be plausible" is not supported.

*We apologize for the sloppy work in presenting the material and have carefully revisited this section. Regarding HCl and ClONO$_2$ from Mahieu et al. (2014), we clarify that Engel and Rigby (2018) reported an update from Mahieu et al. (2014). We now also present the trends listed in the WMO Report (2018). We updated the reference as well as the reported outcome by Froidevaux et al. (2015 and 2019). In addition, we have replaced the sentence " Thus, higher values of Cl$_y$ in the mid-latitudes during PGS seems to be plausible ". The trends shown do not explain the difference shown in Fig. 10, but they do help to understand that there is an interhemispheric difference and in which direction it is moving.*

"[…] In addition, Engel and Rigby (2018) showed an updated report to the long-term total column data for HCl and ClONO2 (representing Cl$_y$) by Mahieu et al. (2014) in the stratosphere, at Jungfraujoch (46.5 °N) and at Lauder (45 °S), through the end of 2016. A negative trend of Cl$_y$ is observed at both stations but with a non-significant trend for the Jungfraujoch data over the last decade and a slightly larger negative trend from the Lauder data. In addition, trends between 1997 and 2016 are given at both stations with -0.42±0.23% year$^{-1}$ for HCl and -0.60±0.39% year$^{-1}$ for ClONO$_2$ at Jungfraujoch station and -0.51±0.12% year$^{-1}$ and -0.74±0.59% year$^{-1}$ at Lauder station, respectively. Furthermore, the Global Ozone Chemistry And Related trace gas Data records for the Stratosphere (GOZCARDS) shows short-term lower-stratospheric HCl trends, which are negative at southern latitudes and slightly positive (marginal significance) at northern mid-latitudes between 2005 and 2015/2018 (Froidevaux et al., 2015, 2019). Although the trends given are indicative of the interhemispheric differences in Cl$_y$, they do not explain the difference in Cl$_y$ at mid-latitudes of about 200 ppt and above shown in Fig. 10 explicitly. […]"

- L391-392: It is not clear to me what is meant by "the lowest 20 K above the local tropopause show weak impact of the stronger Antarctic". Is this sentence referring to the Antarctic vortex? Even if it is, its meaning is unclear. Moreover, in Fig. 10 bins in the lowest 20 K above the tropopause are not filled at the highest EqLs, so the relevance of the vortex is not clear.

  *We agree that this statement is unclear. Moreover, the following sentence does not match well to the given statement here. We have therefore rewritten this sentence to not focus on the impact of the stronger Antarctic vortex. Instead, the statement should be, that the lowest 20 K above the local tropopause show in general minor differences between the two hemispheres, beside the exception given in the text.*

  "[…] The lowest 20 K above the local tropopause show in general minor differences between the two hemispheres […]"

- L392: Two exceptions are singled out for discussion, but other bins in the vicinity show differences nearly as large.

  *Only the exceptions with differences around 200 ppt were singled out. Other bins show slightly higher mixing ratios either toward PGS or SouthTRAC and are therefore not mentioned explicitly.*

- L394: this range --> this θ range; almost zero --> generally small

  *Done; Done.*

Section 5:

- L411: Since a reference is included for the Arctic winter of 2015/2016, it would be appropriate to include one here for the Antarctic winter of 2019 as well.

  *We included a reference for the statement of the weakened and shifted polar vortex in 2019. Reference used are Wargan et al. (2020) and Safieddine et al. (2020).*

- L422: can be --> was (twice)

  *Done.*

- L423: vortex --> vortices

  *Since we refer to respective vortex, vortex seems to be the appropriate form.*

- L425: Elsewhere "inter annual" has been written as "inter-annual" (L18) and "interannual" (L37). Please be consistent.

  *We now use inter-annual in all places in the manuscript.*

- L431: can be --> were (twice)

  *Done.*

- L432-433: The statement "These hemispheric differences can also be found in simulations based on reanalysis, e.g., Konopka et al. (2015)" is a bit misleading here because the preceding sentence was about $Cl_y$, whereas Konopka et al. (2015) did not discuss $Cl_y$. Again, the connection between AoA and the $Cl_y$ calculated in this work needs to be made directly.

  *The connection between mean age and $Cl_y$ was shortly introduced in section 4.2 in the discussion of Fig 7. "… inorganic chlorine increases with mean age of air, as more molecules of the organic sources are converted to the inorganic form". Nonetheless, we agree that this sentence is misleading, as the reader may think $Cl_y$ simulations were done in Konopka et al. (2015) which was not the case. We have rewritten this sentence as follows:*

  "[…] $Cl_y$ increases with increasing mean age of air, for which Konopka et al. (2005) derived similar hemispheric differences based on model simulations using meteorological reanalyses data. […]"

- L434: should be subject to --> should be the subject of

  *Done.*

- L435: which are not only used to --> which are used to not only

  *Done.*

- L436: reveal --> reflect

  *Done.*

Appendix, References, & Supplemental Material:

- L455: this corresponds --> 6 ppb corresponds

  *Done.*

- L460: outlier --> outliers

  *Done.*

- L628: implivations --> implications

  *Done.*

- L632: Arctiv --> Arctic

  *Done.*

- SI, L4: requieres --> requires

  *Done.*

- SI,L5:FigureS5-->FigureS4

  *We changed it to "Figure S 4 and S 5 display … ".*

- SI,L10:Fig.S7-->FigureS6

  *Done.*

- Figure S 7: It needs to be made clear in the caption that the displayed curves were derived from CFC-12 from GhOST-ECD and $N_2O$ from UMAQS.

  *We added the information leading to the following caption:*

  "[…] **Figure S 7**: Indirectly determined inorganic chlorine using balloon based correlations and CFC-12 from the GhOST-ECD (green) and $N_2O$ from UMAQS (black) during the SouthTRAC campaign as the reference substance. Absolute difference in red. […]"

---

## Author Response (AR3)

**Comparison of Inorganic Chlorine in the Antarctic and Arctic lowermost stratosphere by separate Late Winter aircraft measurements**

We would like to thank again for the constructive comments on our manuscript. Reviewer comments are in normal font. Our answers are in italic.

Response to Michelle Santee

- L186: the descending generally stops --> descent generally stops

  *Done.*

- L196: unique --> uniquely

  *Done.*

- L254: where no measurements of all major chlorine containing substances are available --> where measurements of all major chlorine-containing substances are not available

  *Done.*

- L273: indirectly determine --> indirectly determined

  *Done.*

- L355: difference between $Cl_{total}$ from controlled substances during PGS and SouthTRAC --> difference between $Cl_{total}$ from controlled substances during PGS and that during SouthTRAC

  *Done.*

- L383: SouhTRAC --> SouthTRAC

  *Done.*

- L384-386: This wording is confusing. I think this would be clearer: "Taking into account the difference in $Cl_{total}$ between the two campaigns would reduce the observed differences in the mid-latitudes, where mixing ratios of $Cl_y$ were higher during PGS, but would increase the differences in the high latitudes, where higher $Cl_y$ was derived for SouthTRAC."

  *We have adopted the suggested sentence.*

- L387: than they can --> than can; difference of $Cl_{total}$ --> difference in $Cl_{total}$

  *Done; Done.*

- L390: "transported" is not the right word; "converted" or "transformed" would be better

  *We agree and have replaced "transported" with "converted".*

- L391-392: It would be more appropriate to state that the results of this work are consistent with those of Konopka et al. (2015), since that study came first: "Therefore, the observed differences in $Cl_y$, with higher $Cl_y$ values in the southern high latitudes than in the northern high latitudes (see Fig. 9 and 10), are consistent with the differences in the age of air found by Konopka et al. (2015)."

  *We have adopted the suggested sentence.*

- L393-394: showed an updated report to the long-term total column data for HCl and $ClONO_2$ (representing $Cl_y$) by Mahieu et al. (2014) --> updated the long-term total column HCl and $ClONO_2$ (representing $Cl_y$) record reported by Mahieu et al. (2014); also: though --> through

  *Done; Done.*

- L396: add a comma after "stations"

  *Done.*

- L401: I do not understand what is meant by "the trends given are indicative of the interhemispheric differences in $Cl_y$". Why would the rates of change in $Cl_y$ necessarily reflect its absolute abundances in the two hemispheres?

  *You are right, the trends do not reflect its absolute abundance in the two hemispheres. Rather, the trends indicate the interhemispheric difference in $Cl_y$ decline. Thus, we include the word decline after $Cl_y$ in this sentence.*

- L402: of about 200 ppt and above --> of 200 ppt or more; also, why is the word "explicitly" used here? It does not seem appropriate and probably should be deleted.

  *Done; "explicitly" was deleted.*

- L442: was found --> were found (in two places)

  *Done.*

- L444-445: reanalyses data --> reanalysis data

  *Done.*